# Adaptive Extrapolated Proximal Gradient Methods with Variance Reduction for Composite Nonconvex Finite-Sum Minimization

## Abstract

This paper proposes AEPG-SPIDER, an Adaptive Extrapolated Proximal Gradient (AEPG) method with variance reduction for minimizing composite nonconvex finite-sum functions. It integrates three acceleration techniques: adaptive stepsizes, Nesterov's extrapolation, and the recursive stochastic path-integrated estimator SPIDER. Unlike existing methods that adjust the stepsize factor using historical gradients, AEPG-SPIDER relies on past iterate differences for its update. While targeting stochastic finite-sum problems, AEPG-SPIDER simplifies to AEPG in the full-batch, non-stochastic setting, which is also of independent interest. To our knowledge, AEPG-SPIDER and AEPG are the first Lipschitz-free methods to achieve optimal iteration complexity for this class of *composite* minimization problems. Specifically, AEPG achieves the optimal iteration complexity of $\mathcal{O}(N\epsilon^{-2})$, while AEPG-SPIDER achieves $\mathcal{O}(N+\sqrt{N}\epsilon^{-2})$ for finding $\epsilon$-approximate stationary points, where $N$ is the number of component functions. Under the Kurdyka-Lojasiewicz (KL) assumption, we establish non-ergodic convergence rates for both methods. Preliminary experiments on sparse phase retrieval and linear eigenvalue problems demonstrate the superior performance of AEPG-SPIDER and AEPG compared to existing methods.

## 1 Introduction

We consider the following composite nonconvex finite-sum minimization problem (where '$\triangleq$' denotes definition):

$$\min_{\mathbf{x}\in\mathbb{R}^n} F(\mathbf{x}) \triangleq f(\mathbf{x}) + h(\mathbf{x}), \text{ where } f(\mathbf{x}) \triangleq \frac{1}{N}\sum_{i=1}^{N} f_i(\mathbf{x}). \tag{1}$$

The function $f(\cdot)$ is assumed to be differentiable, possibly nonconvex. The function $h(\mathbf{x})$ is assumed to be proper and lower semi-continuous, and may be nonconvex, nonsmooth, and non-Lipschitz. Furthermore, we assume the generalized proximal operator of $h(\mathbf{x})$ is easy to compute.

Problem (1) has diverse applications in machine learning. The function $f(\mathbf{x})$ captures empirical loss, including neural network activations, while nonsmooth regularization $h(\mathbf{x})$ prevents overfitting and improves generalization. It incorporates prior information, such as structured sparsity, low-rank properties, discreteness, orthogonality, and non-negativity, enhancing model accuracy. These capabilities extend to various applications, including sparse phase retrieval Cai et al. (2024); Shechtman et al. (2014), eigenvalue problems Wen & Yin (2013), $\ell_2$-weight decay in neural networks Zhang et al. (2019), and network quantization Bai et al. (2019).

**Stochastic Gradient Descent and Variance Reduction Methods**. In many applications, both $n$ and $N$ in Problem (1) are large, making first-order methods the standard choice due to their efficiency. Vanilla gradient descent (GD) requires $\mathcal{O}(N\epsilon^{-2})$ gradient evaluations, while Stochastic Gradient Descent (SGD) demands $\mathcal{O}(N\epsilon^{-4})$ gradient computations in total Ghadimi & Lan (2013); Ghadimi et al. (2016); Ghadimi & Lan (2016). To harness the advantages of both GD and SGD, the variance reduction (VR) framework Johnson & Zhang (2013); Schmidt et al. (2013) was introduced. This framework combines the faster convergence of GD with the lower per-iteration complexity of SGD by decomposing the finite-sum structure into manageable components. VR methods generate low-variance gradient estimates by balancing periodic full-gradient computations with stochastic mini-batch gradients. Notable approaches, including SAGA Defazio et al. (2014); J. Reddi et al. (2016), SVRG Johnson & Zhang (2013); Li & Li (2018), SARAH Nguyen et al. (2017), SPIDER Fang et al. (2018), SNVRG Zhou et al. (2020), and PAGE Li et al. (2021), have been developed. While earlier

work achieved an iteration complexity of $\mathcal{O}(N + N^{2/3}\epsilon^{-2})$ with a suboptimal dependence on $N$, recent methods Fang et al. (2018); Pham et al. (2020) have improved this to the optimal iteration complexity of $\mathcal{O}(N + N^{1/2}\epsilon^{-2})$.

**Proximal Gradient Methods and Their Accelerated Variants**. Proximal Gradient Methods (PGMs) Nesterov (2003) are widely used for solving composite optimization problems of the form in Problem (1). Each iteration of PGM consists of a gradient descent step on $f(\cdot)$, followed by a proximal mapping with respect to $h(\cdot)$. Accelerated variants of PGMs Beck & Teboulle (2009); Ochs et al. (2014); Pock & Sabach (2016) enhance convergence by incorporating extrapolation or inertial steps inspired by Nesterov's acceleration techniques. These methods exploit momentum from previous iterates to achieve faster convergence–specifically, the optimal rate of $\mathcal{O}(1/k^2)$ for smooth convex functions, compared to the standard $\mathcal{O}(1/k)$ rate of PGMs. Extensions of PGMs to nonconvex and stochastic settings have also been developed Ghadimi & Lan (2016); Li & Lin (2015); J. Reddi et al. (2016); Li & Li (2018); Wen et al. (2017), making them powerful tools for large-scale optimization problems. In particular, they have been successfully applied to training deep neural networks Sutskever et al. (2013), where they improve convergence efficiency with little to no increase in computational cost.

**Adaptive Stepsizes and Coordinate-Wise Scaling**. The choice of stepsize is critical in optimization, influencing both convergence speed and stability. Traditional fixed or manually tuned stepsizes often struggle with complex non-convex problems, leading to suboptimal performance. Although line-search or backtracking methods can improve robustness, they typically incur high computational costs, particularly in the finite-sum setting considered here. Adaptive stepsize methods McMahan & Streeter (2010); Duchi et al. (2011), such as Adam Kingma & Ba (2015); Chen et al. (2022), and AdaGrad Duchi et al. (2011), mitigate these issues by dynamically adjusting the learning rate based on gradient information. Recent advancements, including Polyak stepsize Polyak (1987); Wang et al. (2023); Jiang & Stich (2023), Barzilai-Borwein stepsize Barzilai & Borwein (1988); Zhou et al. (2024), scaled stepsize Oikonomidis et al. (2024), and D-adaptation Defazio & Mishchenko (2023), have primarily focused on convex optimization. This work extends adaptive stepsize techniques Duchi et al. (2011) to address composite non-convex finite-sum problems. To improve adaptivity across different coordinates, we adopt coordinate-wise stepsizes, a strategy also employed in popular methods like Adam. Unlike global stepsizes that use a single scalar learning rate as in AdaGrad-Norm, coordinate-wise approaches rely on diagonal preconditioners that scale updates individually across coordinates based on accumulated gradient statistics. This mechanism assigns larger stepsizes to coordinates with smaller accumulated gradient magnitudes, enabling faster learning in infrequently updated or sparse dimensions. Such adaptivity is particularly beneficial in large-scale settings involving sparse or structured models Duchi et al. (2011); Yun et al. (2021); Bai et al. (2019).

**Theory on Nonconvex Optimization**. (***i***) Iteration complexity. We aim to establish the iteration complexity of nonconvex optimization algorithms, i.e., the number of iterations required to find an $\epsilon$-approximate first-order stationary point $\tilde{\mathbf{x}}$ satisfying $\text{dist}(\mathbf{0}, \partial(f+g)(\tilde{\mathbf{x}})) \leq \epsilon$. However, the iteration complexity of adaptive stepsize methods for solving Problem (1) remains unknown. Existing related work, such as AdaGrad-Norm Ward et al. (2020), AGD Kavis et al. (2022a), STORM Cutkosky & Orabona (2019); Jiang et al. (2024), and ADA-SPIDER Kavis et al. (2022b), only addresses the special case $h(\cdot) = 0$, while methods such as APG Li & Lin (2015), ProxSVRG J. Reddi et al. (2016), Spider Fang et al. (2018), SpiderBoost Wang et al. (2019), and ProxSARAH Pham et al. (2020) rely on non-adaptive stepsizes. Our proposed methods, AEPG-SPIDER and AEPG, with and without variance reduction, respectively, address the general case where $h(\mathbf{x})$ is nonconvex, using an adaptive stepsize strategy. Additionally, our methods incorporate Nesterov's extrapolation and leverage coordinate-wise stepsize techniques. (***ii***) Last-iterate convergence rate. The work of Attouch & Bolte (2009) establishes a unified framework to prove the convergence rates of descent methods under the Kurdyka-Lojasiewicz (KL) assumption for problem (1). Recent works Qian & Pan (2023); Yang (2023) extend this to nonmonotone descent methods. Inspired by these works, we establish the optimal iteration complexity and derive non-ergodic convergence rates for our methods.

**Existing Challenges**. We study the composite nonconvex finite-sum minimization problem in Problem (1), which presents three key challenges. (i) Adaptive stepsize for composite minimization. Given that $h(\cdot)$ can be nonsmooth and non-Lipschitz, the subdifferential $\partial h(x)$ becomes a set, and there is no principled criterion for selecting a representative subgradient. As a result, subgradient

Table 1: Comparison among existing methods for composite nonconvex funite-sum minimization. The notation $\tilde{\mathcal{O}}(\cdot)$ hides polylogarithmic factors, while $\mathcal{O}(\cdot)$ hides constants.

| | Adaptive Stepsize | Nonconvex $h(\mathbf{x})$ | Nesterov Extrapol. | Coordinate -Wise | Iteration Complexity | Last-Iterate Conv. Rate |
|---|---|---|---|---|---|---|
| APG Li & Lin (2015) | ✗ | ✔ | ✔ | ✗ | $\mathcal{O}(N/\epsilon)$ | ✔ |
| SVRG-APG Li et al. (2017) | ✗ | ✔ | ✔ | ✗ | unknown[a] | ✔ |
| ProxSVRG J. Reddi et al. (2016) | ✗ | ✗ | ✔ | ✗ | $\mathcal{O}(N + N^{2/3}\epsilon^{-2})$ | unknown |
| SPIDER Fang et al. (2018) | ✗ | ✗ | ✗ | ✔ | $\mathcal{O}(N + \sqrt{N}\epsilon^{-2})$ | unknown |
| SpiderBoost Wang et al. (2019) | ✗ | ✔ | ✔ | ✗ | $\mathcal{O}(N + \sqrt{N}\epsilon^{-2})$ | unknown |
| ProxSARAH Pham et al. (2020) | ✗[b] | ✗ | ✗ | ✗ | $\mathcal{O}(N + \sqrt{N}\epsilon^{-2})$ | unknown |
| AdaGrad-Norm Ward et al. (2020) | ✔ | ✗ | ✗ | ✗ | $\mathcal{O}(N\epsilon^{-2})$ | unknown |
| AGD Kavis et al. (2022a) | ✔ | ✗ | ✔ | ✗ | $\mathcal{O}(N\epsilon^{-2})$ | unknown |
| ADA-SPIDER Kavis et al. (2022b) | ✔ | ✗ | ✗ | ✗ | $\tilde{\mathcal{O}}(N + \sqrt{N}\epsilon^{-2})$ | unknown |
| AEPG [ours] | ✔ | ✔ | ✔ | ✔ | $\mathcal{O}(N\epsilon^{-2})$ | ✔ [Theorem 4.8] |
| AEPG-SPIDER [ours] | ✔ | ✔ | ✔ | ✔ | $\mathcal{O}(N + \sqrt{N}\epsilon^{-2})$ | ✔ [Theorem 4.13] |

[a] This work only demonstrates that any cluster point is a critical point but fail to establish the iteration complexity.
[b] This algorithm relies on the Lipschitz constant and does not qualify as using adaptive stepsizes in our strict sense.

norms, as used in methods like AdaGrad-Norm, are not well-defined in this context, making them unsuitable for stepsize adaptation. To address this, AEPG adaptively updates the stepsize factor based on *the differences between past iterates*, and integrates both global and local learning rates for improved robustness. (ii) Nesterov's extrapolation. To achieve possible acceleration, we introduce *a novel recursive update rule* for the extrapolation coefficient: $\sigma^t = \theta(1 - \sigma^{t-1})\min(\mathbf{v}^t \div \mathbf{v}^{t+1})$ with $\sigma^t \in (0, \infty)$. To our knowledge, AEPG is the first adaptive gradient method that incorporates extrapolation for this class of nonconvex problems. Its convergence is established via a newly developed sufficient descent condition. (iii) Finite-sum structure. To efficiently handle the finite-sum setting, we incorporate the SPIDER estimator for variance reduction, which yields optimal iteration complexity.

**Contributions**. We provide a detailed comparison of existing methods for composite nonconvex finite-sum minimization in Table 1. Our main contributions are summarized as follows. (***i***) We proposes AEPG-SPIDER, an Adaptive Extrapolated Proximal Gradient method with variance reduction for composite nonconvex finite-sum optimization. It integrates adaptive stepsizes, Nesterov's extrapolation, and the SPIDER estimator for fast convergence. In the full-batch setting, it simplifies to AEPG, which is of independently significant (see Section 2). (***ii***) We prove that AEPG attains the iteration complexity of $\mathcal{O}(N\epsilon^{-2})$, while AEPG-SPIDER achieves $\mathcal{O}(N + \sqrt{N}\epsilon^{-2})$ for finding an $\epsilon$-stationary point, thereby establishing them as the first Lipschitz-free methods (i.e., methods that do not rely on any Lipschitz constant) with optimal complexity for composite minimization (see Section 3). (***iii***) Under the Kurdyka-Lojasiewicz (KL) assumption, we prove that our algorithm terminates in finitely many iterations when $\tilde{\sigma} = 0$, converges linearly for $\tilde{\sigma} \in (0, \frac{1}{2}]$, and sublinearly for $\tilde{\sigma} \in (\frac{1}{2}, 1)$, with convergence measured by the iterate gap (see Section 4). (***iv***) We validate our approaches through experiments on sparse phase retrieval and the linear eigenvalue problem, showcasing its effectiveness (see Section 5).

**Notations**. Vector operations are performed element-wise. Specifically, for any $\mathbf{x}, \mathbf{y} \in \mathbb{R}^n$, the operations $(\mathbf{x} + \mathbf{y})$, $(\mathbf{x} - \mathbf{y})$, $(\mathbf{x} \odot \mathbf{y})$, and $(\mathbf{x} \div \mathbf{y})$ represent element-wise addition, subtraction, multiplication, and division, respectively. We use $\|\mathbf{x}\|_{\mathbf{v}}$ to denote the generalized vector norm, defined as $\|\mathbf{x}\|_{\mathbf{v}} = \sqrt{\sum_{i=1} \mathbf{x}_i^2 \mathbf{v}_i}$. The notations, technical preliminaries, and relevant lemmas are provided in Appendix Section A.

## 2 THE PROPOSED ALGORITHMS

This section provides the proposed AEPG-SPIDER algorithm, an Adaptive Extrapolated Proximal Gradient method with variance reduction for solving Problem (1). Notably, AEPG-SPIDER reduces to AEPG in the full-batch, non-stochastic setting.

First of all, our algorithms are based on the following assumptions imposed on Problem (1).

**Assumption 2.1.** *The generalized proximal operator:* $\text{Prox}_h(\mathbf{a}; \mathbf{v}) \triangleq \arg\min_{\mathbf{x}} h(\mathbf{x}) + \frac{1}{2}\|\mathbf{x} - \mathbf{a}\|_{\mathbf{v}}^2$ *can be exactly and efficiently for all* $\mathbf{a}, \mathbf{v} \in \mathbb{R}^n$.

**Remark 2.2.** *(i) Assumption 2.1 is commonly employed in nonconvex proximal gradient methods. (ii) When* $\mathbf{v} = \mathbf{1}$, *the diagonal preconditioner reduces to the identity preconditioner. Assumption 2.1 holds for certain functions of* $h(\mathbf{x})$. *Common examples include capped-$\ell_1$ penalty Zhang (2010b), log-sum penalty Candes et al. (2008), minimax concave penalty Zhang (2010a), Geman penalty Geman & Yang (1995), $\ell_p$ regularization with $p \in \{0, \frac{1}{2}, \frac{2}{3}, 1\}$, and indicator functions for cardinality*

---

**Algorithm 1** Proposed AEPG and AEPG-SPIDER

---

1: Initialize $\mathbf{x}^0$. Let $\mathbf{x}^{-1} = \mathbf{x}^0$.
2: Let $\underline{v} > 0$, $\alpha > 0$, $\beta \geq 0$, and $\theta \in [0, 1)$.
3: Set $\bar{\mathbf{v}}^0 = \underline{v}\mathbf{1}$, $\mathbf{y}^0 = \mathbf{x}^0$, $\sigma^{-1} = \theta$.
4: **for** $t = 0$ **to** $T$ **do**
5:     Option AEPG: Compute $\mathbf{g}^t = \nabla f(\mathbf{y}^t)$.
6:     Option AEPG-SPIDER: Compute $\mathbf{g}^t$ using (2).
7:     Let $\mathbf{x}^{t+1} \in \text{Prox}_h(\mathbf{y}^t - \mathbf{g}^t \div \mathbf{v}^t; \mathbf{v}^t)$, $\mathbf{d}^t \triangleq \mathbf{x}^{t+1} - \mathbf{x}^t$.
8:     Let $\mathbf{s}^t \triangleq \alpha\|\mathbf{r}^t\|_2^2 \cdot \mathbf{1} + \beta\mathbf{r}^t \odot \mathbf{r}^t$, where $\mathbf{r}^t \triangleq \mathbf{v}^t \odot \mathbf{d}^t$.
9:     Set $\mathbf{v}^{t+1} = \sqrt{\mathbf{v}^0 \odot \mathbf{v}^0 + \sum_{i=0}^t \mathbf{s}^i}$.
10:     Let $\sigma^t \triangleq \theta(1 - \sigma^{t-1}) \cdot \min(\mathbf{v}^t \div \mathbf{v}^{t+1})$.
11:     Set $\mathbf{y}^{t+1} = \mathbf{x}^{t+1} + \sigma^t\mathbf{d}^t$.
12: **end for**

---

*constraints, orthogonality constraints in matrices, and rank constraints in matrices. **(iii)** When $\mathbf{v}$ is a general vector, the variable metric operator can still be evaluated for certain coordinate-wise separable functions of $h(\mathbf{x})$. Examples includes the $\ell_p$ norm with $p \in \{0, \frac{1}{2}, \frac{2}{3}, 1\}$ (with or without bound constraints) Yun et al. (2021), and W-shaped regularizer Bai et al. (2019).*

Given any solution $\mathbf{y}^t$, we use the SPIDER estimator, introduced by Fang et al. (2018), to approximate its stochastic gradient:

$$\mathbf{g}^t = \begin{cases} \nabla f(\mathbf{y}^t), & \text{mod}(t, q) = 0; \\ \mathbf{g}^{t-1} + \nabla f(\mathbf{y}^t; \mathcal{I}^t) - \nabla f(\mathbf{y}^{t-1}; \mathcal{I}^t), & \text{else.} \end{cases} \tag{2}$$

Here, $q$ is an integer frequency parameter that determines how often the full gradient is computed, and $\nabla f(\mathbf{y}; \mathcal{I}^t)$ denotes the average gradient computed over the examples in $\mathcal{I}^t$ at the point $\mathbf{y}$. The mini-batch $\mathcal{I}^t$ is sampled uniformly at random (with replacement) from the index set $\{1, 2, ..., N\}$ with $|\mathcal{I}^t| = b$ for all $t$, where $b$ is the mini-batch size parameter.

The proposed algorithm, AEPG, and its variant, AEPG-SPIDER, form an adaptive proximal gradient optimization framework designed for composite optimization problems. This framework initializes parameters and iteratively updates the solution by computing gradients (either directly or via a variance-reduced SPIDER estimator) and applying a proximal operator. Unlike existing methods that adjust the stepsize factor using historical gradients, AEPG and AEPG-SPIDER update the stepsize factor $\mathbf{v}^t$ dynamically using differences between successive iterates. Additionally, the algorithm incorporates momentum-like updates through the extrapolation parameter $\sigma^t$ to improve convergence speed. These algorithms are designed for efficient and adaptive optimization in both deterministic and stochastic settings. We present AEPG and AEPG-SPIDER in Algorithm 1.

We compare the proposed AEPG with AdaGrad-Norm Ward et al. (2020) by examining a special case for AEPG where $h(\cdot) = 0$ and $\beta = \theta = 0$. The first-order optimality condition for $\mathbf{x}^{t+1}$ becomes: $\mathbf{0} \in \partial h(\mathbf{x}^{t+1}) + \mathbf{v}^t \odot (\mathbf{x}^{t+1} - \mathbf{a}^t)$, where $\mathbf{a}^t = \mathbf{y}^t - \mathbf{g}^t \div \mathbf{v}^t$ and $\mathbf{y}^t = \mathbf{x}^t$. This leads to $\mathbf{v}^t \odot (\mathbf{x}^{t+1} - \mathbf{x}^t) = -\mathbf{g}^t$. Consequently, the update rule for $\mathbf{v}^t$ reduces to $\mathbf{v}^{t+1} = \sqrt{(\mathbf{v}^0)^2 + \alpha \sum_{i=0}^t \|\mathbf{g}^i\|_2^2}$, which resembles "lazy" version of the update used in AdaGrad-Norm that $\mathbf{v}^{t+1} = \sqrt{(\mathbf{v}^0)^2 + \alpha \sum_{i=0}^{t+1} \|\mathbf{g}^i\|_2^2}$. There are three key differences between AEPG and AdaGrad-Norm. **(i)** Update Strategy. AdaGrad-Norm adapts the stepsize based on accumulated gradient norms, while AEPG uses differences between past iterates to update $\mathbf{v}^{t+1}$, resulting in a lazy update scheme. The difference between $\mathbf{v}^t$ and $\mathbf{v}^{t+1}$ plays a central role in our complexity analysis. **(ii)** Extrapolation. Unlike AdaGrad-Norm, AEPG is the first adaptive (proximal) gradient method with extrapolation for this class of nonconvex problem. A novel recursive update rule for the extrapolation coefficient $\sigma^t$ is proposed: $\sigma^t = \theta(1 - \sigma^{t-1}) \min(\mathbf{v}^t \div \mathbf{v}^{t+1})$. **(iii)** Coordinate-wise stepsizes. While AdaGrad-Norm employs a global learning rate, AEPG combines both global and coordinate-wise learning rates. The global scaling factor $\alpha > 0$ ensures that $\mathbf{v}^t$ remains well-conditioned, satisfying $\frac{\max(\mathbf{v}^t)}{\min(\mathbf{v}^t)} \leq \kappa$ for some constant $\kappa \geq 1$.

Finally, we have the following additional remarks on Algorithm 1. (***i***) The cumulative update rule for $\mathbf{v}^{t+1}$ can be equivalently be written as the recursive formula $\mathbf{v}^{t+1} = \sqrt{\mathbf{v}^t \odot \mathbf{v}^t + \mathbf{s}^t}$. (***ii***) We address the non-smoothness of $h(\mathbf{x})$ using its (generalized) proximal operator, the basis of proximal gradient methods, which update the parameter via the gradient of $f(\mathbf{x})$ followed by a (generalized) proximal mapping of $h(\mathbf{x})$. (***iii***) The proximal mapping step incorporates an extrapolated point, combining the current and previous points, following the Nesterov's extrapolation method. (***iv***) The parameter $\alpha > 0$ is required for theoretical convergence guarantees. In practice, setting $\alpha$ to a very small value (e.g., $\alpha = 10^{-4}$) essentially reduces the update to coordinate-wise stepsizes. (***v***) Algorithm 1 involves four parameters: the initial value $\mathbf{v}^0$, the global and local learning rate multipliers $\alpha > 0$ and $\beta \geq 0$, and the extrapolation parameter $\theta$. By default, we typically set $\mathbf{v}^0$ and $\alpha$ to small positive values, $\beta \in \{0, 1\}$, and choose $\theta$ to be a value close to but strictly less than 1.

## 3 ITERATION COMPLEXITY

This section details the iteration complexity of AEPG and AEPG-SPIDER. AEPG-SPIDER generates a random output $\mathbf{x}^t$ with $t = \{0, 1, \ldots\}$, based on the observed realizations of the random variable $\varsigma^{t-1} \triangleq \{\mathcal{I}^0, \mathcal{I}^1, \ldots, \mathcal{I}^{t-1}\}$. The expectation of a random variable is denoted by $\mathbb{E}_{\varsigma^{t-1}}[\cdot] = \mathbb{E}[\cdot]$, where the subscript is omitted for simplicity.

In the sequel of the paper, we make the following assumptions.

**Assumption 3.1.** *There exists a universal positive constant $\bar{\mathrm{x}}$ such that $\|\mathbf{x}\| \leq \bar{\mathrm{x}}$ for all $\mathbf{x} \in \mathrm{dom}(F)$. Furthermore, we assume $\min_{\mathbf{x}} F(\mathbf{x}) > -\infty$.*

**Assumption 3.2.** *Each $f_j(\cdot)$ is $L$-smooth, meaning that $\|\nabla f_j(\mathbf{x}) - \nabla f_j(\tilde{\mathbf{x}})\| \leq L \|\mathbf{x} - \tilde{\mathbf{x}}\|$ for all $j \in [N]$. This property extends to $f(\mathbf{x})$, which is also $L$-smooth.*

**Remark 3.3.** *(i) Assumption 3.1 holds by setting $h(\mathbf{x}) = \iota_\Omega(\mathbf{x})$, where $\iota_\Omega(\cdot)$ is the indicator function of a compact set $\Omega$. For example, it can be enforced by adding simple bound constraints, as in our application. Since any $\mathbf{x} \in \mathrm{dom}(F) \triangleq \{\mathbf{x} : F(\mathbf{x}) < +\infty\}$ is feasible, it follows that $\|\mathbf{x}\| \leq \bar{\mathrm{x}}$ for some $\bar{\mathrm{x}} > 0$. This bounded-domain assumption is standard and mild, widely used in convex composite minimization Liu et al. (2022); Jaggi (2013), non-convex composite minimization Yun et al. (2021), fractional minimization Yuan (2025), and minimax optimization Xu et al. (2023). Without it, meaningful theoretical guarantees are generally intractable. (iii) Assumption 3.2 is a standard requirement in the convergence analysis of nonconvex algorithms.*

For notational convenience, we define $\mathcal{Z}_t \triangleq \mathbb{E}[F(\mathbf{x}^t) - F(\bar{\mathbf{x}}) + \frac{1}{2}\|\mathbf{x}^t - \mathbf{x}^{t-1}\|^2_{\sigma^{t-1}(\mathbf{v}^t + L)}]$, where $\bar{\mathbf{x}} \in \arg\min_{\mathbf{x}} F(\mathbf{x})$. We define $\mathcal{V}_{t+1} \triangleq \sqrt{\underline{\mathrm{v}}^2 + (\alpha + \beta)\mathcal{R}_t}$, where $\mathcal{R}_t \triangleq \sum_{i=0}^t \|\mathbf{r}^i\|^2_2$.

### 3.1 ANALYSIS FOR AEPG

This subsection provides the convergence analysis of AEPG.

We begin with a high-level overview of the proof strategy for AEPG. First, leveraging the optimality of $\mathbf{x}^{t+1}$, the $L$-smoothness of $f(\mathbf{x})$, and the recursive update rule of $\sigma^t$, we derive the following sufficient decrease condition: $\mathcal{Z}_{t+1} - \mathcal{Z}_t \leq c_2 \mathbb{S}_2^t - c_1 \mathbb{S}_1^t$, where $c_1, c_2 > 0$, $\mathbb{S}_1^t \triangleq \frac{\|\mathbf{r}^t\|^2_2}{\min(\mathbf{v}^t)}$, and $\mathbb{S}_2^t \triangleq \frac{\|\mathbf{r}^t\|^2_2}{\min(\mathbf{v}^t)^2}$. Second, based on the update rule for $\mathbf{v}^t$, we establish the bounds $\sum_{t=0}^T \mathbb{S}_1^t \leq \mathcal{O}(\mathcal{V}_{T+1})$ and $\sum_{t=0}^T \mathbb{S}_2^t \leq \mathcal{O}(\sqrt{\mathcal{V}_{T+1}})$. Finally, by analyzing *both* the telescoping sum $\sum_{t=0}^T (\mathcal{Z}_{t+1} - \mathcal{Z}_t)$ and the weighted sum $\sum_{t=0}^T \min(\mathbf{v}^t)(\mathcal{Z}_{t+1} - \mathcal{Z}_t)$, we establish the boundedness of both $\mathcal{Z}_t$ and $\mathcal{V}_t$.

We obtain the following lemma which is crucial to our analysis.

**Lemma 3.4.** *(Proof in Section B.2.1,* Boundedness of $\mathcal{Z}_t$ and $\mathcal{V}_t$) For all $t \geq 0$, we obtain: **(a)** It holds $\mathcal{Z}_t \leq \overline{\mathcal{Z}}$ for some positive constant $\overline{\mathcal{Z}}$. **(b)** It holds $\mathcal{V}_t \leq \overline{\mathrm{v}}$ for some positive constant $\overline{\mathrm{v}}$.

Finally, we present the following results on iteration complexity.

**Theorem 3.5.** *(Proof in Section B.2.2,* Iteration Complexity). Let the sequence $\{\mathbf{x}^t\}_{t=0}^T$ be generated by AEPG.

**(a)** We have $\sum_{t=0}^T \|\mathbf{x}^{t+1} - \mathbf{x}^t\|^2_2 \leq \overline{\mathrm{X}} \triangleq \frac{1}{\alpha}((\overline{\mathrm{v}}/\underline{\mathrm{v}})^2 - 1)$.

**(b)** We have $\frac{1}{T+1}\sum_{t=0}^T \|\nabla f(\mathbf{x}^{t+1}) + \partial h(\mathbf{x}^{t+1})\| = \mathcal{O}(1/\sqrt{T})$. In other words, there exists $\bar{t} \in [T]$ such that $\|\nabla f(\mathbf{x}^{\bar{t}}) + \partial h(\mathbf{x}^{\bar{t}})\| \leq \epsilon$, provided $T \geq \mathcal{O}(\frac{1}{\epsilon^2})$.

**Remark 3.6.** *Theorem 3.5 establishes the first optimal iteration complexity result for Lipschitz-free methods in deterministically minimizing composite functions.*

### 3.2 ANALYSIS FOR AEPG-SPIDER

This subsection provides the convergence analysis of AEPG-SPIDER.

We provide a high-level overview of the proof strategy for AEPG-SPIDER. First, we derive a sufficient decrease condition of the form $\mathcal{Z}_{t+1} - \mathcal{Z}_t \leq \mathbb{E}[c_2' \mathbb{S}_2^t - c_1 \mathbb{S}_1^t + \frac{c_3}{q} \cdot \sum_{i=(r_t-1)q}^{t-1} Y_i]$, where $c_1, c_2', c_3 > 0$, $\mathbb{S}_1^t \triangleq \frac{\|\mathbf{r}^t\|_2^2}{\min(\mathbf{v}^t)}$, $\mathbb{S}_2^t \triangleq \frac{\|\mathbf{r}^t\|_2^2}{\min(\mathbf{v}^t)^2}$, and $Y_i \triangleq \mathbb{E}[\|\mathbf{y}^{i+1} - \mathbf{y}^i\|_2^2]$. Second, using the update rule for $\mathbf{v}^t$, we derive the following upper bounds: $\sum_{t=0}^T V_t Y_t \leq \mathcal{O}(\mathbb{E}[\mathcal{V}_{T+1}])$ and $\sum_{t=0}^T Y_t \leq \mathcal{O}(\mathbb{E}[\sqrt{\mathcal{V}_{T+1}}])$, where $V_j = \min(\mathbf{v}^j)$. Lastly, by analyzing *both* the telescoping sum $\sum_{t=0}^T (\mathcal{Z}_{t+1} - \mathcal{Z}_t)$ and the weighted variant $\sum_{t=0}^T \min(\mathbf{v}^t)(\mathcal{Z}_{t+1} - \mathcal{Z}_t)$, we establish the boundedness of both $\mathcal{Z}_t$ and $\mathcal{V}_t$.

We derive the following critical lemma, which is analogous to Lemma 3.4.

**Lemma 3.7.** *(Proof in Appendix B.3.1,* Boundedness of $\mathcal{Z}_t$ and $\mathcal{V}^t$*) For all $t \geq 0$, we have:* **(a)** *It holds $\mathbb{E}[\mathcal{Z}_t] \leq \overline{\mathcal{Z}}$ for some positive constant $\overline{\mathcal{Z}}$.* **(b)** *It holds $\mathbb{E}[\mathcal{V}^t] \leq \overline{\mathrm{v}}$ for some positive constant $\overline{\mathrm{v}}$.*

Finally, we provide the following results on iteration complexity.

**Theorem 3.8.** *(Proof in Section B.3.2,* Iteration Complexity*). Let the sequence $\{\mathbf{x}^t\}_{t=0}^T$ be generated by Algorithm 1.*

(a) *We have $\mathbb{E}[\sum_{t=0}^T \|\mathbf{x}^{t+1} - \mathbf{x}^t\|_2^2] \leq \overline{\mathrm{X}} \triangleq \frac{1}{\alpha}((\overline{\mathrm{v}}/\underline{\mathrm{v}})^2 - 1)$.*

(b) *We have $\mathbb{E}[\frac{1}{T+1} \sum_{t=0}^T \|\nabla f(\mathbf{x}^{t+1}) + \partial h(\mathbf{x}^{t+1})\|] \leq \mathcal{O}(1/\sqrt{T})$. In other words, there exists $\overline{t} \in [T]$ such that $\mathbb{E}[\|\nabla f(\mathbf{x}^{\overline{t}}) + \partial h(\mathbf{x}^{\overline{t}})\|] \leq \epsilon$, provided $T \geq \frac{1}{\epsilon^2}$.*

(c) *Assume $b = q = \sqrt{N}$. The total iteration complexity required to find an $\epsilon$-approximate critical point, satisfying $\mathbb{E}[\|\nabla f(\mathbf{x}^{\overline{t}}) + \partial h(\mathbf{x}^{\overline{t}})\|] \leq \epsilon$, is given by $\mathcal{O}(N + \sqrt{N}\epsilon^{-2})$.*

**Remark 3.9.** *(i) The work of Kavis et al. (2022b) introduces the first Lipschitz-free variance-reduced method,* ADA-SPIDER*, for solving Problem (1) with $h(\cdot) = 0$. However, its iteration complexity, $\tilde{\mathcal{O}}(N + \sqrt{N}\epsilon^{-2})$, is sub-optimal. In contrast, the proposed* AEPG-SPIDER *successfully* eliminates the logarithmic factor *in* ADA-SPIDER*, achieving optimal iteration complexity. The core novelty of our approach is a recursive bounding inequality of the form $\mathcal{Z}_T \leq \dot{a} + \dot{b}\sqrt{\max_{t=0}^{T-1} \mathcal{Z}_t}$, where $\dot{a}, \dot{b} > 0$ (see Inequalities (23, 39) in the Appendix). (ii) Theorem 3.8 establishes the first optimal iteration complexity result for Lipschitz-free methods in minimizing composite finite-sum functions.*

## 4 CONVERGENCE RATE

This section presents the convergence rates of AEPG and AEPG-SPIDER, leveraging the non-convex analysis tool known as the Kurdyka-Lojasiewicz (KL) assumption Attouch et al. (2010); Bolte et al. (2014); Li & Lin (2015); Li et al. (2023); Qian & Pan (2023).

We make the following additional assumption.

**Assumption 4.1.** *The function $\mathcal{Z}(\mathbf{x}, \mathbf{x}', \sigma, \mathbf{v}) \triangleq F(\mathbf{x}) - F(\overline{\mathbf{x}}) + \frac{1}{2}\|\mathbf{x} - \mathbf{x}'\|_{\sigma(\mathbf{v}+L)}^2$ is a KL function with respect to $\mathbb{W} \triangleq \{\mathbf{x}, \mathbf{x}', \sigma, \mathbf{v}\}$.*

We present the following useful lemma, due to Attouch et al. (2010); Bolte et al. (2014).

**Lemma 4.2.** *(Kurdyka-Łojasiewicz Inequality). For a KL function $\mathcal{Z}(\mathbb{W})$ with $\mathbb{W} \in \mathrm{dom}(\mathcal{Z})$, there exists $\tilde{\eta} \in (0, +\infty)$, $\tilde{\sigma} \in [0, 1)$, a neighborhood $\Upsilon$ of $\mathbb{W}^\infty$, and a continuous concave desingularization function $\varphi(s) \triangleq \tilde{c} s^{1-\tilde{\sigma}}$ with $\tilde{c} > 0$ and $s \in [0, \tilde{\eta})$ such that, for all $\mathbb{W} \in \Upsilon$ satisfying $\mathcal{Z}(\mathbb{W}) - \mathcal{Z}(\mathbb{W}^\infty) \in (0, \tilde{\eta})$, it holds that: $\varphi'(\mathcal{Z}(\mathbb{W}) - \mathcal{Z}(\mathbb{W}^\infty)) \cdot \mathrm{dist}(\mathbf{0}, \partial\mathcal{Z}(\mathbb{W})) \geq 1$.*

**Remark 4.3.** *All semi-algebraic and subanalytic functions satisfy the KL assumption. Examples of semi-algebraic functions include real polynomial functions, $\|\mathbf{x}\|_p$ for $p \geq 0$, the rank function, the indicator function of Stiefel manifolds, and the positive-semidefinite cone.*

We provide the following lemma on subgradient bounds at each iteration.

**Lemma 4.4.** *(Proof in Appendix C.1,* **Subgradient Lower Bound for the Iterates Gap**) *We define $\mathbb{W}^t = \{\mathbf{x}^t, \mathbf{x}^{t-1}, \sigma^{t-1}, \mathbf{v}^t\}$. We have $\|\partial\mathcal{Z}(\mathbb{W}^{t+1})\| \leq \vartheta(\|\mathbf{x}^{t+1} - \mathbf{x}^t\| + \|\mathbf{x}^t - \mathbf{x}^{t-1}\|)$, where $\vartheta > 0$ is a constant.*

## 4.1 Analysis for AEPG

This subsection presents the convergence rate for AEPG. We define $X_t \triangleq \|\mathbf{x}^t - \mathbf{x}^{t-1}\|$, and $S_t \triangleq \sum_{j=t}^{\infty} X_{j+1}$. The following assumption is used in the analysis.

**Assumption 4.5.** *There exists a sufficiently large index $t_\star$ such that $\xi \triangleq c_1 \min(\mathbf{v}^{t_\star}) - c_2 > 0$, where $c_1 \triangleq \frac{1}{2}(\frac{1-\theta}{\kappa})^2$, $c_2 \triangleq \frac{3L}{2}$, and $\kappa \triangleq 1 + \sqrt{\beta/\alpha}$.*

**Remark 4.6.** *Assumption 4.5 holds if $\min(\mathbf{v}^{t_\star}) > \frac{c_2}{c_1} = \frac{3\kappa^2 L}{(1-\theta)^2}$, which requires $\min(\mathbf{v}^{t_\star})$ to be over a multiple of $L$ and is relatively mild.*

We establish a finite-length property of AEPG, which is significantly stronger than the result in Theorem 3.5.

**Theorem 4.7.** *(Proof in Appendix C.2, **Finite-Length Property**). We define $\varphi_t \triangleq \varphi(\mathcal{Z}(\mathbb{W}^t) - \mathcal{Z}(\mathbb{W}^\infty))$. We define $\vartheta$ in Lemma 4.4. We define $\xi$ in Assumption 4.5. For all $t \geq t_\star$, we have:*

  **(a)** It holds that $X_{t+1}^2 \leq \frac{\vartheta}{\xi}(X_t + X_{t-1})(\varphi_t - \varphi_{t+1})$.
  **(b)** It holds that $\forall i \geq t$, $S_i \leq \varpi(X_i + X_{i-1}) + \varpi\varphi_i$, where $\varpi > 0$ is some constant. The sequence $\{X_j\}_{j=t}^\infty$ has the finite length property that $S_t$ is always upper-bounded by a certain constant, i.e., $\sum_{j=i}^\infty X_{j+1} < +\infty$ for all $i$.

Finally, we establish the last-iterate convergence rate for AEPG.

**Theorem 4.8.** *(Proof in Appendix C.3, **Convergence Rate**). There exists $t'$ such that for all $t \geq t'$, we have:*

  **(a)** If $\tilde{\sigma} = 0$, then the sequence $\mathbf{x}^t$ converges in a finite number of steps.
  **(b)** If $\tilde{\sigma} \in (0, \frac{1}{2}]$, then there exist $\acute{\varsigma} \in (0, 1)$ such that $\|\mathbf{x}^t - \mathbf{x}^\infty\| = \mathcal{O}(\varsigma^t)$.
  **(c)** If $\tilde{\sigma} \in (\frac{1}{2}, 1)$, then it follows that $\|\mathbf{x}^t - \mathbf{x}^\infty\| \leq \mathcal{O}(t^{-\acute{\varsigma}})$, where $\acute{\varsigma} \triangleq \frac{1-\tilde{\sigma}}{2\tilde{\sigma}-1} > 0$.

**Remark 4.9.** *(i) Under Assumption 4.2, with the desingularizing function $\varphi(t) = \tilde{c}t^{1-\tilde{\sigma}}$ for some $\tilde{c} > 0$ and $\tilde{\sigma} \in [0, 1)$, Theorem 4.8 establishes that AEPG converges in a finite number of iterations when $\tilde{\sigma} = 0$, achieves linear convergence for $\tilde{\sigma} \in (0, \frac{1}{2}]$, and exhibits sublinear convergence for $\tilde{\sigma} \in (\frac{1}{2}, 1)$ in terms of the gap $\|\mathbf{x}^t - \mathbf{x}^\infty\|$. These findings are consistent with the results reported in Attouch et al. (2010). (ii)Unlike Qian & Pan (2023); Yang (2023), which employ a fixed extrapolation parameter $\sigma^t$, our approach introduces a novel recursive update rule, leading to fundamentally different strategies and analysis.*

## 4.2 Analysis for AEPG-SPIDER

This subsection presents the convergence rate for AEPG-SPIDER. We define $X_t \triangleq \sqrt{\sum_{j=tq-q}^{tq-1} \|\mathbf{d}^j\|_2^2}$, and $S_t \triangleq \sum_{j=t}^\infty X_j$. The following assumption is introduced.

**Assumption 4.10.** *There exists a sufficiently large index $t_\star$ such that $\xi \triangleq c_1 \min(\mathbf{v}^{t_\star}) - c_2' - 2\xi' > 0$, where $\xi' \triangleq 5c_3$, $c_1 \triangleq \frac{1}{2}(\frac{1-\theta}{\kappa})^2$, $c_2' \triangleq \frac{(3+\phi)L}{2}$, and $c_3 \triangleq \frac{L}{2\phi}\frac{q}{b}$.*

**Remark 4.11.** *Assume $q = b$ and $\phi = 1$, we have $c_3 = \frac{L}{2}$ and $c_2' = 2L$. Assumption 4.10 is satisfied if $\min(\mathbf{v}^{t_\star}) > \frac{10c_3 + c_2'}{c_1} = \frac{14\kappa^2 L}{(1-\theta)^2}$, requiring $\min(\mathbf{v}^{t_\star})$ to exceed a multiple of $L$, which is relatively mild.*

We now establish the finite-length property of AEPG-SPIDER.

**Theorem 4.12.** *(Proof in Appendix C.4, **Finite-Length Property**). Assume $q \geq 2$. We define $\vartheta$ in Lemma 4.4. We define $\{\xi, \xi'\}$ in Assumption 4.10. We let $\varphi_t \triangleq \varphi(\mathcal{Z}(\mathbb{W}^t) - \mathcal{Z}(\mathbb{W}^\infty))$. We have:*

  **(a)** It holds that $X_{r_t}^2 + \frac{\xi'}{\xi}(X_{r_t}^2 - X_{r_t-1}^2) \leq \frac{2q\vartheta}{\xi}(\varphi^{(r_t-1)q} - \varphi^{r_t q})(X_{r_t} - X_{r_t-1})$.
  **(b)** It holds that $\forall i \geq 1$, $S_i \triangleq \sum_{t=i}^\infty X_t \leq \varpi X_{i-1} + \varpi\varphi_{(i-1)q}$, where $\varpi > 0$ is some constant. The sequence $\{X_t\}_{t=0}^\infty$ has the finite length property that $S_t$ is always upper-bounded by a certain constant, i.e., $\sum_{j=i}^\infty X_j < +\infty$ for all $i$.

Finally, we establish the last-iterate convergence rate for AEPG-SPIDER.

**Theorem 4.13.** *(Proof in Appendix C.5,* **Convergence Rate***).* Assume $q \geq 2$. There exists $t'$ such that for all $t \geq t'$, we have:

(a) If $\tilde{\sigma} = 0$, then the sequence $\mathbf{x}^t$ converges in a finite number of steps in expectation.

(b) If $\tilde{\sigma} \in (0, \frac{1}{2}]$, then there exist $\dot{\tau} \in [0, 1)$ such that $\mathbb{E}[\|\mathbf{x}^{tq} - \mathbf{x}^{\infty}\|] \leq \mathcal{O}(\dot{\tau}^t)$.

(c) If $\tilde{\sigma} \in (\frac{1}{2}, 1)$, then it follows that $\mathbb{E}[\|\mathbf{x}^{tq} - \mathbf{x}^{\infty}\|] \leq \mathcal{O}(t^{-\dot{\tau}})$, where $\dot{\tau} \triangleq \frac{1-\tilde{\sigma}}{2\tilde{\sigma}-1} > 0$.

**Remark 4.14.** *(i) While derived through different analyses, Theorem 4.13 closely mirrors Theorem 4.8, indicating that* AEPG-SPIDER *attains a comparable convergence rate to* AEPG. *(ii) Unlike* AEPG, *which is assessed at every iteration* $\mathbf{x}^t$, *the convergence rate of* AEPG-SPIDER *is evaluated only at specific checkpoints* $\mathbf{x}^{tq}$, *where* $q \geq 2$. *(iii) No existing work examines the last-iterate convergence rate of VR methods, except for the* SVRG-APG *method Li et al. (2017), a double-looped approach. However, its reliance on objective-based line search limits its practicality for stochastic optimization, and its (Q-linear) convergence rate is established only for the specific case where the KL exponent is* $1/2$. *Importantly, their results do not extend to our* AEPG-SPIDER *method. (iv) Theorem 4.13 establishes the first general convergence rate for variance-reduced methods under the KL framework.*

## 5 EXPERIMENTS

This section presents numerical comparisons of AEPG-SPIDER for solving the sparse phase retrieval problem and AEPG for addressing the linear eigenvalue problem, benchmarked against state-of-the-art methods on both real-world and synthetic datasets.

All methods are implemented in MATLAB and tested on an Intel 2.6 GHz CPU with 64 GB of RAM. The experiments are conducted on a set of 8 datasets, including both randomly generated data and publicly available real-world datasets. Details on the data generation process can be found in Appendix Section D. We compare the objective values of all methods after running for $T$ seconds, where $T$ is chosen to be sufficiently large to ensure the convergence of the compared methods. The code is provided in the **supplemental material**.

### 5.1 AEPG-SPIDER ON SPARSE PHASE RETRIEVAL

Sparse phase retrieval seeks to recover a signal $\mathbf{x} \in \mathbb{R}^n$ from magnitude-only measurements $\mathbf{y}_i = |\langle \mathbf{x}, \mathbf{A}_{:i} \rangle|^2$, where $\mathbf{A}_{:i} \in \mathbb{R}^n$ are known measurement vectors and $\mathbf{y}_i \in \mathbb{R}$ are their squared magnitudes. To address this problem, we incorporate sparsity regularization, resulting in the following optimization model: $\min_{\mathbf{x}} h(\mathbf{x}) + f(\mathbf{x})$, where $f(\mathbf{x}) \triangleq \frac{1}{N} \sum_{i=1}^{N} (\langle \mathbf{x}, \mathbf{A}_{:i} \rangle^2 - \mathbf{y}_i)^2$, $h(\mathbf{x}) \triangleq \iota_{\Omega}(\mathbf{x}) + \dot{\lambda} \| \max(|\mathbf{x}|, \tau) \|_1$, $\Omega \triangleq \{\mathbf{x} \mid \|\mathbf{x}\|_{\infty} \leq \dot{r}\}$, $\dot{r}, \dot{\lambda} > 0$. The regularization term $h(\mathbf{x})$ enforces sparsity using the capped-$\ell_1$ penalty Zhang (2010b) while incorporating bound constraints. Since $\mathbf{x}$ is bounded, the spectral norm of $\nabla^2 f(\mathbf{x}) = \frac{1}{N} 4\mathbf{A}^{\mathsf{T}} \mathrm{diag}(3(\mathbf{A}\mathbf{x}) \odot (\mathbf{A}\mathbf{x}) - \mathbf{y})\mathbf{A}$ is also bounded, which implies that $f(\mathbf{x})$ $L$-smooth.

▶ **Compared Methods**. We compare AEPG-SPIDER with three state-of-the-art general-purpose algorithms designed to solve Problem (1). (*i*) ProxSARAH Pham et al. (2020), (*ii*) SpiderBoost and its Nesterov's extrapolation version SpiderBoost-M Wang et al. (2019), and (*iii*) SGP-SPIDER a sub-gradient projection method Yang et al. (2020) using the SPIDER estimator.

▶ **Experimental Settings**. We set the parameters for the optimization problem as $(\dot{r}, \dot{\delta}) = (10, 0.1)$ and vary $\dot{\lambda} \in \{0.01, 0.001\}$. For ProxSARAH, and SpiderBoost, and SpiderBoost-M, SGP-SPIDER, we report results using a fixed step size of $0.1$. For AEPG-SPIDER, we use the parameter configuration $(\underline{v}, \alpha, \beta) = (0.05, 0.01, 1)$, and evaluate its performance for different values of $\theta \in \{0, 0.1, 0.5, 0.9\}$.

▶ **Experimental Results**. The experimental results depicted in Figure 1 offer the following insights. (*i*) The proposed method, AEPG-SPIDER, converges more quickly than the other methods. (*i*) AEPG-SPIDER-($\theta$) consistently outperforms AEPG-SPIDER-(0), particularly when $\theta$ is close to, but less than, $1$. This underscores the importance of Nesterov's extrapolation strategy in addressing composite minimization problems.

### 5.2 AEPG ON LINEAR EIGENVALUE PROBLEM

Given a symmetric matrix $\mathbf{C} \in \mathbb{R}^{\dot{d} \times \dot{d}}$ and an arbitrary orthogonal matrix $\mathbf{V} \in \mathbb{R}^{\dot{d} \times \dot{r}}$ with $\dot{r} \leq \dot{d}$, the trace of $\mathbf{V}^{\mathsf{T}}\mathbf{C}\mathbf{V}$ is minimized when the columns of $\mathbf{V}$ forms an orthogonal basis for the eigenspace corresponding to the $\dot{r}$ smallest eigenvalues of $\mathbf{C}$. Let $\boldsymbol{\lambda}_1 \leq \ldots \leq \boldsymbol{\lambda}_{\dot{d}} < 0$ be the eigenvalues of $\mathbf{C}$.

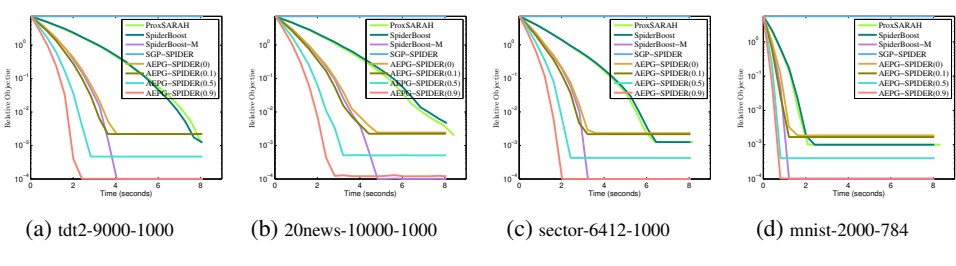

Figure 1: The convergence curve for sparse phase retrieval with $\dot{\lambda} = 0.01$.

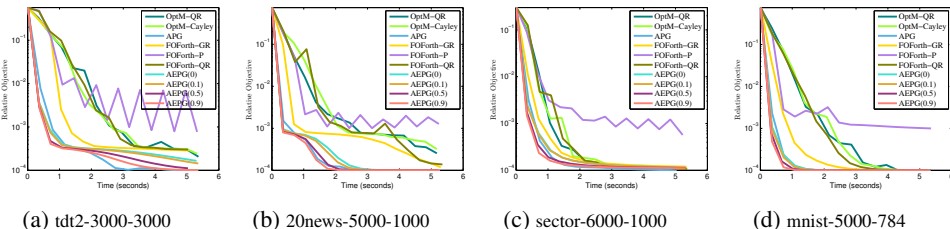

Figure 2: The convergence curve for linear eigenvalue problems with $\dot{r} = 20$.

The problem of finding the $\dot{r}$ smallest eigenvalues can be formulated as: $\min_{\mathbf{V} \in \mathbb{R}^{d \times \dot{r}}} \operatorname{tr}(\mathbf{V}^{\mathsf{T}}\mathbf{C}\mathbf{V}) + |\operatorname{tr}(\mathbf{C})|$, $s.t.\ \mathbf{V}^{\mathsf{T}}\mathbf{V} = \mathbf{I}_{\dot{r}}$.

▶ **Compared Methods**. We compare AEPG with three state-of-the-art methods: APG Li & Lin (2015), FOForth Gao et al. (2018), and OptM Wen & Yin (2013). For FOForth, different retraction strategies are employed to handle the orthogonality constraint, resulting in several variants: FOForth-GR, FOForth-P, and FOForth-QR. Similarly, for OptM, both QR and Cayley retraction strategies are utilized, giving rise to two variants: OptM-QR and OptM-Cayley. It is worth noting that both FOForth and OptM incorporate the Barzilai-Borwein non-monotonic line search in their implementations.

▶ **Experimental Settings**. For both OptM and FOForth, we utilize the implementations provided by their respective authors, using the default solver settings. For AEPG, we configure the parameters as $(\underline{v}, \alpha, \beta) = (0.001, 0.001, 0)$. The performance of all methods is evaluated with varying $\dot{r} \in \{20, 50\}$.

▶ **Experimental Results**. Figure 2 shows the comparisons of objective values for different methods with varying $\dot{r} \in \{20, 50\}$. Several conclusions can be drawn. **(i)** The methods OptM, FOForth, and APG generally deliver comparable performance, with none consistently achieving better results than the others. **(i)** The proposed AEPG method typically demonstrates superior performance compared to all other methods. **(iii)** AEPG-$(\theta)$ consistently achieves better results than AEPG-$(0)$, particularly when $\theta$ is close to, but less than, 1.

## 6 CONCLUSIONS

This paper introduces AEPG-SPIDER, an Adaptive Extrapolated Proximal Gradient method that leverages variance reduction to address the composite nonconvex finite-sum minimization problem. AEPG-SPIDER combines adaptive stepsizes, Nesterov's extrapolation, and the SPIDER estimator to achieve enhanced performance. In the full-batch, non-stochastic setting, it reduces to AEPG. We show that AEPG attains an optimal iteration complexity of $\mathcal{O}(N/\epsilon^2)$, while AEPG-SPIDER achieves $\mathcal{O}(N + \sqrt{N}/\epsilon^2)$ for finding $\epsilon$-approximate stationary points, making them the first Lipschitz-free methods to achieve optimal iteration complexity for this class of composite minimization problems. Under the Kurdyka-Lojasiewicz (KL) assumption, we establish non-ergodic convergence rates for both methods. Preliminary experiments on sparse phase retrieval and linear eigenvalue problems demonstrate the superior performance of AEPG-SPIDER and AEPG over existing methods.

## 7 LLM USAGE

A large language model (LLM) was employed to assist in refining the writing of this paper.

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

# Appendix

The organization of the appendix is as follows:

Appendix A provides notations, technical preliminaries, and relevant lemmas.

Appendix B offers proofs related to Section 3.

Appendix C contains proofs related to Section 4.

Appendix D includes additional experiments details and results.

## A  NOTATIONS, TECHNICAL PRELIMINARIES, AND RELEVANT LEMMAS

### A.1  NOTATIONS

In this paper, bold lowercase letters represent vectors, and uppercase letters denote real-valued matrices. The following notations are used throughout this paper.

- $[n]$: The set $\{1, 2, ..., n\}$.
- $\|\mathbf{x}\|$: Euclidean norm, defined as $\|\mathbf{x}\| = \|\mathbf{x}\|_2 = \sqrt{\langle \mathbf{x}, \mathbf{x} \rangle}$.
- $\langle \mathbf{a}, \mathbf{b} \rangle$ : Euclidean inner product, given by $\langle \mathbf{a}, \mathbf{b} \rangle = \sum_i \mathbf{a}_i \mathbf{b}_i$.
- $\langle \mathbf{a}, \mathbf{b} \rangle_{\mathbf{v}}$ : Generalized inner product, defined as $\langle \mathbf{a}, \mathbf{b} \rangle_{\mathbf{v}} = \sum_i \mathbf{a}_i \mathbf{b}_i \mathbf{v}_i$.
- $\|\mathbf{x}\|_{\mathbf{v}}$: Generalized vector norm, defined as $\|\mathbf{x}\|_{\mathbf{v}} = \sqrt{\sum_{i=1} \mathbf{x}_i^2 \mathbf{v}_i}$, where $\mathbf{v} \geq \mathbf{0}$.
- $\mathbf{a} \leq \alpha$: For $\mathbf{a} \in \mathbb{R}^n$ and $\alpha \in \mathbb{R}$, this means $\mathbf{a}_i \leq \alpha$ for all $i \in n$.
- $\iota_\Omega(\mathbf{x})$ : Indicator function of a set $\Omega$ with $\iota_\Omega(\mathbf{x}) = 0$ if $\mathbf{x} \in \Omega$ and otherwise $+\infty$.
- $\mathbb{E}[v]$: Expected value of the random variable $v$.
- $\{A_i\}_{i=0}^\infty$, $\{B_i\}_{i=0}^\infty$: sequences indexed by the integers $i = 0, 1, 2, 3, \ldots$.
- $\text{dist}(\Omega, \Omega')$ : distance between two sets with $\text{dist}(\Omega, \Omega') \triangleq \inf_{\mathbf{w} \in \Omega, \mathbf{w}' \in \Omega'} \|\mathbf{w} - \mathbf{w}'\|$.
- $\|\partial h(\mathbf{x})\|$: distance from the origin to $\partial h(\mathbf{x})$ with $\|\partial h(\mathbf{x})\| = \inf_{\mathbf{y} \in \partial h(\mathbf{x})} \|\mathbf{y}\| = \text{dist}(\mathbf{0}, \partial h(\mathbf{x}))$.
- $\mathbf{A}^\top$ : the transpose of the matrix $\mathbf{A}$.
- $b$: The mini-batch size parameter of AEPG-SPIDER.
- $q$: The frequency parameter of AEPG-SPIDER (that determines when the full gradient is computed).

### A.2  TECHNICAL PRELIMINARIES

We introduce key concepts from nonsmooth analysis, focusing on the Fréchet subdifferential and the limiting (Fréchet) subdifferential Mordukhovich (2006); Rockafellar & Wets. (2009); Bertsekas (2015). Let $F : \mathbb{R}^n \to (-\infty, +\infty]$ be an extended real-valued, not necessarily convex function. The domain of $F(\cdot)$ is defined as $\text{dom}(F) \triangleq \{\mathbf{x} \in \mathbb{R}^n : |F(\mathbf{x})| < +\infty\}$. The Fréchet subdifferential of $F$ at $\mathbf{x} \in \text{dom}(F)$, denoted as $\hat{\partial} F(\mathbf{x})$, is given by

$$\hat{\partial} F(\mathbf{x}) \triangleq \{\mathbf{v} \in \mathbb{R}^n : \lim_{\substack{\mathbf{z} \to \mathbf{x} \\ \mathbf{z} \neq \mathbf{x}}} \inf \frac{F(\mathbf{z}) - F(\mathbf{x}) - \langle \mathbf{v}, \mathbf{z} - \mathbf{x} \rangle}{\|\mathbf{z} - \mathbf{x}\|} \geq 0\}.$$

The limiting subdifferential of $F(\cdot)$ at $\mathbf{x} \in \text{dom}(F)$, denoted $\partial F(\mathbf{x})$, is defined as:

$$\partial F(\mathbf{x}) \triangleq \{\mathbf{v} \in \mathbb{R}^n : \exists \mathbf{x}^k \to \mathbf{x}, F(\mathbf{x}^k) \to F(\mathbf{x}), \mathbf{v}^k \in \hat{\partial} F(\mathbf{x}^k) \to \mathbf{v}, \forall k\}.$$

It is important to note that $\hat{\partial} F(\mathbf{x}) \subseteq \partial F(\mathbf{x})$. If $F(\cdot)$ is differentiable at $\mathbf{x}$, then $\hat{\partial} F(\mathbf{x}) = \partial F(\mathbf{x}) = \{\nabla F(\mathbf{x})\}$, where $\nabla F(\mathbf{x})$ represents the gradient of $F(\cdot)$ at $\mathbf{x}$. For convex function $F(\cdot)$, both $\hat{\partial} F(\mathbf{x})$ and $\partial F(\mathbf{x})$ reduce to the classical subdifferential for convex functions: $\hat{\partial} F(\mathbf{x}) = \partial F(\mathbf{x}) = \{\mathbf{v} \in \mathbb{R}^n : F(\mathbf{z}) - F(\mathbf{x}) - \langle \mathbf{v}, \mathbf{z} - \mathbf{x} \rangle \geq 0, \forall \mathbf{z} \in \mathbb{R}^n\}$.

## A.3 RELEVANT LEMMAS

We provide a set of useful lemmas, each independent of context and specific methodologies.

**Lemma A.1.** *(Pythagoras Relation) For any vectors* $\mathbf{a}, \mathbf{v}, \mathbf{x}, \mathbf{x}^+ \in \mathbb{R}^n$ *with* $\mathbf{v} \geq \mathbf{0}$, *we have:*

$$\tfrac{1}{2}\|\mathbf{x} - \mathbf{a}\|_{\mathbf{v}}^2 - \tfrac{1}{2}\|\mathbf{x}^+ - \mathbf{a}\|_{\mathbf{v}}^2 = \tfrac{1}{2}\|\mathbf{x} - \mathbf{x}^+\|_{\mathbf{v}}^2 + \langle \mathbf{a} - \mathbf{x}^+, \mathbf{x}^+ - \mathbf{x}\rangle_{\mathbf{v}}.$$

**Lemma A.2.** *For all* $a, b \geq 0$ *and* $c, d > 0$, *we have:* $\frac{a+b}{c+d} \leq \max(\frac{a}{c}, \frac{b}{d})$.

*Proof.* We consider two cases: (*i*) $\frac{a}{c} \geq \frac{b}{d}$. We derive: $b \leq \frac{ad}{c}$, leading to $\frac{a+b}{c+d} \leq \frac{a+\frac{ad}{c}}{c+d} = \frac{a}{c} \cdot \frac{c+d}{c+d} = \frac{a}{c}$. (*ii*) $\frac{a}{c} < \frac{b}{d}$. We have: $a \leq \frac{bc}{d}$, resulting in $\frac{a+b}{c+d} \leq \frac{\frac{bc}{d}+b}{c+d} = \frac{b}{d} \cdot \frac{c+d}{c+d} = \frac{b}{d}$. □

**Lemma A.3.** *Assume* $ax^2 \leq bx + c$, *where* $b, c, x \geq 0$ *and* $a > 0$. *Then, we have:* $x \leq \sqrt{c/a} + b/a$.

*Proof.* Given the quadratic equality $ax^2 \leq bx + c$, we have $\frac{b-\sqrt{b^2+4ac}}{2a} \leq x \leq \frac{b+\sqrt{b^2+4ac}}{2a}$. Since $x \geq 0$, we have $0 \leq x \leq \frac{b+\sqrt{b^2+4ac}}{2a} \leq \frac{b+b+2\sqrt{ac}}{2a} = b/a + \sqrt{c/a}$, where the last inequality uses $\sqrt{a+d} \leq \sqrt{a} + \sqrt{d}$ for all $a, d \geq 0$. □

**Lemma A.4.** *Assume that* $\{A_i\}_{i=0}^n$ *and* $\{B_i\}_{i=0}^{n+1}$ *are two non-negative sequences with* $A_0 \leq A_1 \leq \ldots \leq A_n$. *Then, we have:*

$$\textstyle\sum_{t=0}^n A_t(B_t - B_{t+1}) \leq [\max_{i=0}^n A_i] \cdot [\max_{j=0}^n B_j].$$

*Proof.* We have:

$$
\begin{aligned}
\textstyle\sum_{t=0}^n A_t(B_t - B_{t+1}) &= [\textstyle\sum_{t=1}^n (A_t - A_{t-1})B_t] + A_0 B_0 - A_n B_{n+1} \\
&\overset{①}{\leq} [\textstyle\sum_{t=1}^n (A_t - A_{t-1})B_t] + A_0 B_0 \\
&\overset{②}{\leq} [\textstyle\sum_{t=1}^n (A_t - A_{t-1})] \cdot [\max_{j=0}^n B_j] + A_0[\max_{j=0}^n B_j] \\
&= A_n[\max_{j=0}^n B_j],
\end{aligned}
$$

where step ① uses $A_n, B_{n+1} \geq 0$; step ② uses $\{A_t\}_{t=0}^n$ is non-decreasing. □

**Lemma A.5.** *Let* $\{A_t\}_{t=0}^\infty$ *be a sequence of nonnegative real numbers, and let* $c > 0$. *Then, for every* $t \geq 0$,

$$\sum_{i=0}^t \frac{A_i}{\sqrt{c + \sum_{j=0}^i A_j}} \leq 2\sqrt{c + \sum_{i=0}^t A_i} \tag{3}$$

*Proof.* This lemma extends the result of Lemma 5 in McMahan & Streeter (2010).

We define $S_t \triangleq c + \sum_{i=0}^t A_i$.

Initially, we define $h(x) \triangleq \frac{x}{\sqrt{y}} + 2\sqrt{y - x} - 2\sqrt{y}$, where $y > x \geq 0$. We have $h'(x) = y^{-1/2} - (y - x)^{-1/2} \leq 0$. Therefore, $h(x)$ is non-increasing for all $x \geq 0$. Given $h(0) = 0$, it holds that

$$h(x) \triangleq \frac{x}{\sqrt{y}} + 2\sqrt{y - x} - 2\sqrt{y} \leq 0. \tag{4}$$

We complete the proof of the lemma using mathematical induction.

**Part (a).** The lemma holds $t = 0$, since $\frac{A_0}{\sqrt{c + A_0}} \leq 2\sqrt{c + A_0}$.

**Part (b)**. Now, fix some $t$ and assume that the lemma holds for $t-1$. We proceed as follows:

$$
\begin{aligned}
\sum_{i=0}^{t} A_i/\sqrt{S_i} &= A_t/\sqrt{S_t} + \sum_{i=0}^{t-1} A_i/\sqrt{S_i} \\
&\overset{\text{①}}{\le} A_t/\sqrt{S_t} + 2\sqrt{S_{t-1}} \\
&\overset{\text{②}}{=} A_t/\sqrt{S_t} + 2\sqrt{S_t - A_t} \\
&\overset{\text{③}}{\le} 2\sqrt{S_t},
\end{aligned}
$$

where step ① uses the inductive hypothesis that the conclusion of this lemma holds for $t-1$; step ② uses $S_t \triangleq c + \sum_{i=0}^{t} A_i$; step ③ uses Inequality (4) with $x = A_t \ge 0$ and $y = S_t > 0$.

$\square$

**Lemma A.6.** *Let $\{A_t\}_{t=0}^{\infty}$ be a sequence of nonnegative real numbers, and let $c > 0$ and $p \in (0, 1]$. Then, for every $t \ge 0$,*

$$
\sum_{i=0}^{t} \frac{A_i}{c + \sum_{j=0}^{i} A_j} \le -1 + \frac{1}{p}\left(1 + \frac{1}{c}\sum_{i=0}^{t} A_i\right)^p.
$$

*Proof.* We define $S_t \triangleq \sum_{i=0}^{t} A_i$.

We define $g(x) = \frac{x}{1+x} - \frac{1}{p}(1+x)^p + 1$ and $f(x) = x^p$, where $x \ge 0$ and $p \in (0, 1]$.

First, we have $g(0) = 0$ and $g'(x) = (1+x)^{-2} - (1+x)^{p-1} = \frac{1 - (1+x)^{p+1}}{(1+x)^2} < 0$. We derive:

$$
g(x) \le 0. \tag{5}
$$

Second, since $f(x)$ is concave, we have $\forall x, y > 0$, $f(x) \le f(y) + \langle x - y, f'(y)\rangle$, leading to

$$
\forall x, y > 0,\ x^p \le y^p - p(y - x) \cdot y^{p-1}. \tag{6}
$$

We finish the proof of the lemma using mathematical induction.

**Part (a)**. We first consider $t = 0$. We have

$$
\begin{aligned}
&A_0/(c + S_0) - \frac{1}{p}(1 + S_0/c)^p + 1 \\
&\overset{\text{①}}{=} A_0/(c + A_0) - \frac{1}{p}(1 + A_0/c)^p + 1 \\
&= \frac{A_0/c}{1 + A_0/c} - \frac{1}{p}(1 + A_0/c)^p + 1 \\
&\overset{\text{②}}{\le} 0,
\end{aligned}
$$

where step ① uses $A_0 = S_0$; step ② uses Inequality (5) with $x = A_0/c$. We conclude that the conclusion of this lemma holds for $t = 0$.

**Part (b)**. Now, fix some $t$ and assume that the lemma holds for $t-1$. We derive:

$$
\begin{aligned}
\sum_{i=0}^{t} A_i/(c + S_i) &= A_t/(c + S^t) + \sum_{i=0}^{t-1} A_i/(c + S_i) \\
&\overset{\text{①}}{\le} -1 + A_t/(c + S_t) + \frac{1}{p}c^{-p}(c + S_{t-1})^p \\
&\overset{\text{②}}{=} -1 + A_t/(c + S_t) + \frac{1}{p}c^{-p}(c + S_t - A_t)^p \\
&\overset{\text{③}}{\le} -1 + A_t/(c + S_t) + \frac{1}{p}c^{-p}\{(c + S_t)^p - pA_t(c + S_t)^{p-1}\} \\
&= -1 + \frac{1}{p}c^{-p}(c + S_t)^p + A_t(c + S_t)^{p-1} \cdot \{(c + S_t)^{-p} - c^{-p}\} \\
&\overset{\text{④}}{\le} -1 + \frac{1}{p}c^{-p}(c + S_t)^p + 0,
\end{aligned}
$$

where step ① uses the inductive hypothesis that the conclusion of this lemma holds for $t-1$; step ② uses $S_t \triangleq \sum_{i=0}^{t} A_i$; step ③ uses Inequality (6) with $x = c + A_t - A_t > 0$ and $y = c + S_t > 0$; step ④ uses $(c + S_t)^{-p} \le c^{-p}$ for all $p \in (0, 1]$.

$\square$

**Lemma A.7.** *Let $\{Z_t\}_{t=0}^{\infty}$ be a non-negative sequence satisfying*

$$Z_{t+1} \le a + b\sqrt{\max_{i=0}^{t} Z_i}$$

*for all $t \ge 0$, where $a, b \ge 0$. It follows that:*

$$Z_t \le \overline{Z} \triangleq \max\left(Z_0, \left(\tfrac{1}{2}b + \tfrac{1}{2}\sqrt{b^2 + 4a}\right)^2\right) \tag{7}$$

*for all $t \ge 0$. Furthermore, an alternative valid upper bound for $Z_t$ is given by $\overline{Z}_+ \triangleq \max(Z_0, 2b^2 + 2a)$.*

*Proof.* We define $M_t \triangleq \max_{0 \le i \le t} Z_i$ for all $t \ge 0$.

**Part (a).** For all $t \ge 0$, we have:

$$M_{t+1} \overset{\text{①}}{=} \max(M_t, Z_{t+1}) \overset{\text{②}}{\le} \max(M_t, a + b\sqrt{M_t}),$$

where step ① uses the definition of $M_t$; step ② uses We have $Z_{t+1} \le a + b\sqrt{M_t}$, which is the assumption of this lemma.

**Part (b).** We establish a fixed-point upper bound that the sequence $M_t$ cannot exceed such that $M_t \le \overline{M}$. For $\overline{M}$ to be a valid upper bound, it should satisfy the recurrence relation: $\overline{M} = a + b\sqrt{\overline{M}}$. This is because if $M_t \le \overline{M}$, then we have: $M_{t+1} \le a + b\sqrt{M_t} \le a + b\sqrt{\overline{M}} \le \overline{M}$, which would imply by induction that $M_t \le \overline{M}$ for all $t \ge 0$. Solving the quadratic equation $\overline{M} = a + b\sqrt{\overline{M}}$ yields a positive root $(\tfrac{1}{2}(b + \sqrt{b^2 + 4a}))^2 \triangleq c$. Taking into account the case $M_0$, for all $t \ge 0$, we have $M_t \le \max(M_0, c) = \max(Z_0, c) \triangleq \overline{Z}$.

**Part (c).** We verify that $Z_t \le \overline{Z}$ for all $t \ge 0$ using mathematical induction. (i) The conclusion holds for $t = 0$. (ii) Assume that $Z_t \le \overline{Z}$ holds for some $t$. We now show that it also holds for $t + 1$. We have:

$$Z_{t+1} \overset{\text{①}}{\le} a + b\sqrt{M_t} \overset{\text{②}}{\le} a + b\sqrt{c} \overset{\text{③}}{=} c,$$

where step ① uses the assumption of this lemma that $Z_{t+1} \le a + b\sqrt{\max_{i=0}^{t} Z_i}$; step ② uses $M_t \le c$; step ③ uses the fact that $x = c$ is the positive root for the equation $a + b\sqrt{x} = x$. Therefore, we conclude that $Z_t \le \max(Z_0, c) \triangleq \overline{Z}$ for all $t \ge 0$.

**Part (d).** Finally, we have: $c \triangleq \left(\tfrac{1}{2}b + \tfrac{1}{2}\sqrt{b^2 + 4a}\right)^2 \le (\tfrac{1}{2}(b + b + 2\sqrt{a})^2 = (b + \sqrt{a})^2 \le 2b^2 + 2a$. Hence, $\overline{Z}_+ \triangleq \max(Z_0, 2b^2 + 2a)$ is also a valid upper bound for $Z_t$. $\qquad\square$

**Lemma A.8.** *Assume that $(X_{t+1})^2 \le (X_t + X_{t-1})(P_t - P_{t+1})$ and $P_t \ge P_{t+1}$, where $\{X_t, P_t\}_{t=0}^{\infty}$ are two nonnegative sequences. Then, for all $i \ge 0$, we have: $\sum_{j=i}^{\infty} X_{j+1} \le X_i + X_{i-1} + 4P_i$.*

*Proof.* We define $W_t \triangleq P_t - P_{t+1}$, where $t \ge 0$.

First, for any $i \ge 0$, we have:

$$\sum_{t=i}^{T} W_t = \sum_{t=i}^{T} (P_t - P_{t+1}) = P_i - P_{T+1} \overset{\text{①}}{\le} P_i, \tag{8}$$

where step ① uses $P_i \ge 0$ for all $i$.

Second, we obtain:

$$\begin{aligned}
X_{t+1} &\overset{\text{①}}{\le} \sqrt{(X_t + X_{t-1})W_t} \\
&\overset{\text{②}}{\le} \sqrt{\tfrac{\theta}{4}(X_t + X_{t-1})^2 + (W_t)^2/\theta}, \forall \theta > 0 \\
&\overset{\text{③}}{\le} \tfrac{\sqrt{\theta}}{2}(X_t + X_{t-1}) + \sqrt{1/\theta} \cdot W_t, \forall \theta > 0.
\end{aligned} \tag{9}$$

Here, step ① uses $(X_{t+1})^2 \le (X_t + X_{t-1})(P_t - P_{t+1})$ and $W_t \triangleq P_t - P_{t+1}$; step ② uses the fact that $ab \le \tfrac{\theta}{4}a^2 + \tfrac{1}{\theta}b^2$ for all $\alpha > 0$; step ③ uses the fact that $\sqrt{a + b} \le \sqrt{a} + \sqrt{b}$ for all $a, b \ge 0$.

Assume $\theta < 1$. Telescoping Inequality (9) over $t$ from $i$ to $T$, we obtain:

$$\sqrt{1/\theta} \sum_{t=i}^{T} W_t$$

$$\geq \left( \sum_{t=i}^{T} X_{t+1} \right) - \frac{\sqrt{\theta}}{2} \left( \sum_{t=i}^{T} X_t \right) - \frac{\sqrt{\theta}}{2} \left( \sum_{t=i}^{T} X_{t-1} \right)$$

$$= \left( X_{T+1} + X_T + \sum_{t=i}^{T-2} X_{t+1} \right) - \frac{\sqrt{\theta}}{2} \left( X_i + X_T + \sum_{t=i}^{T-2} X_{t+1} \right) - \frac{\sqrt{\theta}}{2} \left( X_{i-1} + X_i + \sum_{t=i}^{T-2} X_{t+1} \right)$$

$$= X_{T+1} + X_T - \frac{\sqrt{\theta}}{2}(X_i + X_T + X_{i-1} + X_i) + (1 - \sqrt{\theta}) \sum_{t=i}^{T-2} X_{t+1}$$

$$\overset{①}{\geq} 0 + X_T(1 - \frac{\sqrt{\theta}}{2}) - \frac{\sqrt{\theta}}{2}(X_i + X_{i-1} + X_i) + (1 - \sqrt{\theta}) \sum_{t=i}^{T-2} X_{t+1}$$

$$\overset{②}{\geq} - \sqrt{\theta}(X_i + X_{i-1}) + (1 - \sqrt{\theta}) \sum_{t=i}^{T-2} X_{t+1},$$

where step ① uses $X_{T+1} \geq 0$; step ② uses $1 - \frac{\sqrt{\theta}}{2} > 0$. This leads to:

$$\sum_{t=i}^{T-2} X_{t+1} \leq (1 - \sqrt{\theta})^{-1} \cdot \{ \sqrt{\theta}(X_i + X_{i-1}) + \sqrt{\frac{1}{\theta}} \sum_{t=i}^{T} W_t \}$$

$$\overset{①}{=} (X_i + X_{i-1}) + 4 \sum_{t=i}^{T} W_t$$

$$\overset{②}{=} (X_i + X_{i-1}) + 4 P_i,$$

step ① uses the fact that $(1 - \sqrt{\theta})^{-1} \cdot \sqrt{\theta} = 1$ and $(1 - \sqrt{\theta})^{-1} \cdot \sqrt{1/\theta} = 4$ when $\theta = 1/4$; step ② uses Inequality (8). Letting $T \to \infty$, we conclude this lemma.

$\square$

**Lemma A.9.** *Assume that $X_j^2 + \gamma(X_j^2 - X_{j-1}^2) \leq (P_{(j-1)q} - P_{jq})(X_j - X_{j-1})$ for all $j \geq 1$, where $\gamma > 0$ is a constant, $q \geq 1$ is an integer, and $\{X_j, P_j\}_{j=0}^{\infty}$ are two nonnegative sequences with $P_{(j-1)q} \geq P_{jq}$. Then, for all $i \geq 1$, we have: $\sum_{j=i}^{\infty} X_j \leq \gamma' X_{i-1} + \gamma' P_{(i-1)q}$, where $\gamma' \triangleq 16(\gamma + 1)$.*

*Proof.* We define $\overline{P} \triangleq P_{(i-1)q}$, and $\gamma' \triangleq 16(\gamma + 1)$.

Using the recursive formulation, we derive the following results:

$$X_j \leq \sqrt{\frac{\gamma}{1+\gamma} X_{j-1}^2 + \frac{1}{1+\gamma} \cdot (P_{(j-1)q} - P_{jq})(X_j + X_{j-1})}$$

$$\overset{①}{\leq} \sqrt{\frac{\gamma}{1+\gamma}} X_{j-1} + \sqrt{\frac{1}{1+\gamma}} \cdot \sqrt{(X_j + X_{j-1}) \cdot (P_{(j-1)q} - P_{jq})}$$

$$\overset{②}{\leq} \sqrt{\frac{\gamma}{1+\gamma}} X_{j-1} + \sqrt{\frac{1}{1+\gamma}} \sqrt{\tau(X_j + X_{j-1})^2 + \frac{1}{4\tau}(P_{(j-1)q} - P_{jq})^2}$$

$$\overset{③}{\leq} \sqrt{\frac{\gamma}{1+\gamma}} X_{j-1} + \sqrt{\frac{\tau}{1+\gamma}}(X_j + X_{j-1}) + \sqrt{\frac{1}{4\tau(1+\gamma)}} \cdot \left( P_{(j-1)q} - P_{jq} \right),$$

where steps ① and ③ uses $\sqrt{a + b} \leq \sqrt{a} + \sqrt{b}$ for all $a, b \geq 0$; step ② uses $ab \leq \tau a^2 + \frac{1}{4\tau} b^2$ for all $a, b \in \mathbb{R}$, and $\tau > 0$. This further leads to:

$$\sqrt{1+\gamma} X_j \leq \sqrt{\gamma} X_{j-1} + \sqrt{\tau}(X_j + X_{j-1}) + \sqrt{\frac{1}{4\tau}} \cdot \left( P_{(j-1)q} - P_{jq} \right).$$

Summing the inequality above over $j$ from $i$ to $T$ yields:

$$0 \leq (-\sqrt{1+\gamma} + \sqrt{\tau}) \sum_{j=i}^{T} X_j + (\sqrt{\gamma} + \sqrt{\tau}) \sum_{j=i}^{T} X_{j-1} + \sqrt{\frac{1}{4\tau}} \cdot (P_{(i-1)q} - P_{Tq})$$

$$\overset{①}{\leq} (-\sqrt{1+\gamma} + \sqrt{\tau}) \sum_{j=i}^{T} X_j + (\sqrt{\gamma} + \sqrt{\tau}) \sum_{j=i-1}^{T-1} X_j + \sqrt{\frac{1}{4\tau}} \cdot \overline{P}$$

$$= (-\sqrt{\gamma+1} + \sqrt{\gamma} + 2\sqrt{\tau}) \sum_{j=i}^{T-1} X_j + (\sqrt{\gamma} + \sqrt{\tau})X_{i-1} + (\sqrt{\tau} - \sqrt{1+\gamma})X_T + \sqrt{\frac{1}{4\tau}} \cdot \overline{P}$$

$$\overset{②}{=} (-\frac{1}{2\sqrt{\gamma+1}} + 2\sqrt{\tau}) \sum_{j=i}^{T-1} X_j + (\sqrt{\gamma} + \sqrt{\tau})X_{i-1} + (\sqrt{\tau} - \sqrt{1+\gamma})X_T + \sqrt{\frac{1}{4\tau}} \cdot \overline{P}$$

$$\overset{③}{\leq} - \frac{1}{4\sqrt{\gamma+1}} \sum_{j=i}^{T-1} X_j + (\sqrt{\gamma} + \frac{1}{8\sqrt{\gamma+1}})X_{i-1} + 4\sqrt{1+\gamma} \cdot \overline{P},$$

where step ① uses the definition of $\overline{P}$; step ② uses the fact that $\sqrt{\gamma+1} - \sqrt{\gamma} \geq \frac{1}{2\sqrt{\gamma+1}}$ for all $\gamma > 0$; step ③ uses the choice that $\tau = \frac{1}{64(\gamma+1)}$, which leads to $\sqrt{\tau} - \sqrt{1+\gamma} \leq 0$.

Finally, we obtain:

$$
\begin{aligned}
\sum_{j=i}^{T-1} X_j \quad &\leq \quad 4\sqrt{\gamma+1} \cdot \left( (\sqrt{\gamma} + \tfrac{1}{8\sqrt{\gamma+1}}) X_{i-1} + 4\sqrt{1+\gamma} \cdot \overline{P} \right) \\
&\leq \quad (4(\gamma+1) + \tfrac{1}{2}) X_{i-1} + 16(\gamma+1)\overline{P} \\
&\leq \quad \gamma' X_{i-1} + \gamma' \overline{P}.
\end{aligned}
$$

$\square$

**Lemma A.10.** *Assume that*

$$
S_t \leq c(S_{t-1} - S_t)^u,
$$

*where $c > 0$, $u \in (0,1)$, and $\{S_t\}_{t=0}^{\infty}$ is a nonnegative sequence. Then we have*

$$
S_T \leq \mathcal{O}\left( \tfrac{1}{T^{\varsigma}} \right),
$$

*where $\varsigma = \frac{u}{1-u}$.*

*Proof.* We define $\tau \triangleq \frac{1}{u} - 1 > 0$, and $g(s) = s^{-\tau-1}$.

Using the inequality $S_t \leq c(S_{t-1} - S_t)^u$, we obtain:

$$
c^{1/u}(S_{t-1} - S_t) \geq (S_t)^{1/u} \overset{①}{=} (S_t)^{\tau+1} \overset{②}{=} \frac{1}{g(S_t)}, \tag{10}
$$

where step ① uses $1/u = \tau + 1$; step ② uses the definition of $g(\cdot)$.

We let $\kappa > 1$ be any constant, and examine two cases for $g(S_t)/g(S_{t-1})$.

**Case (1).** $g(S_t)/g(S_{t-1}) \leq \kappa$. We define $f(s) \triangleq -\frac{1}{\tau} \cdot s^{-\tau}$. We derive:

$$
\begin{aligned}
1 \quad &\overset{①}{\leq} \quad c^{1/u} \cdot (S_{t-1} - S_t) \cdot g(S_t) \\
&\overset{②}{\leq} \quad c^{1/u} \cdot (S_{t-1} - S_t) \cdot \kappa g(S_{t-1}) \\
&\overset{③}{\leq} \quad c^{1/u} \cdot \kappa \int_{S_t}^{S_{t-1}} g(s)ds \\
&\overset{④}{=} \quad c^{1/u} \cdot \kappa \cdot (f(S_{t-1}) - f(S_t)) \\
&\overset{⑤}{=} \quad c^{1/u} \cdot \kappa \cdot \frac{1}{\tau} \cdot \left( [S_t]^{-\tau} - [S_{t-1}]^{-\tau} \right),
\end{aligned}
$$

where step ① uses Inequality (10); step ② uses $g(S_t) \leq \kappa g(S_{t-1})$; step ③ uses the fact that $g(s)$ is a nonnegative and increasing function that $(a-b)g(a) \leq \int_b^a g(s)ds$ for all $a, b \in [0, \infty)$; step ④ uses the fact that $\nabla f(s) = g(s)$; step ⑤ uses the definition of $f(\cdot)$. This leads to:

$$
[S_t]^{-\tau} - [S_{t-1}]^{-\tau} \geq \frac{\tau}{\kappa c^{1/u}}. \tag{11}
$$

**Case (2).** $g(S_t)/g(S_{t-1}) > \kappa$. We have:

$$
\begin{aligned}
g(S_t) > \kappa g(S_{t-1}) \quad &\overset{①}{\Rightarrow} \quad [S_t]^{-(\tau+1)} > \kappa \cdot [S_{t-1}]^{-(\tau+1)} \\
&\overset{②}{\Rightarrow} \quad ([S_t]^{-(\tau+1)})^{\frac{\tau}{\tau+1}} > \kappa^{\frac{\tau}{\tau+1}} \cdot ([S_{t-1}]^{-(\tau+1)})^{\frac{\tau}{\tau+1}} \\
&\Rightarrow \quad [S_t]^{-\tau} > \kappa^{\frac{\tau}{\tau+1}} \cdot [S_{t-1}]^{-\tau}, \tag{12}
\end{aligned}
$$

where step ① uses the definition of $g(\cdot)$; step ② uses the fact that if $a > b > 0$, then $a^{\dot{\tau}} > b^{\dot{\tau}}$ for any exponent $\dot{\tau} \triangleq \frac{\tau}{\tau+1} \in (0,1)$. For any $t \geq 1$, we derive:

$$
\begin{aligned}
[S_t]^{-\tau} - [S_{t-1}]^{-\tau} \quad &\overset{①}{\geq} \quad (\kappa^{\frac{\tau}{\tau+1}} - 1) \cdot [S_{t-1}]^{-\tau} \\
&\overset{②}{\geq} \quad (\kappa^{\frac{\tau}{\tau+1}} - 1) \cdot [S_0]^{-\tau}, \tag{13}
\end{aligned}
$$

where step ① uses Inequality (12); step ② uses $\tau > 0$ and $S_{t-1} \leq S_0$ for all $t \geq 1$.

In view of Inequalities (11) and (13), we have:

$$[S_t]^{-\tau} - [S_{t-1}]^{-\tau} \geq \underbrace{\min(\tfrac{\tau}{\kappa c^{1/u}}, (\kappa^{\frac{\tau}{\tau+1}} - 1) \cdot [S_0]^{-\tau})}_{\triangleq \ddot{c}}. \tag{14}$$

Telescoping Inequality (14) over $t$ from 1 to $T$, we have:

$$[S_T]^{-\tau} - [S_0]^{-\tau} \geq T\ddot{c}.$$

This leads to:

$$S_T = [S_T^{-\tau}]^{-1/\tau} \leq \mathcal{O}([T]^{-1/\tau}).$$

$\square$

**Lemma A.11.** *Assume that*

$$S_t \leq c(S_{t-2} - S_t)^u,$$

*where $c > 0$, $u \in (0, 1)$, and $\{S_t\}_{t=0}^{\infty}$ is a nonnegative sequence. Then we have*

$$S_T \leq \mathcal{O}\left(\tfrac{1}{T^\varsigma}\right),$$

*where $\varsigma = \frac{u}{1-u} > 0$.*

*Proof.* We analyze two cases under the condition $S_t \leq c(S_{t-2} - S_t)^u$ for all $t \geq 0$.

**Case (1).** $t \in \{0, 2, 4, 6, \ldots\}$. We define the sequence $\{\ddot{S}_t\}_{t=0}^{T}$ as $\ddot{S}_i = S_{2i}$ for $i \geq 0$. It follows that $\ddot{S}_j \leq c(\ddot{S}_{j-1} - \ddot{S}_j)^u$ for all $j \geq 1$. By applying Lemma A.10, we obtain $\ddot{S}_T \leq \mathcal{O}(T^{-\varsigma})$, leading to $S_T = \mathcal{O}(\ddot{S}_{(T/2)}) \leq \mathcal{O}((\tfrac{T}{2})^{-\varsigma}) = \mathcal{O}(T^{-\varsigma})$.

**Case (2).** $t \in \{1, 3, 5, 7, \ldots\}$. We define the sequence $\{\dot{S}_t\}_{t=0}^{T}$ as $\dot{S}_i = S_{2i+1}$ for $i \geq 0$. It follows that $\dot{S}_j \leq c(\dot{S}_{j-1} - \dot{S}_j)^u$ for all $j \geq 1$. By applying Lemma A.10, we have $\dot{S}_T \leq \mathcal{O}(T^{-\varsigma})$, resulting in $S_T = \mathcal{O}(\ddot{S}_{[(T-1)/2]}) \leq \mathcal{O}((\tfrac{T-1}{2})^{-\varsigma}) = \mathcal{O}(T^{-\varsigma})$.

$\square$

## B PROOF OF SECTION 3

We now provide an initial theoretical analysis applicable to both algorithms, followed by a detailed, separate analysis for each.

### B.1 INITIAL THEORETICAL ANALYSIS

We first establish key properties of $\mathbf{v}^t$ and $\sigma^t$ utilized in Algorithm 1.

**Lemma B.1.** *(Properties of $\mathbf{v}^t$)* We define $\mathcal{R}_t \triangleq \sum_{i=0}^{t} \|\mathbf{r}^i\|_2^2 \in \mathbb{R}$. For all $t \geq 0$, we have:

(a) $\sqrt{\underline{v}^2 + \alpha \mathcal{R}_t} \leq \mathbf{v}^{t+1} \leq \sqrt{\underline{v}^2 + (\alpha + \beta)\mathcal{R}_t} \triangleq \mathcal{V}_{t+1}$.

(b) $\frac{\max(\mathbf{v}^t)}{\min(\mathbf{v}^t)} \leq \kappa \triangleq 1 + \sqrt{\beta/\alpha}$.

(c) $\frac{\min(\mathbf{v}^{t+1})}{\min(\mathbf{v}^t)} \leq \dot{\kappa} \triangleq 1 + 2\kappa\overline{x}\sqrt{\alpha + \beta}$.

*Proof.* We define $\mathcal{R}_t \triangleq \sum_{i=0}^{t} \|\mathbf{r}^i\|_2^2 \in \mathbb{R}$, where $\mathbf{r}^t \triangleq \mathbf{v}^t \odot \mathbf{d}^t$, and $\mathbf{d}^t \triangleq \mathbf{x}^{t+1} - \mathbf{x}^t$.

We define $\mathbf{s}^t \triangleq \alpha\|\mathbf{r}^t\|_2^2 + \beta\mathbf{r}^t \odot \mathbf{r}^t \in \mathbb{R}^n$.

**Part (a).** Given $\mathbf{v}^{t+1} = \sqrt{\mathbf{v}^0 \odot \mathbf{v}^0 + \sum_{i=0}^{t} \mathbf{s}^i}$, We derive:

$$\mathbf{v}^{t+1} \odot \mathbf{v}^{t+1} = (\mathbf{v}^0) \odot (\mathbf{v}^0) + \alpha \underbrace{\sum_{i=0}^{t} \|\mathbf{r}^i\|_2^2}_{= \mathcal{R}_t} \cdot \mathbf{1} + \beta \underbrace{\sum_{i=0}^{t} \mathbf{r}^i \odot \mathbf{r}^i}_{\leq \mathcal{R}_t \cdot \mathbf{1}}.$$

This results in the following lower and upper bounds for $\mathbf{v}^{t+1}$ for all $t \geq 0$:

$$\sqrt{\underline{\mathrm{v}}^2 + \alpha \mathcal{R}_t} \leq \mathbf{v}^{t+1} \leq \sqrt{\underline{\mathrm{v}}^2 + (\alpha + \beta)\mathcal{R}_t}.$$

**Part (b)**. For all $t \geq 1$, we derive:

$$\frac{\max(\mathbf{v}^t)}{\min(\mathbf{v}^t)} \overset{①}{\leq} \sqrt{\frac{\overline{\mathrm{v}}^2 + (\alpha+\beta)\mathcal{R}_{t-1}}{\underline{\mathrm{v}}^2 + \alpha\mathcal{R}_{t-1}}} \overset{②}{\leq} \sqrt{\max(\frac{\overline{\mathrm{v}}^2}{\underline{\mathrm{v}}^2}, \frac{(\alpha+\beta)\mathcal{R}_{t-1}}{\alpha\mathcal{R}_{t-1}})} = \sqrt{\max(1, \frac{\alpha+\beta}{\alpha})} \leq 1 + \sqrt{\beta/\alpha}, \quad (15)$$

where step ① uses **Part (a)** of this lemma; step ② uses Lemma A.2 that $\frac{a+b}{c+d} \leq \max(\frac{a}{c}, \frac{b}{d})$ for all $a, b, c, d > 0$. Clearly, Inequality (15) is valid for $t = 1$ as well.

**Part (c)**. For all $t \geq 0$, we derive the following results:

$$\frac{\min(\mathbf{v}^{t+1})}{\min(\mathbf{v}^t)} \overset{①}{=} \sqrt{\frac{\min(\mathbf{v}^t \odot \mathbf{v}^t + \alpha \|\mathbf{r}^t\|_2^2 \cdot \mathbf{1} + \beta \mathbf{r}^t \odot \mathbf{r}^t)}{\min(\mathbf{v}^t \odot \mathbf{v}^t)}}$$

$$\overset{②}{\leq} \sqrt{\frac{\min(\mathbf{v}^t \odot \mathbf{v}^t) + (\alpha+\beta)\|\mathbf{r}^t\|_2^2}{\min(\mathbf{v}^t \odot \mathbf{v}^t)}}$$

$$\overset{③}{\leq} \sqrt{\frac{\min(\mathbf{v}^t \odot \mathbf{v}^t) + (\alpha+\beta)\max(\mathbf{v}^t)^2 \|\mathbf{x}^{t+1} - \mathbf{x}^t\|_2^2}{\min(\mathbf{v}^t \odot \mathbf{v}^t)}}$$

$$\overset{④}{\leq} \sqrt{\frac{\min(\mathbf{v}^t \odot \mathbf{v}^t) + 4(\alpha+\beta)\max(\mathbf{v}^t)^2 \overline{\mathrm{x}}^2}{\min(\mathbf{v}^t \odot \mathbf{v}^t)}}$$

$$\overset{⑤}{\leq} \sqrt{1 + 4(\alpha+\beta)\kappa^2 \overline{\mathrm{x}}^2}$$

$$\leq 1 + 2\kappa\overline{\mathrm{x}}\sqrt{\alpha + \beta} \triangleq \acute{k},$$

where step ① uses the update rule for $\mathbf{v}^{t+1}$ that $\mathbf{v}^{t+1} = \sqrt{\mathbf{v}^t \odot \mathbf{v}^t + \mathbf{s}^t}$; step ② uses the fact that $\mathbf{r}^t \odot \mathbf{r}^t \leq \|\mathbf{r}^t\|_2^2$; step ③ uses $\mathbf{r}^t \triangleq \mathbf{v}^t \odot \mathbf{d}^t$ with $\mathbf{d}^t \triangleq \mathbf{x}^{t+1} - \mathbf{x}^t$; step ④ uses $\|\mathbf{x}^{t+1} - \mathbf{x}^t\| \leq \|\mathbf{x}^{t+1}\| + \|\mathbf{x}^t\| \leq 2\overline{\mathrm{x}}$; step ⑤ uses $\max(\mathbf{v}^t)/\min(\mathbf{v}^t) \leq \kappa$ for all $t$.

$\square$

**Lemma B.2.** *(Properties of $\sigma^t$)* For all $t \geq 0$, we have the following results.

**(a)** $\theta(1 - \theta)/(\kappa\acute{k}) \leq \sigma^t \leq \theta$.

**(b)** $(\sigma^{t-1} - 1)\mathbf{v}^t + \sigma^t \mathbf{v}^{t+1} \leq -(1 - \theta)^2 \mathbf{v}^t$.

*Proof.* We define $\sigma^t \triangleq \theta(1 - \sigma^{t-1}) \cdot \min(\mathbf{v}^t \div \mathbf{v}^{t+1})$, where $t \geq 0$.

**Part (a)**. We now prove that $\sigma^t \in [0, \theta]$. We complete the proof using mathematical induction. First, we consider $t = 0$, we have:

$$\sigma^0 = \min(\mathbf{v}^t \div \mathbf{v}^{t+1}) \cdot \theta(1 - \sigma^{-1})$$

$$\overset{①}{=} \min(\mathbf{v}^t \div \mathbf{v}^{t+1}) \cdot \theta(1 - \theta) \overset{②}{\leq} \theta(1 - \theta) \overset{③}{\leq} \theta,$$

where step ① uses $\sigma^{-1} = \theta$; step ② uses $\min(\mathbf{v}^t \div \mathbf{v}^{t+1}) \in (0, 1]$; step ③ uses $\theta \in [0, 1)$. Second, we fix some $t$ and assume that $\sigma^{t-1} \in [0, \theta]$. We analyze the following term for all $t \geq 1$:

$$\sigma^t \triangleq \theta(1 - \sigma^{t-1}) \cdot \min(\mathbf{v}^t \div \mathbf{v}^{t+1}).$$

Given $\sigma^{t-1} \in [0, \theta]$, $\min(\mathbf{v}^t \div \mathbf{v}^{t+1}) \in (0, 1]$, and $\theta \in [0, 1)$, we conclude that $\sigma^t \in [0, \theta]$.

We now establish the lower bound for $\sigma^t$. For all $t \geq 0$, we have:

$$\sigma^t \triangleq \theta(1 - \sigma^{t-1}) \cdot \min(\mathbf{v}^t \div \mathbf{v}^{t+1})$$

$$\overset{①}{\geq} \theta(1 - \theta) \cdot \min(\mathbf{v}^t \div \mathbf{v}^{t+1})$$

$$\overset{②}{\geq} \theta(1 - \theta) \cdot \frac{\min(\mathbf{v}^t)}{\max(\mathbf{v}^{t+1})}$$

$$= \theta(1 - \theta) \cdot \frac{\min(\mathbf{v}^{t+1})}{\max(\mathbf{v}^{t+1})} \cdot \frac{\min(\mathbf{v}^t)}{\min(\mathbf{v}^{t+1})}$$

$$\overset{③}{\geq} \theta(1 - \theta) \cdot \frac{1}{\kappa} \cdot \frac{1}{\acute{k}},$$

where step ① uses $\sigma^{t-1} \leq \theta$; step ② uses $\min(\mathbf{a} \div \mathbf{b}) \geq \frac{\min(\mathbf{a})}{\max(\mathbf{b})}$ for all $\mathbf{a} \geq \mathbf{0}$ and $\mathbf{b} > \mathbf{0}$; step ③ uses $\frac{\min(\mathbf{v}^{t+1})}{\max(\mathbf{v}^{t+1})} \geq \frac{1}{\kappa}$ and $\frac{\min(\mathbf{v}^t)}{\min(\mathbf{v}^{t+1})} \geq \frac{1}{\kappa}$ for all $t$, as shown in Lemma B.1**(b,c)**.

**Part (b)**. For all $t \geq 0$, we derive the following results:

$$
\begin{aligned}
& (\sigma^{t-1} - 1)\mathbf{v}^t + \sigma^t \mathbf{v}^{t+1} \\
\overset{①}{=}\ & (\sigma^{t-1} - 1)\mathbf{v}^t + \theta(1 - \sigma^{t-1}) \cdot \min(\mathbf{v}^t \div \mathbf{v}^{t+1})\mathbf{v}^{t+1} \\
\overset{②}{\leq}\ & (\sigma^{t-1} - 1)\mathbf{v}^t + \theta(1 - \sigma^{t-1})\mathbf{v}^t \\
=\ & -(1 - \theta)(1 - \sigma^{t-1})\mathbf{v}^t \\
\overset{③}{\leq}\ & -(1 - \theta)(1 - \theta)\mathbf{v}^t \\
=\ & -(1 - \theta)^2 \mathbf{v}^t,
\end{aligned}
$$

where step ① uses the choice for $\sigma^t$ for all $t \geq 0$; step ② uses $\min(\mathbf{a} \div \mathbf{v})\mathbf{v} \leq \mathbf{a}$ for all $\mathbf{a}, \mathbf{v} \in \mathbb{R}^n$ with $\mathbf{v} > \mathbf{0}$; step ③ uses $\sigma^{t-1} \leq \theta < 1$ for all $t \geq 0$.

$\square$

We let $\bar{\mathbf{x}} \in \arg\min_{\mathbf{x}} F(\mathbf{x})$, where $F(\mathbf{x}) \triangleq f(\mathbf{x}) + h(\mathbf{x})$. We define the following sequence:

$$
\mathcal{Z}_t \triangleq \mathbb{E}[F(\mathbf{x}^t) - F(\bar{\mathbf{x}}) + \tfrac{1}{2}\|\mathbf{x}^t - \mathbf{x}^{t-1}\|^2_{\sigma^{t-1}(\mathbf{v}^t + L)}].
$$

**Lemma B.3.** *(Properties of $\mathcal{Z}_t$)* We define $c_1 \triangleq \frac{1}{2}(\frac{1-\theta}{\kappa})^2$, $c_2 \triangleq \frac{3L}{2}$. We have:

$$
\mathcal{Z}_{t+1} - \mathcal{Z}_t \leq \mathbb{E}[\langle \mathbf{d}^t, \nabla f(\mathbf{y}^t) - \mathbf{g}^t \rangle + c_2 \mathbb{S}_2^t - c_1 \mathbb{S}_1^t], \tag{16}
$$

where $\mathbb{S}_1^t \triangleq \frac{\|\mathbf{r}^t\|_2^2}{\min(\mathbf{v}^t)}, \mathbb{S}_2^t \triangleq \frac{\|\mathbf{r}^t\|_2^2}{\min(\mathbf{v}^t)^2}.$

*Proof.* We let $\bar{\mathbf{x}} \in \arg\min_{\mathbf{x}} F(\mathbf{x})$, where $F(\mathbf{x}) \triangleq f(\mathbf{x}) + h(\mathbf{x})$.

We define $\mathbf{a}^t \triangleq \mathbf{y}^t - \mathbf{g}^t \div \mathbf{v}^t$.

We define $Q_t \triangleq \mathbb{E}[\langle \mathbf{d}^t, \nabla f(\mathbf{y}^t) - \mathbf{g}^t \rangle]$, where $\mathbf{d}^t \triangleq \mathbf{x}^{t+1} - \mathbf{x}^t$.

We define $\mathcal{Z}_t \triangleq \mathbb{E}[F(\mathbf{x}^t) - F(\bar{\mathbf{x}}) + \tfrac{1}{2}\|\mathbf{x}^t - \mathbf{x}^{t-1}\|^2_{\sigma^{t-1}(\mathbf{v}^t + L)}].$

We define $\mathcal{X}^t \triangleq \frac{1}{2}\|\mathbf{x}^t - \mathbf{x}^{t-1}\|^2_{\sigma^{t-1}(\mathbf{v}^t + L)}.$

Using the optimality of $\mathbf{x}^{t+1} \in \arg\min_{\mathbf{x}} h(\mathbf{x}) + \frac{1}{2}\|\mathbf{x} - \mathbf{a}^t\|_{\mathbf{v}^t}^2$, we have the following inequality:

$$
\mathbb{E}[h(\mathbf{x}^{t+1}) + \tfrac{1}{2}\|\mathbf{x}^{t+1} - \mathbf{a}^t\|_{\mathbf{v}^t}^2] \leq \mathbb{E}[h(\mathbf{x}^t) + \tfrac{1}{2}\|\mathbf{x}^t - \mathbf{a}^t\|_{\mathbf{v}^t}^2]. \tag{17}
$$

Given $f(\mathbf{x})$ is $L$-smooth, we have:

$$
f(\mathbf{x}^{t+1}) \leq f(\mathbf{x}^t) + \langle \mathbf{x}^{t+1} - \mathbf{x}^t, \nabla f(\mathbf{x}^t) \rangle + \tfrac{L}{2}\|\mathbf{x}^{t+1} - \mathbf{x}^t\|_2^2. \tag{18}
$$

Adding Inequalities (17) and (18) together yields:

$$\mathbb{E}[F(\mathbf{x}^{t+1}) - F(\mathbf{x}^t) - \tfrac{L}{2}\|\mathbf{x}^{t+1} - \mathbf{x}^t\|_2^2]$$

$$\leq \mathbb{E}[\langle \mathbf{x}^{t+1} - \mathbf{x}^t, \nabla f(\mathbf{x}^t)\rangle + \tfrac{1}{2}\|\mathbf{x}^t - \mathbf{a}^t\|_{\mathbf{v}^t}^2 - \tfrac{1}{2}\|\mathbf{x}^{t+1} - \mathbf{a}^t\|_{\mathbf{v}^t}^2]$$

$$\overset{①}{=} \mathbb{E}[\langle \mathbf{x}^{t+1} - \mathbf{x}^t, \nabla f(\mathbf{x}^t)\rangle + \tfrac{1}{2}\|\mathbf{x}^t - \mathbf{x}^{t+1}\|_{\mathbf{v}^t}^2 + \langle \mathbf{a}^t - \mathbf{x}^{t+1}, \mathbf{x}^{t+1} - \mathbf{x}^t\rangle_{\mathbf{v}^t}]$$

$$\overset{②}{=} \mathbb{E}[\langle \mathbf{x}^{t+1} - \mathbf{x}^t, \nabla f(\mathbf{x}^t) - \nabla f(\mathbf{y}^t, \xi^t)\rangle + \tfrac{1}{2}\|\mathbf{x}^t - \mathbf{x}^{t+1}\|_{\mathbf{v}^t}^2 + \langle \mathbf{y}^t - \mathbf{x}^{t+1}, \mathbf{x}^{t+1} - \mathbf{x}^t\rangle_{\mathbf{v}^t}]$$

$$\overset{③}{=} Q_t + \langle \mathbf{x}^{t+1} - \mathbf{x}^t, \nabla f(\mathbf{x}^t) - \nabla f(\mathbf{y}^t)\rangle] + \mathbb{E}[\tfrac{1}{2}\|\mathbf{x}^t - \mathbf{y}^t\|_{\mathbf{v}^t}^2 - \tfrac{1}{2}\|\mathbf{y}^t - \mathbf{x}^{t+1}\|_{\mathbf{v}^t}^2]$$

$$\overset{④}{\leq} Q_t + \mathbb{E}[L\|\mathbf{x}^{t+1} - \mathbf{x}^t\|\|\mathbf{x}^t - \mathbf{y}^t\| + \tfrac{1}{2}\|\mathbf{x}^t - \mathbf{y}^t\|_{\mathbf{v}^t}^2 - \tfrac{1}{2}\|\mathbf{y}^t - \mathbf{x}^{t+1}\|_{\mathbf{v}^t}^2]$$

$$\overset{⑤}{=} Q_t + \mathbb{E}[\sigma^{t-1} L\|\mathbf{x}^{t+1} - \mathbf{x}^t\|\|\mathbf{x}^t - \mathbf{x}^{t-1}\|$$

$$+ \tfrac{(\sigma^{t-1})^2}{2}\|\mathbf{x}^t - \mathbf{x}^{t-1}\|_{\mathbf{v}^t}^2 - \tfrac{1}{2}\|\mathbf{x}^{t+1} - \mathbf{x}^t - \sigma^{t-1}(\mathbf{x}^t - \mathbf{x}^{t-1})\|_{\mathbf{v}^t}^2]$$

$$= Q_t + \mathbb{E}[\sigma^{t-1} L\|\mathbf{x}^{t+1} - \mathbf{x}^t\|\|\mathbf{x}^t - \mathbf{x}^{t-1}\| - \tfrac{1}{2}\|\mathbf{x}^{t+1} - \mathbf{x}^t\|_{\mathbf{v}^t}^2 + \sigma^{t-1}\langle \mathbf{x}^{t+1} - \mathbf{x}^t, \mathbf{x}^t - \mathbf{x}^{t-1}\rangle_{\mathbf{v}^t}]$$

$$\overset{⑥}{\leq} Q_t + \mathbb{E}[\tfrac{\sigma^{t-1} L}{2}\|\mathbf{x}^{t+1} - \mathbf{x}^t\|_2^2 + \tfrac{\sigma^{t-1} L}{2}\|\mathbf{x}^t - \mathbf{x}^{t-1}\|_2^2$$

$$- \tfrac{1}{2}\|\mathbf{x}^{t+1} - \mathbf{x}^t\|_{\mathbf{v}^t}^2 + \tfrac{\sigma^{t-1}}{2}\|\mathbf{x}^{t+1} - \mathbf{x}^t\|_{\mathbf{v}^t}^2 + \tfrac{\sigma^{t-1}}{2}\|\mathbf{x}^t - \mathbf{x}^{t-1}\|_{\mathbf{v}^t}^2]$$

$$= Q_t + \mathbb{E}[-\tfrac{1}{2}\|\mathbf{x}^{t+1} - \mathbf{x}^t\|_{\mathbf{v}^t}^2 + \underbrace{\tfrac{1}{2}\|\mathbf{x}^t - \mathbf{x}^{t-1}\|_{\sigma^{t-1}(\mathbf{v}^t+L)}^2}_{\triangleq \mathcal{X}^t} + \tfrac{1}{2}\|\mathbf{x}^{t+1} - \mathbf{x}^t\|_{\sigma^{t-1}(\mathbf{v}^t+L)}^2], \qquad (19)$$

where step ① uses the Pythagoras Relation as in Lemma A.1 that $\mathbf{x}^+ = \mathbf{x}^{t+1}$, $\mathbf{x} = \mathbf{x}^t$, $\mathbf{a} = \mathbf{a}^t$, and $\mathbf{v} = \mathbf{v}^t$; step ② uses $\mathbf{a}^t \triangleq \mathbf{y}^t - \nabla f(\mathbf{y}^t) \div \mathbf{v}^t$; step ③ the definition of $Q_t$, and the Pythagoras Relation as in Lemma A.1 that $\mathbf{x}^+ = \mathbf{x}^{t+1}$, $\mathbf{x} = \mathbf{x}^t$, $\mathbf{a} = \mathbf{y}^t$, and $\mathbf{v} = \mathbf{v}^t$; step ④ uses $L$-smoothness of $f(\cdot)$; step ⑤ uses $\mathbf{y}^{t+1} - \mathbf{x}^{t+1} = \sigma^t(\mathbf{x}^{t+1} - \mathbf{x}^t)$; step ⑥ uses $ab \leq \tfrac{a^2}{2} + \tfrac{b^2}{2}$ for all $a, b \in \mathbb{R}$, and $\langle \mathbf{a}, \mathbf{b}\rangle_{\mathbf{v}} \leq \tfrac{1}{2}\|\mathbf{a}\|_{\mathbf{v}}^2 + \tfrac{1}{2}\|\mathbf{b}\|_{\mathbf{v}}^2$ for all $\mathbf{a}, \mathbf{b}, \mathbf{v} \in \mathbb{R}^n$ with $\mathbf{v} \geq \mathbf{0}$.

We define $\mathcal{Z}_t \triangleq \mathbb{E}[F(\mathbf{x}^t) - F(\bar{\mathbf{x}}) + \tfrac{1}{2}\|\mathbf{x}^t - \mathbf{x}^{t-1}\|_{\sigma^{t-1}(\mathbf{v}^t+L)}^2]$. Given Inequality (19), we have the following inequalities for all $t \geq 0$:

$$\mathcal{Z}_{t+1} - \mathcal{Z}_t - Q_t$$

$$\leq \mathbb{E}[\tfrac{L}{2}\|\mathbf{x}^{t+1} - \mathbf{x}^t\|_2^2 - \tfrac{1}{2}\|\mathbf{x}^{t+1} - \mathbf{x}^t\|_{\mathbf{v}^t}^2 + \tfrac{1}{2}\|\mathbf{x}^{t+1} - \mathbf{x}^t\|_{\sigma^{t-1}(\mathbf{v}^t+L)}^2 + \mathcal{X}^{t+1}]$$

$$\overset{①}{=} \mathbb{E}[\tfrac{L}{2}\|\mathbf{x}^{t+1} - \mathbf{x}^t\|_2^2 - \tfrac{1}{2}\|\mathbf{x}^{t+1} - \mathbf{x}^t\|_{\mathbf{v}^t}^2 + \tfrac{1}{2}\|\mathbf{x}^{t+1} - \mathbf{x}^t\|_{[\sigma^{t-1}\mathbf{v}^t + \sigma^{t-1}L + \sigma^t \mathbf{v}^{t+1} + \sigma^t L]}^2]$$

$$= \mathbb{E}[\tfrac{L+\sigma^{t-1}L+\sigma^t L}{2}\|\mathbf{x}^{t+1} - \mathbf{x}^t\|_2^2 + \tfrac{1}{2}\|\mathbf{x}^{t+1} - \mathbf{x}^t\|_{[\sigma^{t-1}\mathbf{v}^t + \sigma^t \mathbf{v}^{t+1} - \mathbf{v}^t]}^2]$$

$$\overset{②}{\leq} \mathbb{E}[\tfrac{3L}{2}\|\mathbf{x}^{t+1} - \mathbf{x}^t\|_2^2 - \tfrac{1}{2}(1-\theta)^2\|\mathbf{x}^{t+1} - \mathbf{x}^t\|_{\mathbf{v}^t}^2]$$

$$\overset{③}{\leq} \mathbb{E}[\underbrace{\tfrac{3L}{2}}_{\triangleq c_2} \cdot \underbrace{\tfrac{1}{\min(\mathbf{v}^t)^2}\|\mathbf{v}^t \odot (\mathbf{x}^{t+1} - \mathbf{x}^t)\|_2^2}_{\triangleq \mathbb{S}_2^t}] - \mathbb{E}[\underbrace{\tfrac{1}{2}(\tfrac{1-\theta}{\kappa})^2}_{\triangleq c_1} \cdot \underbrace{\tfrac{1}{\min(\mathbf{v}^t)}\|\mathbf{v}^t \odot (\mathbf{x}^{t+1} - \mathbf{x}^t)\|_2^2}_{\triangleq \mathbb{S}_1^t}],$$

where step ① uses the definition of $\mathcal{X}^t \triangleq \tfrac{1}{2}\|\mathbf{x}^t - \mathbf{x}^{t-1}\|_{\sigma^{t-1}(\mathbf{v}^t+L)}^2$; step ② uses $\sigma^t \leq 1$ for all $t \geq 0$, and $\sigma^t \mathbf{v}^{t+1} + (\sigma^{t-1} - 1)\mathbf{v}^t \leq -(1-\theta)^2 \mathbf{v}^t$ for all $t \geq 0$ as shown in Lemma B.2(b); step ③ uses the following two inequalities for all $\mathbf{d} \in \mathbb{R}^n$ with $\mathbf{d} = \mathbf{x}^{t+1} - \mathbf{x}^t$:

$$\|\mathbf{d}\|_2^2 \leq \tfrac{1}{\min(\mathbf{v}^t)^2}\|\mathbf{v}^t \odot \mathbf{d}\|_2^2,$$

$$\|\mathbf{d}\|_{\mathbf{v}^t}^2 \kappa^2 \geq \|\mathbf{d}\|_{\mathbf{v}^t}^2 \tfrac{\max(\mathbf{v}^t)^2}{\min(\mathbf{v}^t)^2} \geq \|\mathbf{d}\|_2^2 \cdot \min(\mathbf{v}^t) \cdot \tfrac{\max(\mathbf{v}^t)^2}{\min(\mathbf{v}^t)^2} = \|\mathbf{d}\|_2^2 \cdot \tfrac{\max(\mathbf{v}^t)^2}{\min(\mathbf{v}^t)} \geq \|\mathbf{v}^t \odot \mathbf{d}\|_2^2 \cdot \tfrac{1}{\min(\mathbf{v}^t)}.$$

$$\square$$

We now derive the upper bounds for the summation of the terms $\mathbb{S}_1^t$ and $\mathbb{S}_2^t$ as in Lemma B.3.

**Lemma B.4.** *We define $\mathcal{V}_t$ as in Lemma B.1. We have the following results.*

*(a)* $\sum_{t=0}^{T} \mathbb{S}_1^t \leq s_1 \mathcal{V}_{T+1}$, *where* $s_1 \triangleq 2\kappa$.

**(b)** $\sum_{t=0}^{T} \mathbb{S}_2^t \leq s_2 \sqrt{\mathcal{V}_{T+1}}$, *where* $s_2 \triangleq \frac{4\dot{\kappa}^2}{\alpha} \underline{\mathbf{v}}^{-1/2}$.

*Proof.* We define $\mathcal{V}_{t+1} \triangleq \sqrt{\underline{\mathbf{v}}^2 + (\alpha+\beta)\mathcal{R}_t}$, where $\mathcal{R}_t \triangleq \sum_{i=0}^{t} \|\mathbf{r}^i\|_2^2$.

**Part (a)**. We derive the following results:

$$
\begin{aligned}
\sum_{t=0}^{T} \frac{\|\mathbf{r}^t\|_2^2}{\min(\mathbf{v}^t)} &= \sum_{t=0}^{T} \frac{\|\mathbf{r}^t\|_2^2}{\min(\mathbf{v}^{t+1})} \cdot \frac{\min(\mathbf{v}^{t+1})}{\min(\mathbf{v}^t)} \\
&\overset{\text{①}}{\leq} \dot{\kappa} \cdot \sum_{t=0}^{T} \frac{\|\mathbf{r}^t\|_2^2}{\sqrt{\underline{\mathbf{v}}^2 + \alpha \sum_{j=0}^{t} \|\mathbf{r}^j\|_2^2}} \\
&\overset{\text{②}}{\leq} 2\dot{\kappa} \cdot \sqrt{\underline{\mathbf{v}}^2 + \alpha \sum_{t=0}^{T} \|\mathbf{r}^t\|_2^2}, \\
&\leq \underbrace{2\dot{\kappa}}_{\triangleq s_1} \cdot \underbrace{\sqrt{\underline{\mathbf{v}}^2 + (\alpha+\beta) \sum_{t=0}^{T} \|\mathbf{r}^t\|_2^2}}_{\triangleq \mathcal{V}_{T+1}},
\end{aligned}
$$

where step ① uses Lemma B.1(c) that $\min(\mathbf{v}^{t+1})/\min(\mathbf{v}^t) \leq \dot{\kappa}$, and Lemma B.1(a); step ② uses Lemma A.5.

**Part (b)**. We have the following results:

$$
\begin{aligned}
\sum_{t=0}^{T} \frac{\|\mathbf{r}^t\|_2^2}{\min(\mathbf{v}^t)^2} &= \sum_{t=0}^{T} \frac{\|\mathbf{r}^t\|_2^2}{\min(\mathbf{v}^{t+1})^2} \cdot \left(\frac{\min(\mathbf{v}^{t+1})}{\min(\mathbf{v}^t)}\right)^2 \\
&\overset{\text{①}}{\leq} \frac{\dot{\kappa}^2}{\alpha} \cdot \sum_{t=0}^{T} \frac{\|\mathbf{r}^t\|_2^2}{\underline{\mathbf{v}}^2/\alpha + \sum_{j=0}^{t} \|\mathbf{r}^j\|_2^2} \\
&\overset{\text{②}}{\leq} \frac{\dot{\kappa}^2}{\alpha} \cdot \frac{1}{1/4} \cdot \left(1 + \frac{1}{\underline{\mathbf{v}}^2/\alpha} \sum_{t=0}^{T} \|\mathbf{r}^t\|_2^2\right)^{1/4} \\
&= \frac{4\dot{\kappa}^2}{\alpha \cdot \underline{\mathbf{v}}^{1/2}} \cdot \left(\underline{\mathbf{v}}^2 + \alpha \sum_{t=0}^{T} \|\mathbf{r}^t\|_2^2\right)^{1/4} \\
&\overset{\text{③}}{\leq} \underbrace{\frac{4\dot{\kappa}^2}{\alpha \cdot \underline{\mathbf{v}}^{1/2}}}_{\triangleq s_2} \cdot \underbrace{\left(\underline{\mathbf{v}}^2 + (\alpha+\beta) \sum_{t=0}^{T} \|\mathbf{r}^t\|_2^2\right)^{1/4}}_{\triangleq \sqrt{\mathcal{V}_{T+1}}},
\end{aligned}
$$

where step ① uses Lemma B.1(a) and Lemma B.1(c); step ② uses Lemma A.6 with $p = 1/4$, $A_t = \|\mathbf{r}^t\|_2^2$, and $c = \underline{\mathbf{v}}^2/\alpha$; step ③ uses $\beta \geq 0$.

$\square$

## B.2 ANALYSIS FOR AEPG

This subsection provides the convergence analysis of AEPG.

### B.2.1 PROOF OF LEMMA 3.4

*Proof.* We define $\mathcal{V}_{t+1} \triangleq \sqrt{\underline{\mathbf{v}}^2 + (\alpha+\beta)\mathcal{R}_t}$, where $\mathcal{R}_t \triangleq \sum_{i=0}^{t} \|\mathbf{r}^i\|_2^2$.

We define $\mathbf{r}^t \triangleq \mathbf{v}^t \odot \mathbf{d}^t$, where $\mathbf{d}^t \triangleq \mathbf{x}^{t+1} - \mathbf{x}^t$.

Initially, for the full-batch, deterministic setting where $\nabla f(\mathbf{y}^t) = \mathbf{g}^t$, we obtain from Lemma B.3 that

$$
0 \leq \mathcal{Z}_t - \mathcal{Z}_{t+1} + \frac{c_2 \|\mathbf{r}^t\|_2^2}{\min(\mathbf{v}^t)^2} - \frac{c_1 \|\mathbf{r}^t\|_2^2}{\min(\mathbf{v}^t)} \tag{20}
$$

Multiplying both sides of Inequality (20) by $\min(\mathbf{v}^t)$ yields:

$$
0 \leq -c_1 \|\mathbf{r}^t\|_2^2 + \min(\mathbf{v}^t)(\mathcal{Z}_t - \mathcal{Z}_{t+1}) + \frac{c_2 \|\mathbf{r}^t\|_2^2}{\min(\mathbf{v}^t)}.
$$

Summing this inequality over $t$ from $t = 0$ to $T$, we obtain:

$$
\begin{aligned}
0 &\leq -c_1 \sum_{t=0}^{T} \|\mathbf{r}^t\|_2^2 + \sum_{t=0}^{T} \min(\mathbf{v}^t)(\mathcal{Z}_t - \mathcal{Z}_{t+1}) + c_2 \sum_{t=0}^{T} \frac{\|\mathbf{r}^t\|_2^2}{\min(\mathbf{v}^t)} \\
&\overset{\text{①}}{\leq} -c_1 \sum_{t=0}^{T} \|\mathbf{r}^t\|_2^2 + (\max_{t=0}^{T} \mathcal{Z}_t) \cdot \min(\mathbf{v}^T) + c_2 s_1 \mathcal{V}_{T+1} \\
&\overset{\text{②}}{\leq} -\frac{c_1}{\alpha+\beta} \left(\mathcal{V}_{T+1}^2 - \underline{\mathbf{v}}^2\right) + (\max_{t=0}^{T} \mathcal{Z}_t) \cdot \mathcal{V}_{T+1} + c_2 s_1 \mathcal{V}_{T+1}, \tag{21}
\end{aligned}
$$

where step ① uses Lemma A.4 with $A_i = \min(\mathbf{v}^i)$ for all $i \in [T]$ with $A_1 \le A_2 \le \ldots \le A_T$, and $B_j = \mathcal{Z}_j$ for all $j \ge 0$, and Lemma B.4 that $\sum_{t=0}^T \mathbb{S}_1^t \le s_1 \mathcal{V}_{T+1}$; step ② uses the definition of $\mathcal{V}_T$, along with the facts that $\mathbf{v}^t \le \mathbf{v}^{t+1} \le \mathcal{V}_{t+1}$ and $\mathcal{Z}_i \ge 0$.

Notably, the inequality $\sum_{t=0}^T \min(\mathbf{v}^t)(\mathcal{Z}_t - \mathcal{Z}_{t+1}) \le \min(\mathbf{v}^t) \sum_{t=0}^T (\mathcal{Z}_t - \mathcal{Z}_{t+1})$ does not hold in general, as the sequence $\{\mathcal{Z}_t\}_{t=0}^T$ is not necessarily monotonic. To address this, we instead employ the alternative inequality which can be implied by Lemma A.4: $\sum_{t=0}^T \min(\mathbf{v}^t)(\mathcal{Z}_t - \mathcal{Z}_{t+1}) \le \min(\mathbf{v}^T) \max_{t=0}^T \mathcal{Z}_t$.

In view of Inequality (21), we have the following quadratic inequality for all $T \ge 0$:

$$\frac{c_1}{\alpha+\beta}(\mathcal{V}_{T+1})^2 \le \left(c_2 s_1 + \max_{t=0}^T \mathcal{Z}_t\right) \cdot \mathcal{V}_{T+1} + \frac{c_1}{\alpha+\beta}\underline{\mathbf{v}}^2.$$

Applying Lemma A.3 with $a = \frac{c_1}{\alpha+\beta}$, $b = c_2 s_1 + \max_{t=0}^T \mathcal{Z}_t$, $c = \frac{c_1}{\alpha+\beta}\underline{\mathbf{v}}^2$, and $x = \mathcal{V}_{T+1}$ yields:

$$\begin{aligned}
\mathcal{V}_{T+1} &\le \sqrt{c/a} + b/a \\
&= \underline{\mathbf{v}} + \frac{\alpha+\beta}{c_1} \cdot \left(c_2 s_1 + \max_{t=0}^T \mathcal{Z}_t\right) \\
&= \underbrace{\underline{\mathbf{v}} + \frac{\alpha+\beta}{c_1} \cdot c_2 s_1}_{\triangleq w_1} + \underbrace{\frac{\alpha+\beta}{c_1}}_{\triangleq w_2} \cdot \max_{t=0}^T \mathcal{Z}_t,
\end{aligned} \tag{22}$$

The upper bound for $\mathcal{V}_{T+1}$ is established in Inequality (22), but it depends on the unknown variable $(\max_{t=0}^T \mathcal{Z}_t)$.

**Part (a).** We now show that $(\max_{t=0}^T \mathcal{Z}_t)$ is always bounded above by a universal constant $\overline{\mathcal{Z}}$. Dropping the negative term $-\frac{c_1 \|\mathbf{r}^t\|_2^2}{\min(\mathbf{v}^t)}$ on the right-hand side of Inequality (20) and summing over $t$ from $t = 0$ to $T$ yields:

$$\begin{aligned}
\mathcal{Z}_{T+1} &\le \mathcal{Z}_0 + c_2 \sum_{t=0}^T \frac{\|\mathbf{r}^t\|_2^2}{\min(\mathbf{v}^t)^2} \\
&\overset{①}{\le} \mathcal{Z}_0 + c_2 s_2 \sqrt{\mathcal{V}_{T+1}} \\
&\overset{②}{\le} \mathcal{Z}_0 + c_2 s_2 \sqrt{w_1 + w_2 \max_{t=0}^T \mathcal{Z}_t} \\
&\overset{②}{\le} \underbrace{\mathcal{Z}_0 + c_2 s_2 \sqrt{w_1}}_{\triangleq \dot{a}} + \underbrace{c_2 s_2 \sqrt{w_2}}_{\triangleq \dot{b}} \cdot \sqrt{\max_{t=0}^T \mathcal{Z}_t} \tag{23} \\
&\overset{④}{\le} \max(\mathcal{Z}_0, 2\dot{b}^2 + 2\dot{a}) \\
&\overset{⑤}{=} 2\dot{b}^2 + 2\dot{a}, \tag{24}
\end{aligned}$$

where step ① uses Lemma B.4(**b**); step ② uses Inequality (22); step ③ uses $\sqrt{a+b} \le \sqrt{a} + \sqrt{b}$ for all $a, b \ge 0$; step ④ uses Lemma A.7 with $a = \dot{a}$ and $b = \dot{b}$; step ⑤ uses the fact that $\dot{a} \ge \mathcal{Z}_0$. This further leads to:

$$\begin{aligned}
\mathcal{Z}_{T+1} &\le 2\dot{b}^2 + 2\dot{a} \\
&\le \frac{9L^2}{2} s_2^2 w_2 + 2\mathcal{Z}_0 + 3L s_2 \sqrt{w_1} \\
&\le \mathcal{O}(L^2 \underline{\mathbf{v}}^{-1}) + \mathcal{O}(\mathcal{Z}_0) + \mathcal{O}(L\underline{\mathbf{v}}^{-1/2})\sqrt{\mathcal{O}(\underline{\mathbf{v}}) + \mathcal{O}(L)} \triangleq \overline{\mathcal{Z}} \\
&\le \mathcal{O}(1),
\end{aligned}$$

where we use $s_1 \triangleq 2\grave{\kappa} = \mathcal{O}(1)$, $s_2 \triangleq \frac{4\grave{\kappa}^2}{\alpha}\underline{\mathbf{v}}^{-1/2} = \mathcal{O}(\underline{\mathbf{v}}^{-1/2})$, $c_1 \triangleq \frac{1}{2}(\frac{1-\theta}{\kappa})^2 = \mathcal{O}(1)$, $c_2 \triangleq \frac{3L}{2} = \mathcal{O}(L)$, $s_2 \triangleq \frac{4\grave{\kappa}^2}{\alpha \cdot \underline{\mathbf{v}}^{1/2}} = \mathcal{O}(\underline{\mathbf{v}}^{-1/2})$, $w_2 = \frac{\alpha+\beta}{c_1} = \mathcal{O}(1)$, $w_1 = \underline{\mathbf{v}} + \frac{\alpha+\beta}{c_1} \cdot c_2 s_1 = \mathcal{O}(\underline{\mathbf{v}}) + \mathcal{O}(L)$.

**Part (b).** We derive the following inequalities for all $T \ge 0$:

$$\begin{aligned}
\mathcal{V}_{T+1} &\overset{①}{\le} w_1 + w_2 \sqrt{\max_{t=0}^T \mathcal{Z}_t} \\
&\overset{②}{\le} w_1 + w_2 \overline{\mathcal{Z}} \\
&\le \mathcal{O}(\underline{\mathbf{v}}) + \mathcal{O}(L) + \mathcal{O}(L^2 \underline{\mathbf{v}}^{-1}) + \mathcal{O}(\mathcal{Z}_0) + \mathcal{O}(L\underline{\mathbf{v}}^{-1/2})\sqrt{\mathcal{O}(\underline{\mathbf{v}}) + \mathcal{O}(L)} \triangleq \overline{\mathbf{v}} \\
&\le \mathcal{O}(1),
\end{aligned}$$

where step ① uses Inequality (22); step ② uses Inequality (23).

$\square$

### B.2.2 PROOF OF THEOREM 3.5

*Proof.* **Part (a)**. We have the following inequalities:

$$
\begin{aligned}
\sum_{t=0}^{T} \|\mathbf{x}^{t+1} - \mathbf{x}^t\|_2^2 
&\overset{①}{\leq} \tfrac{1}{\underline{v}^2} \min(\mathbf{v}^t)^2 \sum_{t=0}^{T} \|\mathbf{x}^{t+1} - \mathbf{x}^t\|_2^2 \\
&\overset{②}{\leq} \tfrac{1}{\underline{v}^2} \sum_{t=0}^{T} \|\mathbf{v}^t \odot (\mathbf{x}^{t+1} - \mathbf{x}^t)\|_2^2 \\
&\overset{③}{=} \tfrac{1}{\underline{v}^2} \sum_{t=0}^{T} \|\mathbf{r}^t\|_2^2 = \tfrac{1}{\underline{v}^2} \mathcal{R}_T \\
&\overset{④}{\leq} \tfrac{1}{\underline{v}^2} \tfrac{1}{\alpha} (\overline{\mathrm{v}}^2 - \underline{\mathrm{v}}^2) \triangleq \overline{\mathrm{X}},
\end{aligned}
\tag{25}
$$

where step ① uses $\mathbf{v}^t \geq \underline{v}$; step ② uses $\min(\mathbf{v})\|\mathbf{d}\| \leq \|\mathbf{d}\|_{\mathbf{v}}$ for all $\mathbf{v}, \mathbf{d} \in \mathbb{R}^n$ with $\mathbf{v} \geq \mathbf{0}$; step ③ uses the definition of $\mathbf{r}^t \triangleq \mathbf{v}^t \odot \mathbf{d}^t$; step ④ uses Lemma B.1(a) that $\underline{v}^2 + \alpha \mathcal{R}_t \leq (\mathbf{v}^{t+1})^2 \leq \overline{v}^2$ for all $t$.

**Part (b)**. First, by the first-order necessarily condition of $\mathbf{x}^{t+1}$ that $\mathbf{x}^{t+1} \in \mathrm{Prox}_h(\mathbf{y} - \mathbf{g}^t \div \mathbf{v}^t; \mathbf{v}^t) = \arg\min_{\mathbf{x}} h(\mathbf{x}) + \tfrac{1}{2}\|\mathbf{x} - (\mathbf{y} - \mathbf{g}^t \div \mathbf{v}^t)\|_{\mathbf{v}^t}^2$, we have:

$$
\mathbf{0} \in \partial h(\mathbf{x}^{t+1}) + \mathbf{g}^t + \mathbf{v}^t \odot (\mathbf{x}^{t+1} - \mathbf{y}^t).
\tag{26}
$$

Second, we obtain:

$$
\begin{aligned}
\|\nabla f(\mathbf{x}^{t+1}) + \partial h(\mathbf{x}^{t+1})\| 
&\overset{①}{=} \|\nabla f(\mathbf{x}^{t+1}) - \nabla f(\mathbf{y}^t) - \mathbf{v}^t \odot (\mathbf{x}^{t+1} - \mathbf{y}^t)\| \\
&\overset{②}{\leq} L\|\mathbf{y}^t - \mathbf{x}^{t+1}\| + \max(\mathbf{v}^t)\|\mathbf{y}^t - \mathbf{x}^{t+1}\| \\
&\overset{③}{=} (L + \max(\mathbf{v}^t)) \cdot \|\mathbf{x}^t + \sigma^{t-1}(\mathbf{x}^t - \mathbf{x}^{t-1}) - \mathbf{x}^{t+1}\| \\
&\overset{④}{\leq} (L + \overline{\mathrm{v}}) \cdot (\|\mathbf{x}^t - \mathbf{x}^{t+1}\| + \|\mathbf{x}^t - \mathbf{x}^{t-1}\|),
\end{aligned}
\tag{27}
$$

where step ① uses Equality (26) with $\mathbf{g}^t = \nabla f(\mathbf{y})$, as in AEPG; step ② uses the triangle inequality, the fact that $f(\mathbf{y})$ is $L$-smooth, and $\|\mathbf{v}^t \odot \mathbf{a}\| \leq \max(\mathbf{v}^t)\|\mathbf{a}\|$ for all $\mathbf{a} \in \mathbb{R}^n$; step ③ uses $\mathbf{y}^t = \mathbf{x}^t + \sigma^{t-1}(\mathbf{x}^t - \mathbf{x}^{t-1})$; step ④ uses $\mathbf{v}^t \leq \overline{\mathrm{v}}$, the triangle inequality, and $\sigma^{t-1} \leq 1$ for all $t$.

Third, we obtain the following results:

$$
\begin{aligned}
&\sum_{t=0}^{T} \|\partial h(\mathbf{x}^{t+1}) + \nabla f(\mathbf{x}^{t+1})\|_2^2 \\
&\overset{①}{\leq} 2(L + \overline{\mathrm{v}})^2 \cdot \sum_{t=0}^{T} (\|\mathbf{x}^{t+1} - \mathbf{x}^t\|_2^2 + \|\mathbf{x}^t - \mathbf{x}^{t-1}\|_2^2) \\
&= 2(L + \overline{\mathrm{v}})^2 \cdot \{\sum_{t=0}^{T} \|\mathbf{x}^{t+1} - \mathbf{x}^t\|_2^2 + \sum_{t=-1}^{T-1} \|\mathbf{x}^{t+1} - \mathbf{x}^t\|_2^2\} \\
&= 2(L + \overline{\mathrm{v}})^2 \cdot \{\|\mathbf{x}^{-1} - \mathbf{x}^0\|_2^2 - \|\mathbf{x}^{T+1} - \mathbf{x}^T\|_2^2 + 2\sum_{t=0}^{T} \|\mathbf{x}^{t+1} - \mathbf{x}^t\|_2^2\} \\
&\overset{②}{\leq} 2(L + \overline{\mathrm{v}})^2 \cdot \sum_{t=0}^{T} \|\mathbf{x}^{t+1} - \mathbf{x}^t\|_2^2 \\
&\overset{③}{\leq} 2(L + \overline{\mathrm{v}}) \cdot \overline{\mathrm{X}} = \mathcal{O}(1),
\end{aligned}
\tag{28}
$$

where step ① uses Inequality (27); step ② uses the choice $\mathbf{x}^{-1} = \mathbf{x}^0$ as shown in Algorithm 1, and $-\|\mathbf{x}^{T+1} - \mathbf{x}^T\| \leq 0$; step ③ uses Inequality (25).

Finally, using the inequality $\|\mathbf{a}\|_2^2 \geq \tfrac{1}{T+1}(\|\mathbf{a}\|_1)^2$ for all $\mathbf{a} \in \mathbb{R}^{T+1}$, we deduce from Inequality (28) that

$$
\tfrac{1}{T+1} \sum_{t=0}^{T} \|\partial h(\mathbf{x}^{t+1}) + \nabla f(\mathbf{x}^{t+1})\| = \mathcal{O}(\tfrac{1}{\sqrt{T+1}}).
$$

$\square$

### B.3 ANALYSIS FOR AEPG-SPIDER

This subsection provides the convergence analysis of AEPG-SPIDER.

We fix $q \geq 1$. For all $t \geq 0$, we denote $r_t \triangleq \lfloor \frac{t}{q} \rfloor + 1$ [1], leading to $(r_t - 1)q \leq t \leq r_t q - 1$.

We introduce an auxiliary lemma from Fang et al. (2018).

**Lemma B.5.** *(Lemma 1 in Fang et al. (2018)) The SPIDER estimator produces a stochastic gradient* $\mathbf{g}^t$ *that, for all $t$ with $(r_t - 1)q \leq t \leq r_t q - 1$, we have:* $\mathbb{E}[\|\mathbf{g}^t - \nabla f(\mathbf{y}^t)\|_2^2] - \|\mathbf{g}^{t-1} - \nabla f(\mathbf{y}^{t-1})\|_2^2 \leq \frac{L^2}{b} Y_{t-1}$, *where* $Y_i \triangleq \mathbb{E}[\|\mathbf{y}^{i+1} - \mathbf{y}^i\|_2^2]$.

Based on Lemma B.3, we have the following results for AEPG-SPIDER.

**Lemma B.6.** *For any positive constant $\phi$, we define $c_1 \triangleq \frac{1}{2}(\frac{1-\theta}{\kappa})^2$, $c_2' \triangleq \frac{(3+\phi)L}{2}$, $c_3 \triangleq \frac{L}{2\phi}\frac{q}{b}$. We define $Y_i \triangleq \mathbb{E}[\|\mathbf{y}^{i+1} - \mathbf{y}^i\|_2^2]$. For all $t$ with $(r_t - 1)q \leq t \leq r_t q - 1$, we have:*

*(a)* $\mathbb{E}[\|\mathbf{g}^t - \nabla f(\mathbf{y}^t)\|_2^2] \leq \frac{L^2}{b} \sum_{i=(r_t-1)q}^{t-1} Y_i$.

*(b)* $\mathcal{Z}_{t+1} - \mathcal{Z}_t + \mathbb{E}[c_1 \mathbb{S}_1^t - c_2' \mathbb{S}_2^t] \leq \frac{c_3}{q} \sum_{i=(r_t-1)q}^{t-1} Y_i$.

*Proof.* We define $c_1 \triangleq \frac{1}{2}(\frac{1-\theta}{\kappa})^2$, $c_2' \triangleq \frac{(3+\phi)L}{2}$, and $c_3 \triangleq \frac{L}{2\phi}\frac{q}{b}$, where $\phi > 0$ can be any constant.

We define $\mathcal{Z}_t \triangleq F(\mathbf{x}^t) - F(\bar{\mathbf{x}}) + \frac{1}{2}\|\mathbf{x}^t - \mathbf{x}^{t-1}\|_{\sigma^{t-1}(\mathbf{v}^t+L)}^2$.

We define $Y_i \triangleq \mathbb{E}[\|\mathbf{y}^{i+1} - \mathbf{y}^i\|_2^2]$.

We define $\mathbb{S}_1^t \triangleq \frac{\|\mathbf{r}^t\|_2^2}{\min(\mathbf{v}^t)}$, and $\mathbb{S}_2^t \triangleq \frac{\|\mathbf{r}^t\|_2^2}{\min(\mathbf{v}^t)^2}$.

**Part (a).** Telescoping the inequality $\mathbb{E}[\|\mathbf{g}^t - \nabla f(\mathbf{y}^t)\|_2^2] - \|\mathbf{g}^{t-1} - \nabla f(\mathbf{y}^{t-1})\|_2^2 \leq \frac{L^2}{b}\mathbb{E}[\|\mathbf{y}^t - \mathbf{y}^{t-1}\|_2^2]$ (as stated in Lemma B.5) over $t$ from $(r_t - 1)q + 1$ to $t$, where $t \leq r_t q - 1$, we obtain:

$$\mathbb{E}[\|\mathbf{g}^t - \nabla f(\mathbf{y}^t)\|_2^2]$$
$$\leq \quad \mathbb{E}[\|\mathbf{g}^{(r_t-1)q} - \nabla f(\mathbf{y}^{(r_t-1)q})\|_2^2] + \frac{L^2}{b}\sum_{i=(r_t-1)q+1}^t \mathbb{E}[\|\mathbf{y}^i - \mathbf{y}^{i-1}\|_2^2]$$
$$\overset{①}{=} \quad 0 + \frac{L^2}{b}\sum_{i=(r_t-1)q}^{t-1} \underbrace{\mathbb{E}[\|\mathbf{y}^{i+1} - \mathbf{y}^i\|_2^2]}_{\triangleq Y_i}, \tag{29}$$

where step ① uses $\mathbf{g}^j = \nabla f(\mathbf{y}^j)$ when $j$ is a multiple of $q$. Notably, Inequality (29) holds for every $t$ of the form $t = (r_t - 1)q$, since at these points we have $\mathbf{g}^t = \nabla f(\mathbf{y}^t)$.

**Part (b).** For all $t$ with $(r_t - 1)q \leq t \leq r_t q - 1$, we have:

$$\mathcal{Z}_{t+1} - \mathcal{Z}_t + c_1 \mathbb{S}_1^t \overset{①}{\leq} \mathbb{E}[\langle \mathbf{d}^t, \nabla f(\mathbf{y}^t) - \mathbf{g}^t \rangle + c_2 \mathbb{S}_2^t]$$
$$= \mathbb{E}[\langle \mathbf{d}^t, \nabla f(\mathbf{y}^t) - \mathbf{g}^t \rangle + \frac{3L}{2\min(\mathbf{v}^t)^2}\|\mathbf{r}^t\|_2^2]$$
$$\overset{②}{\leq} \frac{3L}{2\min(\mathbf{v}^t)^2}\|\mathbf{r}^t\|_2^2 + \frac{\phi L}{2}\|\mathbf{d}^t\|_2^2 + \frac{1}{2\phi L}\|\nabla f(\mathbf{y}^t) - \mathbf{g}^t\|_2^2$$
$$\overset{③}{\leq} \frac{3L}{2\min(\mathbf{v}^t)^2}\|\mathbf{r}^t\|_2^2 + \frac{\phi L}{2\min(\mathbf{v}^t)^2}\|\mathbf{r}^t\|_2^2 + \frac{L}{2b\phi}\cdot\sum_{i=(r_t-1)q}^{t-1} Y_i$$
$$= \underbrace{\frac{(3+\phi)L}{2}}_{\triangleq c_2'} \cdot \underbrace{\frac{1}{\min(\mathbf{v}^t)^2}\|\mathbf{r}^t\|_2^2}_{\mathbb{S}_2^t} + \underbrace{\frac{L}{2\phi}\frac{q}{b}}_{\triangleq c_3} \cdot \frac{1}{q}\sum_{i=(r_t-1)q}^{t-1} Y_i,$$

where step ① uses Lemma B.3; step ② uses $\langle \mathbf{a}, \mathbf{b} \rangle \leq \frac{\phi L}{2}\|\mathbf{a}\|_2^2 + \frac{1}{2\phi L}\|\mathbf{b}\|_2^2$ for all $\mathbf{a}, \mathbf{b} \in \mathbb{R}^n$, and $\phi > 0$; step ③ uses $\min(\mathbf{v}^t)\|\mathbf{d}^t\| \leq \|\mathbf{d}^t \odot \mathbf{v}^t\|$, and Inequality (29).

$\square$

---

[1]For example, if $q = 3$ and $t \in \{0, 1, 2, 3, 4, 5, 6, 7, 8, 9\}$, then the corresponding values of $r_t$ are $\{1, 1, 1, 2, 2, 2, 3, 3, 3, 4\}$.

Based on Lemma B.4, we obtain the following results.

**Lemma B.7.** *We define $V_t \triangleq \min(\mathbf{v}^t)$, and $Y_i \triangleq \mathbb{E}[\|\mathbf{y}^{i+1} - \mathbf{y}^i\|_2^2]$. We have:*

*(a)* $\sum_{t=0}^T V_t Y_t \leq u_1 \mathbb{E}[\mathcal{V}_{T+1}]$, *where* $u_1 \triangleq (8 + 2\dot{\kappa}) s_1$.

*(b)* $\sum_{t=0}^T Y_t \leq u_2 \mathbb{E}[\sqrt{\mathcal{V}_{T+1}}]$, *where* $u_2 \triangleq 10 s_2$.

*Proof.* We define $V_t \triangleq \min(\mathbf{v}^t)$, and $Y_i \triangleq \mathbb{E}[\|\mathbf{y}^{i+1} - \mathbf{y}^i\|_2^2]$.

First, for all $\tau > 0$, we have the following results:

$$
\begin{aligned}
\|\mathbf{y}^{t+1} - \mathbf{y}^t\|_2^2 \quad &\overset{①}{=} \quad \|(\mathbf{x}^{t+1} + \sigma^t \mathbf{d}^t) - (\mathbf{x}^t + \sigma^{t-1} \mathbf{d}^{t-1})\|_2^2 \\
&= \quad \|(1 + \sigma^t)\mathbf{d}^t - \sigma^{t-1} \mathbf{d}^{t-1}\|_2^2 \\
&\overset{②}{\leq} \quad [(1 + \tau)\|(1 + \sigma^t)\mathbf{d}^t\|_2^2 + (1 + 1/\tau)\|\sigma^{t-1}\mathbf{d}^{t-1}\|_2^2] \\
&\overset{③}{\leq} \quad 4(1 + \tau)\|\mathbf{d}^t\|_2^2 + (1 + 1/\tau)\|\mathbf{d}^{t-1}\|_2^2,
\end{aligned}
\tag{30}
$$

where step ① uses $\mathbf{y}^{t+1} = \mathbf{x}^{t+1} + \sigma^t \mathbf{d}^t$; step ② uses $\|\mathbf{a} + \mathbf{b}\|_2^2 \leq (1 + \tau)\|\mathbf{a}\|_2^2 + (1 + 1/\tau)\|\mathbf{b}\|_2^2$ for all $\tau > 0$; step ③ uses $\sigma^t \leq \theta < 1$.

Second, we obtain the following inequalities:

$$
\begin{aligned}
\min(\mathbf{v}^t)\|\mathbf{y}^{t+1} - \mathbf{y}^t\|_2^2 \quad &\overset{①}{=} \quad \min(\mathbf{v}^t) \cdot [4(1 + \tau)\|\mathbf{d}^t\|_2^2 + (1 + 1/\tau)\|\mathbf{d}^{t-1}\|_2^2] \\
&\overset{②}{\leq} \quad 4(1 + \tau)\min(\mathbf{v}^t)\|\mathbf{d}^t\|_2^2 + (1 + 1/\tau)\dot{\kappa}\min(\mathbf{v}^{t-1})\|\mathbf{d}^{t-1}\|_2^2 \\
&\overset{③}{\leq} \quad (4 + \dot{\kappa})\left(\min(\mathbf{v}^t)\|\mathbf{d}^t\|_2^2 + \min(\mathbf{v}^{t-1})\|\mathbf{d}^{t-1}\|_2^2\right),
\end{aligned}
\tag{31}
$$

where step ① uses Inequality (30); step ② uses $\min(\mathbf{v}^t) \leq \min(\mathbf{v}^{t-1})\dot{\kappa}$, as shown in Lemma B.1(**c**); step ③ uses the choice $\tau = \frac{\dot{\kappa}}{4}$.

**Part (a)**. We have the following inequities:

$$
\begin{aligned}
\sum_{t=0}^T \min(\mathbf{v}^t)\|\mathbf{y}^{t+1} - \mathbf{y}^t\|_2^2 \quad &\overset{①}{\leq} \quad (4 + \dot{\kappa})\left(\sum_{t=0}^T \min(\mathbf{v}^t)\|\mathbf{d}^t\|_2^2 + \sum_{t=0}^T \min(\mathbf{v}^{t-1})\|\mathbf{d}^{t-1}\|_2^2\right) \\
&= \quad (4 + \dot{\kappa})\left(\sum_{t=0}^T \min(\mathbf{v}^t)\|\mathbf{d}^t\|_2^2 + \sum_{t=-1}^{T-1} \min(\mathbf{v}^t)\|\mathbf{d}^t\|_2^2\right) \\
&\overset{②}{=} \quad (4 + \dot{\kappa})\left(\sum_{t=0}^T \min(\mathbf{v}^t)\|\mathbf{d}^t\|_2^2 + \sum_{t=0}^{T-1} \min(\mathbf{v}^t)\|\mathbf{d}^t\|_2^2\right) \\
&\leq \quad (8 + 2\dot{\kappa})\left(\sum_{t=0}^T \min(\mathbf{v}^t)\|\mathbf{d}^t\|_2^2\right) \\
&\overset{③}{\leq} \quad (8 + 2\dot{\kappa})\sum_{t=0}^T \min(\mathbf{v}^t) \cdot \frac{1}{\min(\mathbf{v}^t)^2}\|\mathbf{v}^t \odot \mathbf{d}^t\|_2^2 \\
&\overset{④}{\leq} \quad \underbrace{(8 + 2\dot{\kappa}) \cdot s_1}_{\triangleq u_1} \cdot \mathcal{V}_{T+1},
\end{aligned}
\tag{32}
$$

where step ① uses Inequality (31); step ② uses $\mathbf{d}^{-1} = \mathbf{x}^0 - \mathbf{x}^{-1} = \mathbf{0}$; step ③ uses $\min(\mathbf{v}^t)\|\mathbf{d}^t\| \leq \|\mathbf{v}^t \odot \mathbf{d}^t\|$; step ④ uses $\sum_{t=0}^T \mathbb{S}_1^t \leq s_1 \mathcal{V}_{T+1}$ with $\mathbb{S}_1^t \triangleq \frac{\|\mathbf{r}^t\|_2^2}{\min(\mathbf{v}^t)}$, as shown in Lemma B.4(**a**).

**Part (b).** We obtain the following inequities:

$$
\begin{aligned}
\sum_{t=0}^{T} \|\mathbf{y}^{t+1} - \mathbf{y}^t\|_2^2 
&\overset{\textcircled{1}}{\leq} \sum_{t=0}^{T} \left(8\|\mathbf{d}^t\|_2^2 + 2\|\mathbf{d}^{t-1}\|_2^2\right) \\
&\overset{\textcircled{2}}{=} 8\left(\sum_{t=0}^{T} \|\mathbf{d}^t\|_2^2\right) + 2\left(\sum_{t=1}^{T} \|\mathbf{d}^{t-1}\|_2^2\right) \\
&= 8\left(\sum_{t=0}^{T} \|\mathbf{d}^t\|_2^2\right) + 2\left(\sum_{t=0}^{T-1} \|\mathbf{d}^t\|_2^2\right) \\
&\leq 10\left(\sum_{t=0}^{T} \|\mathbf{d}^t\|_2^2\right) \\
&\overset{\textcircled{3}}{\leq} 10\left(\sum_{t=0}^{T} \tfrac{1}{\min(\mathbf{v}^t)^2}\|\mathbf{v}^t \odot \mathbf{d}^t\|_2^2\right) \\
&\overset{\textcircled{4}}{\leq} \underbrace{10 \cdot s_2}_{\triangleq u_2} \cdot \sqrt{\mathcal{V}_{T+1}},
\end{aligned}
$$

where step $\textcircled{1}$ uses Inequality (30) with $\tau = 1$; step $\textcircled{2}$ uses $\mathbf{d}^{-1} = \mathbf{x}^0 - \mathbf{x}^{-1} = \mathbf{0}$; step $\textcircled{3}$ uses $\min(\mathbf{v}^t)\|\mathbf{d}^t\| \leq \|\mathbf{v}^t \odot \mathbf{d}^t\|$; step $\textcircled{4}$ uses $\sum_{t=0}^{T} \mathbb{S}_2^t \leq s_2 \sqrt{\mathcal{V}_{T+1}}$ with $\mathbb{S}_2^t \triangleq \frac{\|\mathbf{r}^t\|_2^2}{\min(\mathbf{v}^t)^2}$, as shown in Lemma B.4(b).

$\square$

The following lemma simplifies the analysis by reducing double summations involving $V_i$ and $Y_i$ to single summations, thereby facilitating the bounding of cumulative terms.

**Lemma B.8.** *We define* $Y_i \triangleq \mathbb{E}[\|\mathbf{y}^{i+1} - \mathbf{y}^i\|_2^2]$, $V_i \triangleq \min(\mathbf{v}^i)$, *and* $q' \triangleq q\grave{\kappa}^{q-1}$. *For all* $t$ *with* $(r_t - 1)q \leq t \leq r_t q - 1$, *we have:*

*(a)* $\sum_{j=(r_t-1)q}^{t}[V_j \sum_{i=(r_j-1)q}^{j-1} Y_i] \leq q' \sum_{i=(r_t-1)q}^{t-1} V_i Y_i$.

*(b)* $\sum_{j=(r_t-1)q}^{t}[\sum_{i=(r_j-1)q}^{j-1} Y_i] \leq (q-1) \sum_{i=(r_t-1)q}^{t-1} Y_i$.

*(c)* $\sum_{t=0}^{T}[\sum_{i=(r_t-1)q}^{t-1} Y_i] \leq (q-1) \sum_{t=1}^{T} Y_t$.

*Proof.* We define $V_t \triangleq \min(\mathbf{v}^t)$, where $\{V_j\}_{j=0}^{\infty}$ is non-decreasing. We define $q' \triangleq q\grave{\kappa}^{q-1}$.

For any integer $t \geq 0$, we derive the following inequalities:

$$
t - (r_t - 1)q \overset{\textcircled{1}}{=} t - (\lfloor \tfrac{t}{q} \rfloor + 1 - 1)q = t - \lfloor \tfrac{t}{q} \rfloor q \overset{\textcircled{2}}{\leq} q - 1, \tag{33}
$$

where step $\textcircled{1}$ uses $r_t \triangleq \lfloor \tfrac{t}{q} \rfloor + 1$; step $\textcircled{2}$ uses the fact that $t - \lfloor \tfrac{t}{q} \rfloor q \leq q - 1$ for all integer $t \geq 0$ and $q \geq 1$.

**Part (a).** For any $t$ with $t \geq (r_t - 1)q$, we have the following results:

$$
\begin{aligned}
\frac{\min(\mathbf{v}^t)}{\min(\mathbf{v}^{(r_t-1)q})} 
&= \frac{\min(\mathbf{v}^{(r_t-1)q+1})}{\min(\mathbf{v}^{(r_t-1)q})} \cdot \frac{\min(\mathbf{v}^{(r_t-1)q+2})}{\min(\mathbf{v}^{(r_t-1)q+1})} \cdots \cdot \frac{\min(\mathbf{v}^t)}{\min(\mathbf{v}^{t-1})} \\
&\overset{\textcircled{1}}{\leq} \grave{\kappa}^{q-1}, 
\end{aligned} \tag{34}
$$

where step $\textcircled{1}$ uses the fact that the product length is at most $([t] - [(r_t - 1)q + 1] + 1)$ and Inequality (33).

For all $t$ with $(r_t - 1)q \leq t \leq r_t q - 1$, we have:

$$
\begin{aligned}
\sum_{j=(r_t-1)q}^{t} \left( V_j \cdot \sum_{i=(r_j-1)q}^{j-1} Y_i \right)
&\overset{①}{\leq} \sum_{j=(r_t-1)q}^{t} \left( V_j \cdot \sum_{i=(r_j-1)q}^{t-1} Y_i \right) \\
&\overset{②}{=} \sum_{j=(r_t-1)q}^{t} \left( V_j \cdot \sum_{i=(r_t-1)q}^{t-1} Y_i \right) \\
&\overset{③}{=} \sum_{i=(r_t-1)q}^{t-1} \left( Y_i \cdot \sum_{j=(r_t-1)q}^{t} V_j \right) \\
&\overset{④}{\leq} \sum_{i=(r_t-1)q}^{t-1} \left( Y_i \cdot \sum_{j=(r_t-1)q}^{t} V_t \right) \\
&\overset{⑤}{\leq} q V_t \sum_{i=(r_t-1)q}^{t-1} Y_i \\
&\overset{⑥}{\leq} q(\grave{\kappa}^{q-1} V_{(r_t-1)q}) \sum_{i=(r_t-1)q}^{t-1} Y_i \\
&\overset{⑦}{\leq} \underbrace{q \grave{\kappa}^{q-1}}_{\triangleq q'} \cdot \sum_{i=(r_t-1)q}^{t-1} V_i Y_i,
\end{aligned}
$$

where step ① uses $j \leq t$ for all $j \in [(r_t - 1)q, t]$; step ② uses $r_j = r_t$ for all $j \in [(r_t - 1)q, t]$ with $t \in [(r_t - 1)q, r_t q - 1]$; step ③ uses the fact that $\sum_{j=\underline{j}}^{\bar{j}}(\mathbf{a}_j \sum_{i=\underline{i}}^{\bar{i}} \mathbf{b}_i) = \sum_{i=\underline{i}}^{\bar{i}}(\mathbf{b}_i \sum_{j=\underline{j}}^{\bar{j}} \mathbf{a}_j)$ for all $\underline{i} \leq \bar{i}$ and $\underline{j} \leq \bar{j}$; step ④ uses $V_j \leq V_t$ as $j \leq t$; step ⑤ uses $t - (r_t - 1)q + 1 \leq q$; step ⑥ uses Inequality (34); step ⑦ uses $i \geq (r_t - 1)q$.

**Part (b)**. For all $t$ with $(r_t - 1)q \leq t \leq r_t q - 1$, we have:

$$
\begin{aligned}
\sum_{j=(r_t-1)q}^{t} \sum_{i=(r_j-1)q}^{j-1} Y_i
&\overset{①}{=} \sum_{i=(r_t-1)q}^{t-1} (t - i) Y_i \\
&\overset{②}{\leq} ([t-1] - [(r_t - 1)q] + 1) \cdot \sum_{i=(r_t-1)q}^{t-1} Y_i \\
&\overset{③}{\leq} (q-1) \sum_{i=(r_t-1)q}^{t-1} Y_i,
\end{aligned}
$$

where step ① uses basic reduction; step ② uses $i \geq (r_t - 1)q$; step ③ uses Inequality (33).

**Part (c)**. We have the following results:

$$
\begin{aligned}
\sum_{t=0}^{T} \left[ \sum_{i=(r_t-1)q}^{t-1} Y_i \right]
&\overset{①}{\leq} ([t-1] - [(r_t - 1)q] + 1) \sum_{t=0}^{T} Y_t \\
&\overset{②}{\leq} (q-1) \sum_{t=0}^{T} Y_t,
\end{aligned}
$$

step ① uses the fact that the length of the summation is $([t-1] - [(r_t - 1)q] + 1)$; step ② uses Inequality (33).

$\square$

### B.3.1 PROOF OF LEMMA 3.7

*Proof.* We define $V_j = \min(\mathbf{v}^j)$ and $Y_i \triangleq \mathbb{E}[\|\mathbf{y}^{i+1} - \mathbf{y}^i\|_2^2]$.

We define $\mathbb{S}_1^t \triangleq \frac{\|\mathbf{r}^t\|_2^2}{\min(\mathbf{v}^t)}$, and $\mathbb{S}_2^t \triangleq \frac{\|\mathbf{r}^t\|_2^2}{\min(\mathbf{v}^t)^2}$.

**Part (a)**. For all $t$ with $(r_t - 1)q \leq t \leq r_t q - 1$, we have from Lemma B.6:

$$
\mathcal{Z}_{t+1} - \mathcal{Z}_t \leq \mathbb{E}[c_2' \cdot \underbrace{\frac{1}{\min(\mathbf{v}^t)^2} \|\mathbf{r}^t\|_2^2}_{\triangleq \mathbb{S}_2^t} - \frac{c_1}{\min(\mathbf{v}^t)} \|\mathbf{r}^t\|_2^2 + \frac{c_3}{q} \cdot \sum_{i=(r_t-1)q}^{t-1} Y_i]. \tag{35}
$$

Multiplying both sides by $\min(\mathbf{v}^t)$ yields:

$$
0 \leq \min(\mathbf{v}^t)[\mathcal{Z}_t - \mathcal{Z}_{t+1}] + \mathbb{E}[c_2' \cdot \underbrace{\frac{1}{\min(\mathbf{v}^t)} \|\mathbf{r}^t\|_2^2}_{\triangleq \mathbb{S}_1^t} - c_1 \|\mathbf{r}^t\|_2^2 + \frac{c_3}{q} \cdot \min(\mathbf{v}^t) \cdot \sum_{i=(r_t-1)q}^{t-1} Y_i]. \tag{36}
$$

Telescoping Inequality (36) over $t$ from $(r_t - 1)q$ to $t$ with $t \leq r_t q - 1$, we have:

$$0 \leq \sum_{j=(r_t-1)q}^{t} \left(V_j(\mathcal{Z}_j - \mathcal{Z}_{j+1}) + \mathbb{E}[c_2'\mathbb{S}_1^t - c_1\|\mathbf{r}^j\|_2^2]\right) + \frac{c_3}{q}\sum_{j=(r_t-1)q}^{t}[V_j \cdot \sum_{i=(r_j-1)q}^{j-1} Y_i]$$

$$\overset{①}{\leq} \underbrace{\sum_{j=(r_t-1)q}^{r_t q-1} \left(V_j(\mathcal{Z}_j - \mathcal{Z}_{j+1}) + \mathbb{E}[c_2'\mathbb{S}_1^t - c_1\|\mathbf{r}^j\|_2^2 + \frac{c_3}{q}\cdot q' \cdot V_j Y_j]\right)}_{\triangleq \mathbb{U}^j},$$

where step ① uses Lemma B.8(a). We further derive the following results:

$$r_t = 1, \ 0 \leq \sum_{j=0}^{q-1} \mathbb{U}^j$$

$$r_t = 2, \ 0 \leq \sum_{j=q}^{2q-1} \mathbb{U}^j$$

$$r_t = 3, \ 0 \leq \sum_{j=2q}^{3q-1} \mathbb{U}^j$$

$$\cdots$$

$$r_t = s, \ 0 \leq \sum_{j=sq}^{sq-1} \mathbb{U}^j.$$

Assume that $T = sq$, where $s \geq 0$ is an integer. Summing all these inequalities together yields:

$$0 \leq \sum_{t=0}^{T-1} \mathbb{U}^t$$

$$\overset{①}{=} \sum_{t=0}^{T-1} V_t(\mathcal{Z}_t - \mathcal{Z}_{t+1}) + c_2'\mathbb{E}[\sum_{t=0}^{T-1}\mathbb{S}_1^t] - c_1\underbrace{\mathbb{E}[\sum_{t=0}^{T-1}\|\mathbf{r}^t\|_2^2]}_{\triangleq \mathcal{R}_{T-1}} + \frac{c_3}{q}\cdot q' \cdot \sum_{t=0}^{T-1} V_t Y_t$$

$$\overset{②}{\leq} V_{T-1}[\max_{t=0}^{T-1}\mathcal{Z}_t] + c_2's_1\mathbb{E}[\mathcal{V}_T] - c_1\mathbb{E}[\mathcal{R}_{T-1}] + \frac{c_3}{q}\cdot q' \cdot \sum_{t=0}^{T-1} V_t Y_t$$

$$\overset{③}{\leq} V_T[\max_{t=0}^{T-1}\mathcal{Z}_t] + c_2's_1\mathbb{E}[\mathcal{V}_T] - \frac{c_1}{\alpha+\beta}\mathbb{E}[\mathcal{V}_T^2 - \underline{v}^2] + \frac{c_3}{q}\cdot q' \cdot u_1 \cdot \mathbb{E}[\mathcal{V}_{T-1}]$$

$$\overset{④}{\leq} \mathbb{E}[\mathcal{V}_T][\max_{t=0}^{T-1}\mathcal{Z}_t] + c_2's_1\mathbb{E}[\mathcal{V}_T] - \frac{c_1}{\alpha+\beta}(\mathbb{E}[\mathcal{V}_T])^2 + \frac{c_1}{\alpha+\beta}\underline{v}^2 + \frac{c_3}{q}\cdot q' \cdot u_1 \cdot \mathbb{E}[\mathcal{V}_T], \quad (37)$$

where step ① uses the definition of $\mathbb{U}^t$; step ② uses Lemma A.4, and Lemma B.4(a); step ③ uses the upper bound for $\mathcal{R}_T$ that $\mathcal{R}_T \triangleq (\mathcal{V}_{T+1}^2 - \underline{v}^2)/(\alpha + \beta)$, as shown in Lemma B.1(a); step ④ uses $V_{t+1} \leq \mathbf{v}^{t+1} \leq \mathcal{V}_{t+1}$ (as shown in Lemma B.1(a)), and the fact that $(\mathbb{E}[v])^2 \leq \mathbb{E}[v^2]$ for any random variable $v$ (which is a direct consequence of the Cauchy-Schwarz inequality in probability theory).

We have from Inequality (37):

$$\frac{c_1}{\alpha+\beta}(\mathbb{E}[\mathcal{V}_T])^2 \leq \left(\frac{c_3}{q}q'u_1 + [\max_{t=0}^{T-1}\mathcal{Z}_t] + c_2's_1\right)\cdot\mathbb{E}[\mathcal{V}_T] + \frac{c_1}{\alpha+\beta}\underline{v}^2.$$

By applying Lemma A.3 with the parameters $a = \frac{c_1}{\alpha+\beta}$, $b = \frac{c_3}{q}q'u_1 + [\max_{t=0}^{T-1}\mathcal{Z}_t] + c_2's_1$, $c = \frac{c_1}{\alpha+\beta}\underline{v}^2$, and $x = \mathbb{E}[\mathcal{V}_T]$, we have, for all $T \geq 0$:

$$\mathbb{E}[\mathcal{V}_T] \leq \sqrt{c/a} + b/a = \underline{v} + \frac{\alpha+\beta}{c_1}\cdot(\frac{c_3}{q}q'u_1 + [\max_{t=0}^{T-1}\mathcal{Z}_t] + c_2's_1)$$

$$= \underbrace{\underline{v} + \frac{\alpha+\beta}{c_1}\cdot(\frac{c_3}{q}q'u_1 + c_2's_1)}_{\triangleq w_1} + \underbrace{\frac{\alpha+\beta}{c_1}}_{\triangleq w_2}\cdot[\max_{t=0}^{T-1}\mathcal{Z}_t]. \quad (38)$$

The upper bound for $\mathbb{E}[\mathcal{V}_T]$ is established in Inequality (38); however, it involves an unknown variable $(\max_{t=0}^{T-1}\mathcal{Z}_t)$.

**Part (b)**. We now prove that $(\max_{t=0}^{T-1}\mathcal{Z}_t)$ is always bounded above by a universal constant $\overline{\mathcal{Z}}$. Dropping the negative term $-\frac{c_1}{\min(\mathbf{v}^t)}\|\mathbf{r}^t\|_2^2$ on the right-hand side of Inequality (35), and summing over $t$ from $(r_t - 1)q$ to $t$ where $t \leq r_t q - 1$ yields:

$$0 \leq \mathbb{E}[\sum_{j=(r_t-1)q}^{t}(\mathcal{Z}_j - \mathcal{Z}_{j+1} + c_2'\mathbb{S}_2^t) + \frac{c_3}{q}\sum_{j=(r_t-1)q}^{t}\sum_{i=(r_j-1)q}^{j-1} Y_i]$$

$$\overset{①}{\leq} \mathbb{E}[\sum_{j=(r_t-1)q}^{t}[\mathcal{Z}_j - \mathcal{Z}_{j+1}] + c_2'\mathbb{S}_2^t + c_3\frac{q-1}{q}\sum_{j=(r_t-1)q}^{t} Y_i]$$

$$\leq \mathbb{E}[\underbrace{\sum_{j=(r_t-1)q}^{t}[\mathcal{Z}_j - \mathcal{Z}_{j+1}] + c_2'\mathbb{S}_2^t + c_3\sum_{j=(r_t-1)q}^{t} Y_i}_{\triangleq \mathbb{K}^i}],$$

where step ① uses Lemma B.8(**b**). We further derive the following results:

$$r_t = 1, \ 0 \leq \mathbb{E}[\sum_{j=0}^{q-1} \mathbb{K}^j]$$
$$r_t = 2, \ 0 \leq \mathbb{E}[\sum_{j=q}^{2q-1} \mathbb{K}^j]$$
$$r_t = 3, \ 0 \leq \mathbb{E}[\sum_{j=2q}^{3q-1} \mathbb{K}^j]$$
$$\cdots$$
$$r_t = s, \ 0 \leq \mathbb{E}[\sum_{j=sq}^{sq-1} \mathbb{K}^j].$$

Assume that $T = sq$, where $s \geq 0$ is an integer. Summing all these inequalities together yields:

$$
\begin{aligned}
\mathcal{Z}_T \ &\leq \ \mathcal{Z}_T + \mathbb{E}[\sum_{t=0}^{T-1} \mathbb{K}^t] \\
&\overset{①}{=} \ \mathcal{Z}_T + \mathbb{E}[\sum_{t=0}^{T-1}(\mathcal{Z}_t - \mathcal{Z}_{t+1})] + c_2' \mathbb{E}[\sum_{t=0}^{T-1} \mathbb{S}_2^t] + c_3 \sum_{t=0}^{T-1} Y_t] \\
&\overset{②}{\leq} \ \mathcal{Z}_T + (\mathcal{Z}_0 - \mathcal{Z}_T) + c_2' s_2 \mathbb{E}[\sqrt{\mathcal{V}_T}] + c_3 u_2 \mathbb{E}[\sqrt{\mathcal{V}_T}] \\
&= \ \mathcal{Z}_0 + (c_2' s_2 + c_3 u_2) \cdot \mathbb{E}[\sqrt{\mathcal{V}_T}] \\
&\overset{③}{\leq} \ \mathcal{Z}_0 + (c_2' s_2 + c_3 u_2) \cdot \sqrt{\mathbb{E}[\mathcal{V}_T]} \\
&\overset{④}{\leq} \ \mathcal{Z}_0 + (c_2' s_2 + c_3 u_2) \cdot \sqrt{w_1 + w_2 \max_{t=0}^{T-1} \mathcal{Z}_t} \\
&\overset{⑤}{\leq} \ \underbrace{\mathcal{Z}_0 + (c_2' s_2 + c_3 u_2) \cdot \sqrt{w_1}}_{\triangleq \dot{a}} + \underbrace{(c_2' s_2 + c_3 u_2) \cdot \sqrt{w_2}}_{\triangleq \dot{b}} \cdot \sqrt{\max_{t=0}^{T-1} \mathcal{Z}_t} \qquad (39) \\
&\overset{⑥}{\leq} \ \max(\mathcal{Z}_0, 2\dot{b}^2 + 2\dot{a}) = 2\dot{b}^2 + 2\dot{a} \triangleq \overline{\mathcal{Z}}, \qquad (40)
\end{aligned}
$$

where step ① uses the definition of $\mathbb{K}^t$; step ② uses $\sum_{t=0}^{T} \mathbb{S}_t^2 \leq s_2 \sqrt{\mathcal{V}_{T+1}}$ (as shown in Lemma B.4(**b**)), and $\sum_{t=0}^{T-1} Y_t \leq u_2 \mathbb{E}[\sqrt{\mathcal{V}_{T+1}}]$ (as shown in Lemma B.7(**b**)); step ③ uses $\mathbb{E}[\sqrt{x}] \leq \sqrt{\mathbb{E}[x]}$ for all $x \geq 0$, which can be derived by Jensen's inequality for the convex function $f(x) = -\sqrt{x}$ with $x \geq 0$; step ④ uses Inequality (38); step ⑤ uses $\sqrt{a+b} \leq \sqrt{a} + \sqrt{b}$ for all $a, b \geq 0$; step ⑥ uses Lemma A.7. This further leads to:

$$
\begin{aligned}
\mathcal{Z}_T \ &\leq \ 2\dot{b}^2 + 2\dot{a} \\
&\leq \ 2((c_2' s_2 + c_3 u_2) \cdot \sqrt{w_2})^2 + 2\mathcal{Z}_0 + 2(c_2' s_2 + c_3 u_2) \cdot \sqrt{w_1} \\
&= \ \mathcal{O}(L^2 \underline{v}^{-1}) + \mathcal{O}(\mathcal{Z}_0) + \mathcal{O}(L\underline{v}^{-1/2}) \cdot \left(\mathcal{O}(\sqrt{\underline{v}}) + \mathcal{O}(\sqrt{L})\right) \triangleq \overline{\mathcal{Z}} \qquad (41) \\
&\leq \ \mathcal{O}(1),
\end{aligned}
$$

where we use $s_1 \triangleq 2\dot{\kappa} = \mathcal{O}(1)$, $s_2 \triangleq \frac{4\dot{\kappa}^2}{\alpha} \underline{v}^{-1/2} = \mathcal{O}(\underline{v}^{-1/2})$, $c_1 \triangleq \frac{1}{2}(\frac{1-\theta}{\kappa})^2 = \mathcal{O}(1)$, $c_2' \triangleq \frac{(3+\phi)L}{2} = \mathcal{O}(L)$, $c_3 \triangleq \frac{L}{2\phi}\frac{q}{b}$, $w_2 \triangleq \frac{\alpha+\beta}{c_1} = \mathcal{O}(1)$, $w_1 = \underline{v} + \frac{\alpha+\beta}{c_1} \cdot (\frac{c_3}{q} q' u_1 + c_2' s_1) = \mathcal{O}(\underline{v}) + \mathcal{O}(L)$.

**Part (c).** Finally, we derive the following inequalities for all $T \geq 0$:

$$
\begin{aligned}
\mathbb{E}[\mathcal{V}_T] \ &\overset{①}{\leq} \ w_1 + w_2 \cdot [\max_{t=0}^{T} \mathcal{Z}_t] \\
&\overset{②}{\leq} \ w_1 + w_2 \overline{\mathcal{Z}} \\
&\leq \ \mathcal{O}(\underline{v}) + \mathcal{O}(L) + \mathcal{O}(L^2 \underline{v}^{-1}) + \mathcal{O}(\mathcal{Z}_0) + \mathcal{O}(L\underline{v}^{-1/2}) \cdot \left(\mathcal{O}(\sqrt{\underline{v}}) + \mathcal{O}(\sqrt{L})\right) \triangleq \overline{\mathrm{v}} \\
&\leq \ \mathcal{O}(1),
\end{aligned}
$$

where step ① uses Inequality (38); step ② uses Inequality (41).

$\square$

### B.3.2 PROOF OF THEOREM 3.8

*Proof.* We define $Y_i \triangleq \mathbb{E}[\|\mathbf{y}^{i+1} - \mathbf{y}^i\|_2^2]$.

**Part (a)**. We have the following inequality:

$$\mathbb{E}[\sum_{t=0}^{T} \|\mathbf{x}^{t+1} - \mathbf{x}^t\|_2^2] \quad \leq \quad \tfrac{1}{\underline{v}^2}\tfrac{1}{\alpha}(\overline{v}^2 - \underline{v}^2) \triangleq \overline{\mathrm{X}}, \tag{42}$$

where we employ the same strategies used in deriving Inequality (25).

**Part (b)**. First, we have the following inequalities:

$$\sum_{t=0}^{T} \mathbb{E}[\|\mathbf{g}^t - \nabla f(\mathbf{y}^t)\|_2^2] \overset{①}{\leq} \sum_{t=0}^{T}\left(\tfrac{L^2}{b}\sum_{i=(r_t-1)q}^{t-1} Y_i\right)$$

$$\overset{②}{\leq} \tfrac{L^2}{b}\cdot(q-1)\cdot\sum_{t=0}^{T} Y_i$$

$$\overset{③}{\leq} \tfrac{L^2}{b}\cdot(q-1)\cdot u_2\sqrt{\mathcal{V}_{T+1}}$$

$$\overset{④}{\leq} \tfrac{L^2}{b}\cdot(q-1)\cdot u_2\sqrt{\overline{\mathcal{V}}} = \mathcal{O}(1), \tag{43}$$

where step ① uses Lemma B.6**(a)**; step ② uses Lemma B.8**(c)**; step ③ uses Lemma B.7**(b)**; step ④ uses $\mathcal{V}_t \leq \overline{\mathcal{V}}$ for all $t$.

Second, we obtain the following results:

$$\sum_{t=0}^{T} \|\mathbf{y}^t - \mathbf{x}^{t+1}\|_2^2 \overset{①}{=} \sum_{t=0}^{T} \|\mathbf{x}^t + \sigma^{t-1}(\mathbf{x}^t - \mathbf{x}^{t-1}) - \mathbf{x}^{t+1}\|_2^2$$

$$\overset{②}{\leq} \sum_{t=0}^{T}(\|\mathbf{x}^t - \mathbf{x}^{t+1}\|_2^2 + \|\mathbf{x}^t - \mathbf{x}^{t-1}\|_2^2)$$

$$= \sum_{t=0}^{T} \|\mathbf{x}^t - \mathbf{x}^{t+1}\|_2^2 + \sum_{t=-1}^{T-1} \|\mathbf{x}^{t+1} - \mathbf{x}^t\|_2^2$$

$$\overset{③}{\leq} 2\sum_{t=0}^{T} \|\mathbf{x}^t - \mathbf{x}^{t+1}\|_2^2$$

$$\overset{④}{\leq} 2\overline{\mathrm{X}} = \mathcal{O}(1), \tag{44}$$

where step ① uses $\mathbf{y}^t = \mathbf{x}^t + \sigma^{t-1}(\mathbf{x}^t - \mathbf{x}^{t-1})$; step ② uses $\sigma^{t-1} \leq 1$ for all $t$; step ③ uses $\mathbf{x}^{-1} = \mathbf{x}^0$; step ④ uses Inequality (42),

Third, we have the following inequalities:

$$\mathbb{E}[\sum_{t=0}^{T} \|\partial h(\mathbf{x}^{t+1}) + \nabla f(\mathbf{x}^{t+1})\|_2^2]$$

$$\overset{①}{=} \mathbb{E}[\sum_{t=0}^{T} \|\nabla f(\mathbf{x}^{t+1}) - \mathbf{g}^t - \mathbf{v}^t \odot (\mathbf{x}^{t+1} - \mathbf{y}^t)\|_2^2]$$

$$= \mathbb{E}[\sum_{t=0}^{T} \|[\nabla f(\mathbf{x}^{t+1}) - \nabla f(\mathbf{y}^t)] + [\nabla f(\mathbf{y}^t) - \mathbf{g}^t] - \mathbf{v}^t \odot (\mathbf{x}^{t+1} - \mathbf{y}^t)\|_2^2]$$

$$\overset{②}{\leq} 3\mathbb{E}[\sum_{t=0}^{T} \|\nabla f(\mathbf{x}^{t+1}) - \nabla f(\mathbf{y}^t)\|_2^2 + \sum_{t=0}^{T} \|\nabla f(\mathbf{y}^t) - \mathbf{g}^t\|_2^2 + \sum_{t=0}^{T} \|\mathbf{v}^t \odot (\mathbf{x}^{t+1} - \mathbf{y}^t)\|_2^2]$$

$$\overset{③}{\leq} \mathbb{E}[3L\sum_{t=0}^{T} \|\mathbf{x}^{t+1} - \mathbf{y}^t\|_2^2 + 3\sum_{t=0}^{T} \|\nabla f(\mathbf{y}^t) - \mathbf{g}^t\|_2^2 + 3\overline{v}\sum_{t=0}^{T} \|\mathbf{x}^{t+1} - \mathbf{y}^t\|_2^2]$$

$$\overset{③}{\leq} \mathcal{O}(1) + \mathcal{O}(1) + \mathcal{O}(1) \leq \mathcal{O}(1), \tag{45}$$

where step ① uses the first-order necessarily optimality condition that $\mathbf{0} \in \partial h(\mathbf{x}^{t+1}) + \mathbf{g}^t + \mathbf{v}^t \odot (\mathbf{x}^{t+1} - \mathbf{y}^t)$; step ② uses $\|\mathbf{a} + \mathbf{b} + \mathbf{c}\|_2^2 \leq 3(\|\mathbf{a}\|_2^2 + \|\mathbf{b}\|_2^2 + \|\mathbf{c}\|_2^2)$ for all $\mathbf{a}, \mathbf{b}, \mathbf{c} \in \mathbb{R}^n$; step ③ uses Inequalities (44) and (43).

Fourth, using the inequality $\|\mathbf{a}\|_2^2 \geq \tfrac{1}{T+1}(\|\mathbf{a}\|_1)^2$ for all $\mathbf{a} \in \mathbb{R}^{T+1}$, we deduce from Inequality (45) that

$$\mathbb{E}[\tfrac{1}{T+1}\sum_{t=0}^{T} \|\partial h(\mathbf{x}^{t+1}) + \nabla f(\mathbf{x}^{t+1})\|] = \mathcal{O}(\tfrac{1}{\sqrt{T+1}}).$$

In other words, there exists $\bar{t} \in [T]$ such that $\mathbb{E}[\|\nabla f(\mathbf{x}^{\bar{t}}) + \partial h(\mathbf{x}^{\bar{t}})\|] \leq \epsilon$, provided $T \geq \tfrac{1}{\epsilon^2}$.

**Part (c)**. Let $b$ denote the mini-batch size, and $q$ the frequency parameter of AEPG-SPIDER. Assume the algorithm converges in $T = \mathcal{O}(\tfrac{1}{\epsilon^2})$ iteration. When $\text{mod}(t, q) = 0$, the full-batch gradient $\nabla f(\mathbf{y}^t)$ is computed in $\mathcal{O}(N)$ time, occurring $\lceil\tfrac{T}{q}\rceil$ times; when $\text{mod}(t, q) \neq 0$, the mini-batch

gradient is computed in $b$ time, occurring $(T - \lceil \frac{T}{q} \rceil)$ times. Hence, the total iteration complexity is:

$$
\begin{aligned}
N \cdot \lceil \tfrac{T}{q} \rceil + b \cdot (T - \lceil \tfrac{T}{q} \rceil) \quad &\overset{}{\le} \quad N \cdot \tfrac{T+q}{q} + b \cdot T \\
&\overset{\text{①}}{\le} \quad N \cdot \tfrac{T+\sqrt{N}}{\sqrt{N}} + \sqrt{N} \cdot T \\
&\overset{\text{②}}{=} \quad \sqrt{N} \cdot \mathcal{O}(\tfrac{1}{\epsilon^2}) + N + \sqrt{N} \cdot \mathcal{O}(\tfrac{1}{\epsilon^2}),
\end{aligned}
$$

where step ① uses the choice that $q = b = \sqrt{N}$; step ② uses $T = \mathcal{O}(\tfrac{1}{\epsilon^2})$.

$\square$

## C  PROOF FOR SECTION 4

### C.1  PROOF OF LEMMA 4.4

*Proof.* We define $\mathbb{W}^t = \{\mathbf{x}^t, \mathbf{x}^{t-1}, \sigma^{t-1}, \mathbf{v}^t\}$, and $\mathbb{W} \triangleq \{\mathbf{x}, \mathbf{x}^-, \sigma, \mathbf{v}\}$.

We define $\mathcal{Z}(\mathbf{x}, \mathbf{x}', \sigma, \mathbf{v}) \triangleq F(\mathbf{x}) - F(\bar{\mathbf{x}}) + \frac{1}{2}\|\mathbf{x} - \mathbf{x}'\|_{\sigma(\mathbf{v}+L)}^2$.

First, we derive the following inequalities:

$$
\begin{aligned}
\|\partial_{\mathbf{x}} \mathcal{Z}(\mathbb{W}^{t+1})\| \quad &\overset{\text{①}}{=} \quad \|\nabla f(\mathbf{x}^{t+1}) + \partial h(\mathbf{x}^{t+1}) + \sigma^t(\mathbf{x}^{t+1} - \mathbf{x}^t) \odot (\mathbf{v}^{t+1} + L)\| \\
&\overset{\text{②}}{=} \quad \|\nabla f(\mathbf{x}^{t+1}) - \nabla f(\mathbf{y}^t) - \mathbf{v}^t \odot (\mathbf{x}^{t+1} - \mathbf{y}^t) + \sigma^t(\mathbf{x}^{t+1} - \mathbf{x}^t) \odot (\mathbf{v}^{t+1} + L)\| \\
&\overset{\text{③}}{\le} \quad (L + \max(\mathbf{v}^t))\|\mathbf{x}^{t+1} - \mathbf{y}^t\| + (L + \max(\mathbf{v}^{t+1}))\|\mathbf{x}^{t+1} - \mathbf{x}^t\| \\
&\overset{\text{④}}{\le} \quad (L + \overline{\mathcal{V}})(\|\mathbf{x}^{t+1} - \mathbf{y}^t\| + \|\mathbf{x}^{t+1} - \mathbf{x}^t\|) \\
&\overset{\text{⑤}}{=} \quad (L + \overline{\mathcal{V}})(\|\mathbf{x}^{t+1} - \mathbf{x}^t - \sigma^{t-1}(\mathbf{x}^t - \mathbf{x}^{t-1})\| + \|\mathbf{x}^{t+1} - \mathbf{x}^t\|) \\
&\overset{\text{⑥}}{\le} \quad (L + \overline{\mathcal{V}})(\|\mathbf{x}^t - \mathbf{x}^{t-1}\| + 2\|\mathbf{x}^{t+1} - \mathbf{x}^t\|),
\end{aligned} \tag{46}
$$

where step ① uses the definition of $\mathcal{Z}(\cdot, \cdot, \cdot, \cdot)$; step ② uses the first-order necessarily condition of $\mathbf{x}^{t+1}$ that $\mathbf{x}^{t+1} \in \text{Prox}_h(\mathbf{y} - \nabla f(\mathbf{y}^t) \div \mathbf{v}^t; \mathbf{v}^t) = \arg\min_{\mathbf{x}} h(\mathbf{x}) + \frac{1}{2}\|\mathbf{x} - (\mathbf{y} - \nabla f(\mathbf{y}^t) \div \mathbf{v}^t)\|_{\mathbf{v}^t}^2$, which leads to:

$$
\mathbf{0} \in \partial h(\mathbf{x}^{t+1}) + \nabla f(\mathbf{y}^t) + \mathbf{v}^t \odot (\mathbf{x}^{t+1} - \mathbf{y}^t);
$$

step ③ uses $L$-smoothness of $f(\mathbf{x})$, and $\sigma^t \le 1$; step ④ uses $\max(\mathbf{v}^t) \le \overline{\mathcal{V}}$, as shown in Lemma 3.4; step ⑤ uses $\mathbf{y}^t = \mathbf{x}^t + \sigma^{t-1}(\mathbf{x}^t - \mathbf{x}^{t-1})$; step ⑥ uses $\sigma^t \le 1$.

Second, we obtain the following result:

$$
\begin{aligned}
&\|\partial_{\mathbf{x}'} \mathcal{Z}(\mathbb{W}^{t+1})\| + |\partial_\sigma \mathcal{Z}(\mathbb{W}^{t+1})| + \|\partial_{\mathbf{v}} \mathcal{Z}(\mathbb{W}^{t+1})\| \\
&\overset{\text{①}}{=} \quad \left(\sigma^t\|(\mathbf{x}^t - \mathbf{x}^{t+1}) \odot (\mathbf{v}^{t+1} + L)\|\right) + \left(\tfrac{1}{2}\|\mathbf{x}^t - \mathbf{x}^{t+1}\|_{\mathbf{v}^{t+1}}^2\right) + \left(\tfrac{\sigma^t}{2}(\mathbf{x}^t - \mathbf{x}^{t+1}) \odot (\mathbf{x}^t - \mathbf{x}^{t+1})\right) \\
&\overset{\text{②}}{\le} \quad (L + \overline{\mathcal{V}})\|\mathbf{x}^t - \mathbf{x}^{t+1}\| + \tfrac{1}{2}\overline{\mathcal{V}}2\overline{\mathbf{x}}\|\mathbf{x}^t - \mathbf{x}^{t+1}\| + \tfrac{1}{2}2\overline{\mathbf{x}}\|\mathbf{x}^t - \mathbf{x}^{t+1}\| \\
&= \quad (L + \overline{\mathcal{V}} + \overline{\mathcal{V}}\overline{\mathbf{x}} + \overline{\mathbf{x}})\|\mathbf{x}^t - \mathbf{x}^{t+1}\|,
\end{aligned} \tag{47}
$$

where step ① uses the definition of $\mathcal{Z}(\cdot, \cdot, \cdot, \cdot)$; step ② uses $\max(\mathbf{v}^t) \le \overline{\mathcal{V}}$, and $\sigma^t \le 1$.

Finally, we have:

$$
\begin{aligned}
&\|\partial \mathcal{Z}(\mathbb{W}^{t+1})\| \\
&= \quad \sqrt{\|\partial_{\mathbf{x}} \mathcal{Z}(\mathbb{W}^{t+1})\|_2^2 + \|\partial_{\mathbf{x}'} \mathcal{Z}(\mathbb{W}^{t+1})\|_2^2 + |\partial_\sigma \mathcal{Z}(\mathbb{W}^{t+1})|^2 + \|\partial_{\mathbf{v}} \mathcal{Z}(\mathbb{W}^{t+1})\|_2^2} \\
&\overset{\text{①}}{\le} \quad \|\partial_{\mathbf{x}} \mathcal{Z}(\mathbb{W}^{t+1})\| + \|\partial_{\mathbf{x}'} \mathcal{Z}(\mathbb{W}^{t+1})\| + |\partial_\sigma \mathcal{Z}(\mathbb{W}^{t+1})| + \|\partial_{\mathbf{v}} \mathcal{Z}(\mathbb{W}^{t+1})\| \\
&\overset{\text{②}}{\le} \quad 2(L + \overline{\mathcal{V}})\|\mathbf{x}^{t+1} - \mathbf{x}^t\| + (2L + 2\overline{\mathcal{V}} + \overline{\mathcal{V}}\overline{\mathbf{x}} + \overline{\mathbf{x}})\|\mathbf{x}^t - \mathbf{x}^{t+1}\| \\
&\overset{\text{③}}{\le} \quad \vartheta\|\mathbf{x}^{t+1} - \mathbf{x}^t\| + \vartheta\|\mathbf{x}^t - \mathbf{x}^{t+1}\|,
\end{aligned}
$$

where step ① uses $\|\mathbf{x}\| \leq \|\mathbf{x}\|_1$ for all $\mathbf{x} \in \mathbb{R}^4$; step ② uses Inequalities (46) and (47); step ③ uses the choice $\vartheta \triangleq 2L + 2\overline{\mathcal{V}} + \overline{\mathcal{V}}\overline{\mathrm{x}} + \overline{\mathrm{x}}$.

$\square$

## C.2 PROOF OF THEOREM 4.7

*Proof.* We define $\mathbb{W}^t = \{\mathbf{x}^t, \mathbf{x}^{t-1}, \sigma^{t-1}, \mathbf{v}^t\}$, and $\mathbb{W} \triangleq \{\mathbf{x}, \mathbf{x}^-, \sigma, \mathbf{v}\}$.

We define $\mathcal{Z}(\mathbb{W}) \triangleq F(\mathbf{x}) - F(\bar{\mathbf{x}}) + \frac{1}{2}\|\mathbf{x} - \mathbf{x}^-\|^2_{\sigma(\mathbf{v}+L)}$, and $\mathcal{Z}(\mathbb{W}^t) \triangleq F(\mathbf{x}^t) - F(\bar{\mathbf{x}}) + \frac{1}{2}\|\mathbf{x}^t - \mathbf{x}^{t-1}\|^2_{\sigma^{t-1}(\mathbf{v}^t+L)}$.

We define $\mathcal{Z}^t \triangleq \mathcal{Z}(\mathbb{W}^t)$ and $\mathcal{Z}^\infty \triangleq \mathcal{Z}(\mathbb{W}^\infty)$.

We define $\xi \triangleq c_1 \min(\mathbf{v}^{t_\star}) - c_2 > 0$. We assume that $t \geq t_\star$.

First, since the desingularization function $\varphi(\cdot)$ is concave, we have: $\varphi(b) - \varphi(a) + (a - b)\varphi'(a) \leq 0$. Applying this inequality with $a = \mathcal{Z}^t - \mathcal{Z}^\infty$ and $b = \mathcal{Z}^{t+1} - \mathcal{Z}^\infty$, we have:

$$0 \;\geq\; [\mathcal{Z}^t - \mathcal{Z}^{t+1}] \cdot \varphi'(\mathcal{Z}^t - \mathcal{Z}^\infty) + \underbrace{\varphi(\mathcal{Z}^{t+1} - \mathcal{Z}^\infty) - \varphi(\mathcal{Z}^t - \mathcal{Z}^\infty)}_{\triangleq \varphi^{t+1} - \varphi^t}$$

$$\overset{①}{\geq}\; [\mathcal{Z}^t - \mathcal{Z}^{t+1}] \cdot \tfrac{1}{\text{dist}(0, \partial\mathcal{Z}(\mathbb{W}^t))} + \varphi^{t+1} - \varphi^t$$

$$\overset{②}{\geq}\; [\mathcal{Z}^t - \mathcal{Z}^{t+1}] \cdot \tfrac{1}{\vartheta(\|\mathbf{x}^t - \mathbf{x}^{t-1}\| + \|\mathbf{x}^{t-1} - \mathbf{x}^{t-2}\|)} + \varphi^{t+1} - \varphi^t, \tag{48}$$

where step ① uses Lemma 4.2 that $\frac{1}{\varphi'(\mathcal{Z}(\mathbb{W}^t) - \mathcal{Z}(\mathbb{W}^\infty))} \leq \text{dist}(0, \partial\mathcal{Z}(\mathbb{W}^t))$, which is due to our assumption that $\mathcal{Z}(\mathbb{W})$ is a KL function; step ② uses Lemma 4.4 that $\|\partial\mathcal{Z}(\mathbb{W}^{t+1})\| \leq \vartheta(\|\mathbf{x}^{t+1} - \mathbf{x}^t\| + \|\mathbf{x}^t - \mathbf{x}^{t-1}\|)$.

**Part (a)**. We derive the following inequalities:

$$\|\mathbf{x}^{t+1} - \mathbf{x}^t\|_2^2 \overset{①}{\leq} \tfrac{1}{\min(\mathbf{v}^t)^2} \cdot \|\mathbf{v}^t \odot (\mathbf{x}^{t+1} - \mathbf{x}^t)\|_2^2$$

$$\overset{②}{\leq} \tfrac{1}{\xi\min(\mathbf{v}^t)^2} \cdot \|\mathbf{r}^t\|_2^2 \cdot \xi$$

$$\overset{③}{=} \tfrac{1}{\xi\min(\mathbf{v}^t)^2} \cdot \|\mathbf{r}^t\|_2^2 \cdot (c_1\min(\mathbf{v}^{t_\star}) - c_2)$$

$$\overset{④}{\leq} \tfrac{1}{\xi\min(\mathbf{v}^t)^2} \cdot \|\mathbf{r}^t\|_2^2 \cdot (c_1\min(\mathbf{v}^t) - c_2)$$

$$\overset{⑤}{=} \tfrac{1}{\xi}\left(\tfrac{c_1\|\mathbf{r}^t\|_2^2}{\min(\mathbf{v}^t)} - \tfrac{c_2\|\mathbf{r}^t\|_2^2}{\min(\mathbf{v}^t)^2}\right) = \tfrac{1}{\xi}\left(c_1\mathbb{S}_1^t - c_2\mathbb{S}_2^t\right)$$

$$\overset{⑥}{\leq} \tfrac{1}{\xi}\left(\mathcal{Z}^t - \mathcal{Z}^{t+1}\right) = \tfrac{1}{\xi}\left(\mathcal{Z}(\mathbb{W}^t) - \mathcal{Z}(\mathbb{W}^{t+1})\right)$$

$$\overset{⑦}{\leq} \tfrac{\vartheta}{\xi}(\varphi^t - \varphi^{t+1}) \cdot (\|\mathbf{x}^t - \mathbf{x}^{t-1}\| + \|\mathbf{x}^{t-1} - \mathbf{x}^{t-2}\|), \tag{49}$$

where step ① uses $\|\mathbf{d}^t\|\min(\mathbf{v}^t) \leq \|\mathbf{d}^t \odot \mathbf{v}^t\|$; step ② uses $\mathbf{v}^t \odot (\mathbf{x}^{t+1} - \mathbf{x}^t) = \mathbf{v}^t$; step ③ uses the definition of $\xi$; step ④ uses $t \geq t_\star$; step ⑤ uses the definitions of $\{\mathbb{S}_2^t, \mathbb{S}_1^t\}$; step ⑥ uses Lemma B.3 with $\mathbf{g}^t = \nabla f(\mathbf{y}^t)$; step ⑦ uses Inequality (48).

**Part (b)**. In view of Inequality (49), we apply Lemma A.8 with $P_t = \frac{\vartheta}{\xi}\varphi_t$ (satisfying $P_t \geq P_{t+1}$). Then for all $i \geq t$,

$$S_t \triangleq \textstyle\sum_{j=t}^\infty X_{j+1} \leq \varpi(X_t + X_{t-1}) + \varpi\varphi_t,$$

where $\varpi = \max(1, \frac{4\vartheta}{\xi})$.

$\square$

## C.3 PROOF OF THEOREM 4.8

*Proof.* We define $\varphi^t \triangleq \varphi(s^t)$, where $s^t \triangleq \mathcal{Z}(\mathbb{W}^t) - \mathcal{Z}(\mathbb{W}^\infty)$.

We define $X_{t+1} \triangleq \|\mathbf{x}^{t+1} - \mathbf{x}^t\|$, and $S_i = \sum_{j=i}^{\infty} X_{j+1}$.

We let $X_i \triangleq \sqrt{\sum_{j=iq-q}^{iq-1} \|\mathbf{x}^{j+1} - \mathbf{x}^j\|_2^2}$, $S_t \triangleq \sum_{j=t}^{\infty} X_j$.

First, for any $T \geq t \geq 0$, using the triangle inequality, we have: $\|\mathbf{x}^t - \mathbf{x}^T\| \leq \sum_{j=t}^{T-1} \|\mathbf{x}^j - \mathbf{x}^{j+1}\|$. Letting $T \to \infty$ yields:

$$\|\mathbf{x}^t - \mathbf{x}^\infty\| \leq \sum_{j=t}^{\infty} \|\mathbf{x}^j - \mathbf{x}^{j+1}\| = \sum_{j=t}^{\infty} X_{j+1} = S_t. \tag{50}$$

Inequality (50) implies that establishing the convergence rate of $S_t$ is sufficient to demonstrate the convergence of $\|\mathbf{x}^t - \mathbf{x}^\infty\|$.

Second, we obtain the following results:

$$
\begin{aligned}
\frac{1}{\varphi'(s^t)} &\overset{①}{\leq} \|\partial \mathcal{Z}(\mathbb{W}^t)\|_{\mathsf{F}} \\
&\overset{②}{\leq} \vartheta(\|\mathbf{x}^t - \mathbf{x}^{t-1}\| + \|\mathbf{x}^{t-1} - \mathbf{x}^{t-2}\|),
\end{aligned} \tag{51}
$$

where step ① uses uses Lemma 4.2 that $\varphi'(\mathcal{Z}(\mathbb{W}^t) - \mathcal{Z}(\mathbb{W}^\infty)) \cdot \|\partial \mathcal{Z}(\mathbb{W}^t)\| \geq 1$; step ② uses Lemma 4.4.

Third, using the definition of $S_t$, we derive:

$$
\begin{aligned}
S_t &\triangleq \sum_{j=t}^{\infty} X_{j+1} \\
&\overset{①}{\leq} \varpi(X_t + X_{t-1}) + \varpi \cdot \varphi^t \\
&\overset{②}{=} \varpi(X_t + X_{t-1}) + \varpi \cdot \tilde{c} \cdot \{[s^t]^{\tilde{\sigma}}\}^{\frac{1-\tilde{\sigma}}{\tilde{\sigma}}} \\
&\overset{③}{=} \varpi(X_t + X_{t-1}) + \varpi \cdot \tilde{c} \cdot \{\tilde{c}(1-\tilde{\sigma}) \cdot \frac{1}{\varphi'(s^t)}\}^{\frac{1-\tilde{\sigma}}{\tilde{\sigma}}} \\
&\overset{④}{\leq} \varpi(X_t + X_{t-1}) + \varpi \cdot \tilde{c} \cdot \{\tilde{c}(1-\tilde{\sigma}) \cdot \vartheta \cdot (X_t + X_{t-1})\}^{\frac{1-\tilde{\sigma}}{\tilde{\sigma}}} \\
&\overset{⑤}{=} \varpi(X_t + X_{t-1}) + \varpi \cdot \tilde{c} \cdot \{\tilde{c}(1-\tilde{\sigma}) \cdot \vartheta \cdot (S_{t-2} - S_t)\}^{\frac{1-\tilde{\sigma}}{\tilde{\sigma}}} \\
&= \varpi(S_{t-2} - S_t) + \underbrace{\varpi \cdot \tilde{c} \cdot [\tilde{c}(1-\tilde{\sigma})\vartheta]^{\frac{1-\tilde{\sigma}}{\tilde{\sigma}}}}_{\triangleq \rho} \cdot \{S_{t-2} - S_t\}^{\frac{1-\tilde{\sigma}}{\tilde{\sigma}}},
\end{aligned} \tag{52}
$$

where step ① uses Theorem 4.7(**b**); step ② uses the definitions that $\varphi^t \triangleq \varphi(s^t)$, and $\varphi(s) = \tilde{c} s^{1-\tilde{\sigma}}$; step ③ uses $\varphi'(s) = \tilde{c}(1-\tilde{\sigma}) \cdot [s]^{-\tilde{\sigma}}$, leading to $[s^t]^{\tilde{\sigma}} = \tilde{c}(1-\tilde{\sigma}) \cdot \frac{1}{\varphi'(s^t)}$; step ④ uses Inequality (51); step ⑤ uses the fact that $X_t = S_{t-1} - S_t$, resulting in $S_{t-2} - S_t = (S_{t-1} - S_t) + (S_{t-2} - S_{t-1}) = X_t - X_{t-1}$.

Finally, we consider three cases for $\tilde{\sigma} \in [0, 1)$.

**Part (a)**. We consider $\tilde{\sigma} = 0$. We have the following inequalities:

$$\vartheta(\|\mathbf{x}^t - \mathbf{x}^{t-1}\| + \|\mathbf{x}^{t-1} - \mathbf{x}^{t-2}\|) \overset{①}{\geq} \frac{1}{\varphi'(s^t)} \overset{②}{=} \frac{1}{\tilde{c}(1-\tilde{\sigma}) \cdot [s^t]^{-\tilde{\sigma}}} \overset{③}{=} \frac{1}{\tilde{c}}, \tag{53}$$

where step ① from Inequality (51); step ② uses $\varphi'(s) = \tilde{c}(1-\tilde{\sigma}) \cdot [s]^{-\tilde{\sigma}}$; step ③ uses $\tilde{\sigma} = 0$.

Since $\|\mathbf{x}^t - \mathbf{x}^{t-1}\| + \|\mathbf{x}^{t-1} - \mathbf{x}^{t-2}\| \to 0$, and $\vartheta, c > 0$, Inequality (53) results in a contradiction $(\|\mathbf{x}^t - \mathbf{x}^{t-1}\| + \|\mathbf{x}^{t-1} - \mathbf{x}^{t-2}\|) \geq \frac{1}{\tilde{c}\vartheta} > 0$. Therefore, there exists $t'$ such that $\|\mathbf{x}^t - \mathbf{x}^{t-1}\| = 0$ for all $t > t' > t_\star$, ensuring that the algorithm terminates in a finite number of steps.

**Part (b)**. We consider $\tilde{\sigma} \in (0, \frac{1}{2}]$. We define $u \triangleq \frac{1-\tilde{\sigma}}{\tilde{\sigma}} \in [1, \infty)$.

We have: $S_{t-2} - S_t = X_t + X_{t-1} = \|\mathbf{x}^t - \mathbf{x}^{t-1}\| + \|\mathbf{x}^{t-1} - \mathbf{x}^{t-2}\| \leq 4\overline{x} \triangleq R$.

For all $t \geq t' > t_\star$, we have from Inequality (52):

$$
\begin{aligned}
S_t &\leq \varpi(S_{t-2} - S_t) + (S_{t-2} - S_t)^{\frac{1-\tilde{\sigma}}{\tilde{\sigma}}} \cdot \rho \\
&\overset{①}{\leq} \varpi(S_{t-2} - S_t) + (S_{t-2} - S_t) \cdot \underbrace{R^{u-1} \cdot \rho}_{\triangleq \tilde{\rho}} \\
&\leq S_{t-2} \cdot \frac{\tilde{\rho} + \varpi}{\tilde{\rho} + \varpi + 1},
\end{aligned} \tag{54}
$$

where step ① uses the fact that $\frac{x^u}{x} \leq R^{u-1}$ for all $u \geq 1$, and $x \in (0, R]$. By induction we obtain for even indices

$$S_{2T} \leq S_0 \cdot \left( \frac{\tilde{\rho}+\varpi}{\tilde{\rho}+\varpi+1} \right)^T,$$

and similarly for odd indices (up to a constant shift). In other words, the sequence $\{S_t\}_{t=0}^\infty$ converges linearly at the rate $S_t = \mathcal{O}(\dot{\varsigma}^t)$, where $\dot{\varsigma} \triangleq \sqrt{\frac{\tilde{\rho}+\varpi}{\tilde{\rho}+\varpi+1}} \in (0, 1)$.

**Part (c)**. We consider $\tilde{\sigma} \in (\frac{1}{2}, 1)$. We define $u \triangleq \frac{1-\tilde{\sigma}}{\tilde{\sigma}} \in (0, 1)$, and $\varsigma \triangleq \frac{1-\tilde{\sigma}}{2\tilde{\sigma}-1} > 0$.

We have: $S_{t-2} - S_t = X_t + X_{t-1} = \|\mathbf{x}^t - \mathbf{x}^{t-1}\| + \|\mathbf{x}^{t-1} - \mathbf{x}^{t-2}\| \leq 4\overline{\mathbf{x}} \triangleq R$.

We obtain: $S_{t-1} - S_t = X_t \|\mathbf{x}^t - \mathbf{x}^{t-1}\| \leq 2\overline{\mathbf{x}} < R$.

For all $t \geq t' > t_\star$, we have from Inequality (52):

$$
\begin{aligned}
S_t &\leq \rho \cdot (S_{t-2} - S_t)^{\frac{1-\tilde{\sigma}}{\tilde{\sigma}}} + \varpi(S_{t-2} - S_t) \\
&\overset{①}{=} \rho(S_{t-2} - S_t)^u + \varpi(S_{t-2} - S_t)^u \cdot (X_t)^{1-u} \\
&\overset{②}{\leq} \rho(S_{t-2} - S_t)^u + \varpi(S_{t-2} - S_t)^u \cdot R^{1-u} \\
&= (S_{t-2} - S_t)^u \cdot (\rho + \varpi R^{1-u}) \\
&\overset{③}{\leq} \mathcal{O}(T^{-\frac{u}{1-u}}) = \mathcal{O}(T^{-\varsigma}),
\end{aligned}
$$

where step ① uses the definition of $u$ and the fact that $S_{t-1} - S_t = X_t$; step ② uses the fact that $\max_{x \in (0, R]} x^{1-u} \leq R^{1-u}$ if $u \in (0, 1)$ and $R > 0$; step ③ uses Lemma A.11 with $c = \rho + \varpi R^{1-u}$.

$\square$

## C.4 PROOF OF THEOREM 4.12

*Proof.* We define $\mathbb{W}^t = \{\mathbf{x}^t, \mathbf{x}^{t-1}, \sigma^{t-1}, \mathbf{v}^t\}$, and $\mathbb{W} \triangleq \{\mathbf{x}, \mathbf{x}^-, \sigma, \mathbf{v}\}$.

We define $\mathcal{Z}(\mathbb{W}) \triangleq \mathcal{Z}(\mathbf{x}, \mathbf{x}^-, \sigma, \mathbf{v}) \triangleq F(\mathbf{x}) - F(\bar{\mathbf{x}}) + \frac{1}{2}\|\mathbf{x} - \mathbf{x}^-\|_{\sigma(\mathbf{v}+L)}^2$.

We define $\mathcal{Z}^t \triangleq \mathcal{Z}(\mathbb{W}^t) \triangleq \mathcal{Z}(\mathbf{x}^t, \mathbf{x}^{t-1}, \sigma^{t-1}, \mathbf{v}^t) \triangleq F(\mathbf{x}^t) - F(\bar{\mathbf{x}}) + \frac{1}{2}\|\mathbf{x}^t - \mathbf{x}^{t-1}\|_{\sigma^{t-1}(\mathbf{v}^t+L)}^2$.

We define $X_i \triangleq \sqrt{\sum_{j=iq-q}^{iq-1} \|\mathbf{x}^{j+1} - \mathbf{x}^j\|_2^2}$.

We define $\xi \triangleq c_1 \min(\mathbf{v}^{t_\star}) - c_2' - 3\xi' > 0$, where $\xi' \triangleq 5c_3$.

We define $\underline{r}_t \triangleq (r_t - 1)q$, and $\overline{r}_t \triangleq r_t q - 1$. We assume that $q \geq 2$.

We define $\mathbb{S}_1^t \triangleq \frac{\|\mathbf{r}^t\|_2^2}{\min(\mathbf{v}^t)}, \mathbb{S}_2^t \triangleq \frac{\|\mathbf{r}^t\|_2^2}{\min(\mathbf{v}^t)^2}$.

We define $\varphi^t \triangleq \varphi(\mathcal{Z}^t - \mathcal{Z}^\infty)$.

First, since $\varphi(\cdot)$ is a concave desingularization function, we have: $\varphi(b) + (a - b)\varphi'(a) \leq \varphi(a)$. Applying the inequality above with $a = \mathcal{Z}^t - \mathcal{Z}^\infty$ and $b = \mathcal{Z}^{t+1} - \mathcal{Z}^\infty$, we have:

$$
\begin{aligned}
\varphi(\mathcal{Z}^t - \mathcal{Z}^\infty) - \varphi(\mathcal{Z}^{t+1} - \mathcal{Z}^\infty) &\triangleq \varphi^t - \varphi^{t+1} \\
&\geq (\mathcal{Z}^t - \mathcal{Z}^{t+1}) \cdot \varphi'(\mathcal{Z}^t - \mathcal{Z}^\infty) \\
&\overset{①}{\geq} (\mathcal{Z}^t - \mathcal{Z}^{t+1}) \cdot \frac{1}{\text{dist}(0, \partial\mathcal{Z}(\mathbb{W}^t))} \\
&\overset{②}{\geq} (\mathcal{Z}_t - \mathcal{Z}_{t+1}) \cdot \frac{1}{\vartheta(\|\mathbf{x}^t - \mathbf{x}^{t-1}\| + \|\mathbf{x}^{t-1} - \mathbf{x}^{t-2}\|)},
\end{aligned}
\tag{55}
$$

step ① uses the inequality that $\frac{1}{\varphi'(\mathcal{Z}(\mathbb{W}^t) - \mathcal{Z}(\mathbb{W}^\infty))} \leq \text{dist}(0, \partial\mathcal{Z}(\mathbb{W}^t))$, which is due to Lemma 4.2 since $\mathcal{Z}(\mathbb{W})$ is a KL function by our assumption; step ② uses Lemma 4.4.

Second, we have the following inequalities:

$$
\begin{aligned}
\|\mathbf{y}^{t+1} - \mathbf{y}^t\|_2^2 &= \|(\mathbf{x}^{t+1} + \sigma^t \mathbf{d}^t) - (\mathbf{x}^t + \sigma^{t-1}\mathbf{d}^{t-1})\|_2^2 \\
&= \|\mathbf{x}^{t+1} - \mathbf{x}^t + \sigma^t \mathbf{d}^t - \sigma^{t-1}\mathbf{d}^{t-1}\|_2^2 \\
&= \|(1 + \sigma^t)\mathbf{d}^t - \sigma^{t-1}\mathbf{d}^{t-1}\|_2^2 \\
&\overset{①}{\leq} (1+\tau)(1+\sigma^t)^2\|\mathbf{d}^t\|_2^2 + (1+1/\tau)\|\sigma^{t-1}\mathbf{d}^{t-1}\|_2^2, \ \forall \tau > 0 \\
&\overset{②}{\leq} 5\|\mathbf{d}^t\|_2^2 + 5\|\mathbf{d}^{t-1}\|_2^2,
\end{aligned}
\tag{56}
$$

where step ① uses $\|\mathbf{a} + \mathbf{b}\|_2^2 \leq (1+\tau)\|\mathbf{a}\|_2^2 + (1+1/\tau)\|\mathbf{b}\|_2^2$ for all $\tau > 0$; step ② uses $\tau = 1/4$ and $\sigma^t \leq 1$.

**Part (a)**. For all $t$ with $\underline{r}_t \leq t \leq \overline{r}_t$, we have from Lemma B.6:

$$
\begin{aligned}
\mathcal{Z}_{t+1} - \mathcal{Z}_t &\leq \underbrace{-\frac{c_1\|\mathbf{r}^t\|_2^2}{\min(\mathbf{v}^t)}}_{=c_1 \mathbb{S}_1^t} + \underbrace{\frac{c_2'\|\mathbf{r}^t\|_2^2}{\min(\mathbf{v}^t)^2}}_{=c_2' \mathbb{S}_2^t} + \frac{c_3}{q}\sum_{i=(r_t-1)q}^{t-1}\mathbb{E}[\|\mathbf{y}^{i+1} - \mathbf{y}^i\|_2^2] \\
&\overset{①}{\leq} -\left(\frac{c_1\min(\mathbf{v}^{t_\star})\|\mathbf{r}^t\|_2^2}{\min(\mathbf{v}^t)^2} - \frac{c_2'\|\mathbf{r}^t\|_2^2}{\min(\mathbf{v}^t)^2}\right) + \frac{c_3}{q}\sum_{i=(r_t-1)q}^{t-1}\mathbb{E}[\|\mathbf{y}^{i+1} - \mathbf{y}^i\|_2^2] \\
&\overset{②}{=} -(\xi + 3\xi') \cdot \frac{\|\mathbf{r}^t\|_2^2}{\min(\mathbf{v}^t)^2} + \frac{c_3}{q}\sum_{i=(r_t-1)q}^{t-1}\mathbb{E}[\|\mathbf{y}^{i+1} - \mathbf{y}^i\|_2^2] \\
&\overset{③}{\leq} -(\xi + 3\xi') \cdot \|\mathbf{d}^t\|_2^2 + \frac{c_3}{q}\sum_{i=(r_t-1)q}^{t-1}\mathbb{E}[\|\mathbf{y}^{i+1} - \mathbf{y}^i\|_2^2],
\end{aligned}
\tag{57}
$$

where step ① uses the definition of $\min(\mathbf{v}^t) \geq \min(\mathbf{v}^{t_\star})$ for all $t \geq t_\star$; step ② uses the definition of $\xi \triangleq c_1 \min(\mathbf{v}^{t_\star}) - c_2' - 3\xi' > 0$, which leads to $c_1 \min(\mathbf{v}^{t_\star}) - c_2' = \xi + 3\xi'$; step ③ uses $\|\mathbf{r}^t\| = \|\mathbf{v}^t \odot \mathbf{d}^t\| \geq \min(\mathbf{v}^t)\|\mathbf{d}^t\|$.

Telescoping Inequality (57) over $t$ from $\underline{r}_t$ to $\overline{r}_t$, we have:

$$
\begin{aligned}
\mathbb{Z} &\triangleq \sum_{j=\underline{r}_t}^{\overline{r}_t}\mathbb{E}[\mathcal{Z}_j - \mathcal{Z}_{j+1}] \\
&\geq (\xi + 3\xi')\sum_{j=\underline{r}_t}^{\overline{r}_t}\|\mathbf{d}^j\|_2^2 - \frac{c_3}{q}\sum_{j=\underline{r}_t}^{\overline{r}_t}\sum_{i=(r_j-1)q}^{j-1}\mathbb{E}[\|\mathbf{y}^{i+1} - \mathbf{y}^i\|_2^2] \\
&\overset{①}{\geq} (\xi + 3\xi')\sum_{j=\underline{r}_t}^{\overline{r}_t}\|\mathbf{d}^j\|_2^2 - 5c_3\sum_{j=\underline{r}_t}^{\overline{r}_t}(\|\mathbf{d}^j\|_2^2 + \|\mathbf{d}^{j-1}\|_2^2) \\
&\overset{②}{=} (\xi + 2\xi')\sum_{j=\underline{r}_t}^{\overline{r}_t}\|\mathbf{d}^j\|_2^2 - \xi'\sum_{j=\underline{r}_t}^{\overline{r}_t}\|\mathbf{d}^{j-1}\|_2^2 \\
&= (\xi + 2\xi')\sum_{j=\underline{r}_t}^{\overline{r}_t}\|\mathbf{d}^j\|_2^2 - \xi'\sum_{j=\underline{r}_t-1}^{\overline{r}_t-1}\|\mathbf{d}^j\|_2^2 \\
&= -\xi'\|\mathbf{d}^{\underline{r}_t-1}\|_2^2 + (\xi + 2\xi')\|\mathbf{d}^{\overline{r}_t}\|_2^2 + (\xi + 2\xi' - \xi')\sum_{j=\underline{r}_t}^{\overline{r}_t-1}\|\mathbf{d}^j\|_2^2 \\
&\overset{③}{\geq} -\xi'\|\mathbf{d}^{\underline{r}_t-1}\|_2^2 + \left[\min(\xi + 2\xi', \xi + 2\xi' - \xi')\sum_{j=\underline{r}_t}^{\overline{r}_t}\|\mathbf{d}^j\|_2^2\right] \\
&\overset{④}{\geq} -\xi'\underbrace{\sum_{j=\underline{r}_t-q}^{\overline{r}_t-q}\|\mathbf{d}^j\|_2^2}_{\triangleq (X_{r_t-1})^2} + (\xi + \xi') \cdot \underbrace{[\sum_{j=\underline{r}_t}^{\overline{r}_t}\|\mathbf{d}^j\|_2^2]}_{\triangleq (X_{r_t})^2} \\
&= -\xi'(X_{r_t-1}^2 - X_{r_t}^2) + \xi X_{r_t}^2,
\end{aligned}
\tag{58}
$$

where step ① uses Lemma B.8(**b**) and $q - 1 < q$; step ② uses $\xi' \triangleq 5c_3$; step ③ uses the fact that $ab + cd \geq \min(a,c)(b+d)$ for all $a, b, c, d \geq 0$; step ④ uses $\|\mathbf{d}^{\underline{r}_t-1}\|_2^2 = \sum_{j=\underline{r}_t-1}^{\underline{r}_t-1}\|\mathbf{d}^j\|_2^2 \leq \sum_{j=\underline{r}_t-q}^{\overline{r}_t-q}\|\mathbf{d}^j\|_2^2$, which is due to the fact that $\underline{r}_t - 1 = \overline{r}_t - q$ and $\underline{r}_t - 1 \geq \underline{r}_t - q$.

Now now focus on the upper bound for $\mathbb{Z}$ in Inequality (58). We derive:

$$
\begin{aligned}
\mathbb{Z} \quad &\triangleq \quad \sum_{j=\underline{r}_t}^{\overline{r}_t} \mathbb{E}[\mathcal{Z}_j - \mathcal{Z}_{j+1}] = \sum_{j=\underline{r}_t}^{\overline{r}_t}[\mathcal{Z}(\mathbb{W}^j) - \mathcal{Z}(\mathbb{W}^{j+1})] \\
&\overset{①}{\leq} \quad \vartheta \cdot \sum_{j=\underline{r}_t}^{\overline{r}_t}(\varphi^j - \varphi^{j+1}) \cdot (\|\mathbf{x}^j - \mathbf{x}^{j-1}\| + \|\mathbf{x}^{j-1} - \mathbf{x}^{j-2}\|) \\
&\overset{②}{\leq} \quad \vartheta\sqrt{q} \cdot (\sum_{j=\underline{r}_t}^{\overline{r}_t}(\varphi^j - \varphi^{j+1})) \cdot (\sum_{j=\underline{r}_t}^{\overline{r}_t}\|\mathbf{x}^t - \mathbf{x}^{t-1}\| + \sum_{j=\underline{r}_t}^{\overline{r}_t}\|\mathbf{x}^{t-1} - \mathbf{x}^{t-2}\|) \\
&\overset{③}{=} \quad \vartheta\sqrt{q} \cdot (\varphi^{\underline{r}_t} - \varphi^{\overline{r}_t+1}) \cdot (\sum_{j=\underline{r}_t-1}^{\overline{r}_t-1}\|\mathbf{d}^t\| + \sum_{j=\underline{r}_t-2}^{\overline{r}_t-2}\|\mathbf{d}^t\|) \\
&\overset{④}{=} \quad \vartheta\sqrt{q} \cdot (\varphi^{(r_t-1)q} - \varphi^{r_t q}) \cdot \Big((2[\sum_{j=\underline{r}_t}^{\overline{r}_t-2}\|\mathbf{d}^t\|] + \|\mathbf{d}^{\overline{r}_t-1}\|) + \|\mathbf{d}^{\underline{r}_t-2}\| + 2\|\mathbf{d}^{\underline{r}_t-1}\|\Big) \\
&\overset{⑤}{\leq} \quad \vartheta\sqrt{q} \cdot (\varphi^{(r_t-1)q} - \varphi^{r_t q}) \cdot \Big(2[\sum_{j=\underline{r}_t}^{\overline{r}_t}\|\mathbf{d}^t\|] + \|\mathbf{d}^{\underline{r}_t-2}\| + 2\|\mathbf{d}^{\underline{r}_t-1}\|\Big) \\
&\leq \quad \vartheta\sqrt{q} \cdot (\varphi^{(r_t-1)q} - \varphi^{r_t q}) \cdot \Big(2[\sum_{j=\underline{r}_t}^{\overline{r}_t}\|\mathbf{d}^t\|] + 2[\sum_{j=\underline{r}_t-2}^{\underline{r}_t-1}\|\mathbf{d}^j\|]\Big) \\
&\overset{⑥}{=} \quad \vartheta\sqrt{q} \cdot (\varphi^{(r_t-1)q} - \varphi^{r_t q}) \cdot 2\Big([\sum_{j=\underline{r}_t}^{\overline{r}_t}\|\mathbf{d}^t\|] + [\sum_{j=\underline{r}_t-p}^{\overline{r}_t-p}\|\mathbf{d}^j\|]\Big) \\
&\overset{⑦}{=} \quad \vartheta\sqrt{q} \cdot (\varphi^{(r_t-1)q} - \varphi^{r_t q}) \cdot 2\sqrt{q} \cdot (\underbrace{\sqrt{\sum_{j=\underline{r}_t}^{\overline{r}_t}\|\mathbf{d}^t\|_2^2}}_{\triangleq X_{r_t}} + \underbrace{\sqrt{\sum_{j=\underline{r}_t-q}^{\overline{r}_t-q}\|\mathbf{d}^t\|_2^2}}_{\triangleq X_{r_t-1}}),
\end{aligned}
\tag{59}
$$

where step ① uses Inequality (55); step ② uses $\langle \mathbf{a}, \mathbf{b} \rangle \leq \sqrt{q}\|\mathbf{a}\|_1\|\mathbf{b}\|_1$ for all $\mathbf{a}, \mathbf{b} \in \mathbb{R}^q$; step ③ uses $\mathbf{d}^t = \mathbf{x}^{t+1} - \mathbf{x}^t$; step ④ uses $\underline{r}_t \triangleq (r_t - 1)q$, and $\overline{r}_t \triangleq r_t q - 1$; step ⑤ uses $(2[\sum_{j=\underline{r}_t}^{\overline{r}_t-2}\|\mathbf{d}^t\|] + \|\mathbf{d}^{\overline{r}_t-1}\|) \leq 2[\sum_{j=\underline{r}_t}^{\overline{r}_t-1}\|\mathbf{d}^t\|] \leq 2[\sum_{j=\underline{r}_t}^{\overline{r}_t}\|\mathbf{d}^t\|]$; step ⑥ uses $\underline{r}_t - 1 = \overline{r}_t - p$ and $p \geq 2$; step ⑦ uses $\|\mathbf{a}\|_1 \leq \sqrt{q}\|\mathbf{a}\|$ for all $\mathbf{a} \in \mathbb{R}^q$.

Combining Inequalities (58) and (59) yields:

$$
X_{r_t}^2 + \tfrac{\xi'}{\xi}(X_{r_t}^2 - X_{r_t-1}^2) \leq \tfrac{2q\vartheta}{\xi} \cdot (\varphi^{(r_t-1)q} - \varphi_{r_t q})(X_{r_t} - X_{r_t-1}).
$$

**Part (b)**. Applying Lemma A.9 with $j = r_t$, $P_{jq} = \frac{2q\vartheta}{\xi}\varphi_{jq}$ with $P_t \geq P_{t+1}$, we have:

$$
\forall i \geq 1, \underbrace{\sum_{t=i}^{\infty} X_t}_{\triangleq S_i} \leq \underbrace{16(\tfrac{\xi'}{\xi} + 1)}_{\triangleq \varpi} \cdot X_{i-1} + \underbrace{16(\tfrac{\xi'}{\xi} + 1)}_{\triangleq \varpi} \cdot \varphi_{(i-1)q}.
$$

$\square$

## C.5   Proof of Theorem 4.13

*Proof.* We define $\varphi^t \triangleq \varphi(s^t)$, where $s^t \triangleq \mathcal{Z}(\mathbb{W}^t) - \mathcal{Z}(\mathbb{W}^\infty)$.

We let $X_i \triangleq \sqrt{\sum_{j=iq-q}^{iq-1}\|\mathbf{x}^{j+1} - \mathbf{x}^j\|_2^2}$, $S_t \triangleq \sum_{j=t}^{\infty} X_j$.

First, for all $s > i \geq 1$, we have:

$$
\begin{aligned}
\|\mathbf{x}^{iq} - \mathbf{x}^{sq}\| \quad &\overset{①}{\leq} \quad \sum_{j=iq}^{sq-1}\|\mathbf{x}^{j+1} - \mathbf{x}^j\| \\
&\overset{②}{=} \quad \sum_{k=1}^{s-i}\Big(\sum_{l=(k+i)q-q}^{(k+i)q-1}\|\mathbf{x}^{l+1} - \mathbf{x}^l\|\Big) \\
&\overset{③}{\leq} \quad \sqrt{q}\sum_{k=1}^{s-i}\underbrace{\sqrt{\sum_{l=(k+i)q-q}^{(k+i)q-1}\|\mathbf{x}^{l+1} - \mathbf{x}^l\|_2^2}}_{\triangleq X_{k+i}} \\
&= \quad \sqrt{q}\sum_{k=1}^{s-i}X_{k+i} \\
&= \quad \sqrt{q}\sum_{k=1+i}^{s}X_k,
\end{aligned}
$$

where step ① uses the triangle inequality; step ② uses basic reduction; step ③ uses $\|\mathbf{x}\|_1 \leq \sqrt{q}\|\mathbf{x}\|$ for all $\mathbf{x} \in \mathbb{R}^q$. Letting $s \to \infty$ yields:

$$
\|\mathbf{x}^{iq} - \mathbf{x}^\infty\| \leq \sqrt{q}\sum_{k=1+i}^{\infty}X_k = \sqrt{q}S_{i+1}.
\tag{60}
$$

Inequality (60) implies that establishing the convergence rate of $S_T$ is sufficient to demonstrate the convergence of $\|\mathbf{x}^T - \mathbf{x}^\infty\|$.

Second, we obtain the following results:

$$
\begin{aligned}
\frac{1}{\varphi'(s^{(t-1)q})} \quad &\overset{①}{\leq} \quad \|\partial\mathcal{Z}(\mathbb{W}^{(t-1)q})\|_{\mathsf{F}} \\[2mm]
&\overset{②}{\leq} \quad \vartheta(\|\mathbf{x}^{(t-1)q} - \mathbf{x}^{(t-1)q-1}\| + \|\mathbf{x}^{(t-1)q-1} - \mathbf{x}^{(t-1)q-2}\|) \quad (61) \\[2mm]
&= \quad \vartheta \sum_{j=(t-1)q-2}^{(t-1)q-1} \|\mathbf{x}^{j+1} - \mathbf{x}^j\| \\[2mm]
&\overset{③}{\leq} \quad \vartheta \sum_{j=(t-1)q-q}^{(t-1)q-1} \|\mathbf{x}^{j+1} - \mathbf{x}^j\| \\[2mm]
&\overset{④}{\leq} \quad \vartheta\sqrt{q} \cdot \underbrace{\sqrt{\sum_{j=(t-1)q-q}^{(t-1)q-1} \|\mathbf{x}^{j+1} - \mathbf{x}^j\|_2^2}}_{\triangleq X_{t-1}}, \quad (62)
\end{aligned}
$$

where step ① uses Lemma 4.2 that $\varphi'(\mathcal{Z}(\mathbb{W}^t) - \mathcal{Z}(\mathbb{W}^\infty)) \cdot \|\partial\mathcal{Z}(\mathbb{W}^t)\| \geq 1$; step ② uses Lemma 4.4; step ③ uses $q \geq 2$; step ④ uses $\|\mathbf{x}\|_1 \leq \sqrt{q}\|\mathbf{x}\|$ for all $\mathbf{x} \in \mathbb{R}^q$.

Third, using the definition of $S_t$, we derive:

$$
\begin{aligned}
S_t \quad &\triangleq \quad \sum_{j=t}^\infty X_j \\[2mm]
&\overset{①}{\leq} \quad \varpi X_{t-1} + \varpi\varphi_{(t-1)q} \\[2mm]
&\overset{②}{=} \quad \varpi X_{t-1} + \varpi \cdot \tilde{c} \cdot \{[s^{(t-1)q}]^{\tilde\sigma}\}^{\frac{1-\tilde\sigma}{\tilde\sigma}} \\[2mm]
&\overset{③}{=} \quad \varpi X_{t-1} + \varpi \cdot \tilde{c} \cdot \{\tilde{c}(1-\tilde\sigma) \cdot \frac{1}{\varphi'(s^{(t-1)q})}\}^{\frac{1-\tilde\sigma}{\tilde\sigma}} \\[2mm]
&\overset{④}{\leq} \quad \varpi X_{t-1} + \varpi \cdot \tilde{c} \cdot \{\tilde{c}(1-\tilde\sigma) \cdot \vartheta\sqrt{q}X_{t-1}\}^{\frac{1-\tilde\sigma}{\tilde\sigma}} \\[2mm]
&\overset{⑤}{=} \quad \varpi(S_{t-1} - S_t) + \underbrace{\varpi \cdot \tilde{c} \cdot [\tilde{c}(1-\tilde\sigma)\vartheta\sqrt{q}]^{\frac{1-\tilde\sigma}{\tilde\sigma}}}_{\triangleq \rho} \cdot \{S_{t-1} - S_t\}^{\frac{1-\tilde\sigma}{\tilde\sigma}}, \quad (63)
\end{aligned}
$$

where step ① uses Theorem 4.7(**b**); step ② uses the definitions that $\varphi^t \triangleq \varphi(s^t)$, and $\varphi(s) = \tilde{c}s^{1-\tilde\sigma}$; step ③ uses $\varphi'(s) = \tilde{c}(1-\tilde\sigma) \cdot [s]^{-\tilde\sigma}$, leading to $[s^t]^{\tilde\sigma} = \tilde{c}(1-\tilde\sigma) \cdot \frac{1}{\varphi'(s^t)}$; step ④ uses Inequality (62); step ⑤ uses the fact that $X_{t-1} = S_{t-1} - S_t$.

Finally, we consider three cases for $\tilde\sigma \in [0, 1)$.

**Part (a)**. We consider $\tilde\sigma = 0$. We define $A_t \triangleq \|\mathbf{x}^{(t-1)q} - \mathbf{x}^{(t-1)q-1}\| + \|\mathbf{x}^{(t-1)q-1} - \mathbf{x}^{(t-1)q-2}\|$. We have:

$$
\vartheta A_t \overset{①}{\geq} \frac{1}{\varphi'(s^{(t-1)q})} \overset{②}{=} \frac{1}{\tilde{c}(1-\tilde\sigma) \cdot [s^{(t-1)q}]^{-\tilde\sigma}} \overset{③}{=} \frac{1}{\tilde{c}}, \quad (64)
$$

where step ① from Inequality (61); step ② uses $\varphi'(s) = \tilde{c}(1-\tilde\sigma) \cdot [s]^{-\tilde\sigma}$; step ③ uses $\tilde\sigma = 0$.

Since $A_t \to 0$, and $\vartheta, c > 0$, Inequality (64) results in a contradiction $A_t \geq \frac{1}{\tilde{c}\vartheta} > 0$. Therefore, there exists $t'$ such that $\|\mathbf{x}^t - \mathbf{x}^{t-1}\| = 0$ for all $t > t' > t_\star$, ensuring that the algorithm terminates in a finite number of steps.

**Part (b)**. We consider $\tilde\sigma \in (0, \frac{1}{2}]$. We define $u \triangleq \frac{1-\tilde\sigma}{\tilde\sigma} \in [1, \infty)$.

We have: $S_{t-1} - S_t = X_{t-1} = \sqrt{\sum_{j=(t-1)q-q}^{(t-1)q-1} \|\mathbf{x}^{j+1} - \mathbf{x}^j\|_2^2} \leq \sqrt{q(2\bar{\mathbf{x}})^2} \triangleq R$.

For all $t \geq t' > t_\star$, we have from Inequality (63):

$$
\begin{aligned}
S_t \quad &\leq \quad \varpi(S_{t-1} - S_t) + (S_{t-1} - S_t)^{\frac{1-\tilde\sigma}{\tilde\sigma}} \cdot \rho \\[2mm]
&\overset{①}{\leq} \quad (S_{t-1} - S_t)(\underbrace{\varpi + R^{u-1} \cdot \rho}_{\triangleq \tilde\rho}) \\[2mm]
&\leq \quad S_{t-1} \cdot \frac{\tilde\rho}{\tilde\rho+1}, \quad (65)
\end{aligned}
$$

where step ① uses the fact that $\frac{x^u}{x} \leq R^{u-1}$ for all $u \geq 1$, and $x \in (0, R]$. By induction we obtain

$$S_T \leq S_0 \cdot (\tfrac{\tilde{\rho}}{\tilde{\rho}+1})^T.$$

In other words, the sequence $\{S_t\}_{t=0}^\infty$ converges Q-linearly at the rate $S_t = \mathcal{O}(\dot{\tau}^t)$, where $\dot{\tau} \triangleq \frac{\tilde{\rho}}{\tilde{\rho}+1}$.

**Part (c)**. We consider $\tilde{\sigma} \in (\frac{1}{2}, 1)$. We define $u \triangleq \frac{1-\tilde{\sigma}}{\tilde{\sigma}} \in (0, 1)$, and $\dot{\varsigma} \triangleq \frac{1-\tilde{\sigma}}{2\tilde{\sigma}-1} > 0$.

We have: $S_{t-1} - S_t = X_{t-1} = \sqrt{\sum_{j=(t-1)q-q}^{(t-1)q-1} \|\mathbf{x}^{j+1} - \mathbf{x}^j\|_2^2} \leq \sqrt{q(2\bar{x})^2} \triangleq R$.

For all $t \geq t' > t_\star$, we have from Inequality (63):

$$
\begin{aligned}
S_t \quad &\leq \quad \rho(S_{t-1} - S_t)^{\frac{1-\tilde{\sigma}}{\tilde{\sigma}}} + \varpi(S_{t-1} - S_t) \\
&\overset{①}{=} \quad \rho(S_{t-1} - S_t)^u + \varpi(S_{t-1} - S_t)^u \cdot (X_{t-1})^{1-u} \\
&\overset{②}{\leq} \quad \rho(S_{t-1} - S_t)^u + \varpi(S_{t-1} - S_t)^u \cdot R^{1-u} \\
&= \quad (S_{t-1} - S_t)^u \cdot (\rho + \varpi R^{1-u}) \\
&\overset{③}{\leq} \quad \mathcal{O}(T^{-\frac{u}{1-u}}) = \mathcal{O}(T^{-\dot{\varsigma}}),
\end{aligned}
$$

where step ① uses the definition of $u$ and the fact that $S_{t-1} - S_t = X_{t-1}$; step ② uses the fact that $x^{1-u} \leq R^{1-u}$ for all $x \in (0, R]$, $u \in (0, 1)$, and $R > 0$; step ③ uses Lemma A.10 with $c = \rho + \varpi R^{1-u}$.

$\qquad\square$

# D ADDITIONAL EXPERIMENT DETAILS AND RESULTS

This section provides additional details and results of the experiments.

## D.1 DATASETS

We utilize eight datasets in our experiments, comprising both randomly generated data and publicly available real-world data. These datasets are represented as data matrices $\mathbf{D} \in \mathbb{R}^{\dot{m} \times \dot{d}}$. The dataset names are as follows: 'tdt2-$\dot{m}$-$\dot{d}$', '20news-$\dot{m}$-$\dot{d}$', 'sector-$\dot{m}$-$\dot{d}$', 'mnist-$\dot{m}$-$\dot{d}$', 'cifar-$\dot{m}$-$\dot{d}$', 'gisette-$\dot{m}$-$\dot{d}$', 'cnncaltech-$\dot{m}$-$\dot{d}$', and 'randn-$\dot{m}$-$\dot{d}$'. Here, randn$(m, n)$ refers to a function that generates a standard Gaussian random matrix with dimensions $m \times n$. The matrix $\mathbf{D} \in \mathbb{R}^{\dot{m} \times \dot{d}}$ is constructed by randomly selecting $\dot{m}$ examples and $\dot{d}$ dimensions from the original real-world datasets available at `http://www.cad.zju.edu.cn/home/dengcai/Data/TextData.html` and `https://www.csie.ntu.edu.tw/~cjlin/libsvm/`. We normalize the data matrix $\mathbf{D}$ to ensure it has a unit Frobenius norm using the operation $\mathbf{D} \leftarrow \mathbf{D}/\|\mathbf{D}\|_\mathsf{F}$. (*i*) For the linear eigenvalue problem, we generate the data matrix $\mathbf{C}$ using the formula $\mathbf{C} = -\mathbf{D}^\mathsf{T}\mathbf{D}$. (*ii*) For the sparse phase retrieval problem, we use the matrix $\mathbf{D}$ as the measurement matrix $\mathbf{A} \in \mathbb{R}^{m \times n}$. The observation vector $\mathbf{y} \in \mathbb{R}^m$ is generated as follows: A sparse signal $\mathbf{x} \in \mathbb{R}^n$ is created by randomly selecting a support set of size $0.1n$, with its values sampled from a standard Gaussian distribution. The observation vector $\mathbf{y}$ is then computed as $\mathbf{y} = \mathbf{u} + 0.001 \cdot \|\mathbf{u}\| \cdot \text{randn}(m, 1)$, where $\mathbf{u} = (\mathbf{Ax}) \odot (\mathbf{Ax})$.

## D.2 PROJECTION ON ORTHOGONALITY CONSTRAINTS

When $h(\mathbf{x}) = \iota_{\mathcal{M}}(\text{mat}(\mathbf{x}))$ with $\mathcal{M} \triangleq \{\mathbf{V} \,|\, \mathbf{V}^\mathsf{T}\mathbf{V} = \mathbf{I}\}$, the computation of the generalized proximal operator reduces to solving the following optimization problem:

$$\bar{\mathbf{x}} \in \arg\min_{\mathbf{x}} \tfrac{\mu}{2}\|\mathbf{x} - \mathbf{x}'\|_2^2, \; s.t. \; \text{mat}(\mathbf{x}) \in \mathcal{M} \triangleq \{\mathbf{V} \,|\, \mathbf{V}^\mathsf{T}\mathbf{V} = \mathbf{I}\}.$$

This corresponds to the nearest orthogonal matrix problem, whose optimal solution is given by $\bar{\mathbf{x}} = \text{vec}(\hat{\mathbf{U}}\hat{\mathbf{V}}^\mathsf{T})$, where $\text{mat}(\mathbf{x}') = \hat{\mathbf{U}}\text{Diag}(\mathbf{s})\hat{\mathbf{U}}^\mathsf{T}$ represents the singular value decomposition (SVD) of $\text{mat}(\mathbf{x}')$. Here, $\text{vec}(\mathbf{V})$ denotes the vector formed by stacking the column vectors of $\mathbf{V}$ with $\text{vec}(\mathbf{V}) \in \mathbb{R}^{d' \times r'}$, and $\text{mat}(\mathbf{x})$ converts $\mathbf{x} \in \mathbb{R}^{(d' \cdot r') \times 1}$ into a matrix with $\text{mat}(\text{vec}(\mathbf{V})) = \mathbf{V}$ with $\text{mat}(\mathbf{x}) \in \mathbb{R}^{d' \times r'}$.

### D.3    PROXIMAL OPERATOR FOR GENERALIZED CAPPED $\ell_1$ NORM

When $h(\mathbf{x}) = \dot\lambda \|\max(|\mathbf{x}|, \dot\tau)\|_1 + \iota_\Omega(\mathbf{x})$, where $\Omega \triangleq \{\mathbf{x} \mid \|\mathbf{x}\|_\infty \leq \dot r\}$, the generalized proximal operator reduces to solving the following nonconvex optimization problem:

$$\bar{\mathbf{x}} \in \arg\min_{\mathbf{x} \in \mathbb{R}^n} \dot\lambda \|\max(|\mathbf{x}|, \dot\tau)\|_1 + \tfrac{1}{2}\|\mathbf{x} - \mathbf{a}\|_{\mathbf{c}}^2, \; s.t. - \dot r \leq \mathbf{x} \leq \dot r.$$

This problem decomposes into $n$ dependent sub-problems:

$$\bar{\mathbf{x}}_i \in \arg\min_x q_i(x) \triangleq \tfrac{\mathbf{c}_i}{2}(x - \mathbf{a}_i)^2 + \dot\lambda |\max(|x|, \dot\tau)|, \; s.t. \; -\dot r \leq x \leq \dot r. \tag{66}$$

To simplify, we define $\mathcal{P}(x) \triangleq \max(-\dot r, \min(\dot r, x))$ and identify seven cases for $x$.

**(a)** $x_1 = 0$, $x_2 = -\dot r$, and $x_3 = \dot r$.

**(b)** $\dot r > x_4 > 0$ and $|x_4| \geq \dot\tau$. Problem (66) reduces to $\bar{\mathbf{x}}_i \in \arg\min_x q_i(x) \triangleq \tfrac{\mathbf{c}_i}{2}(x - \mathbf{a}_i)^2 + \dot\lambda x$. The optimality condition gives $x_4 = \mathbf{a}_i - \dot\lambda/\mathbf{c}_i$, and incorporating bound constraints yields $x_4 = \mathcal{P}(\mathbf{a}_i - \dot\lambda/\mathbf{c}_i)$.

**(c)** $\dot r > x_5 > 0$ and $|x_5| < \dot\tau$. Problem (66) simplifies to $\bar{\mathbf{x}}_i \in \arg\min_x q_i(x) \triangleq \tfrac{\mathbf{c}_i}{2}(x - \mathbf{a}_i)^2$, leading to $x_5 = \mathcal{P}(\mathbf{a}_i)$.

**(d)** $-\dot r < x_6 < 0$ and $|x_6| \geq \dot\tau$. Problem (66) reduces to $\bar{\mathbf{x}}_i \in \arg\min_x q_i(x) \triangleq \tfrac{\mathbf{c}_i}{2}(x - \mathbf{a}_i)^2 - \dot\lambda x$. The optimality condition gives $x_6 = \mathbf{a}_i + \dot\lambda/\mathbf{c}_i$, and incorporating bound constraints results in $x_6 = \mathcal{P}(\mathbf{a}_i + \dot\lambda/\mathbf{c}_i)$.

**(e)** $-\dot r < x_7 < 0$ and $|x_7| < \dot\tau$. Problem (66) simplifies to $\bar{\mathbf{x}}_i \in \arg\min_x q_i(x) \triangleq \tfrac{\mathbf{c}_i}{2}(x - \mathbf{a}_i)^2$, leading to $x_7 = \mathcal{P}(\mathbf{a}_i)$, identical to $x_5$.

Thus, the one-dimensional sub-problem in Problem (66) has six critical points, and the optimal solution is computed as:

$$\bar{\mathbf{x}}_i = \arg\min_x q_i(x), \; s.t. \; x \in \{x_1, x_2, x_3, x_4, x_5, x_6\}.$$

### D.4    ADDITIONAL EXPERIMENT RESULTS

We present the experimental results for AEPG-SPIDER on the sparse phase retrieval problem in Figures 3, 4, 5, 6, and for AEPG on the linear eigenvalue problem in Figures 7, 8, 9 and 10. The key findings are as follows: (*i*) The proposed method AEPG does not outperform on dense, randomly generated datasets labeled as 'randn-10000-1000' and 'randn-2000-500'. These results align with the widely accepted understanding that adaptive methods typically excel on sparse, structured datasets but may perform less efficiently on dense datasets Kingma & Ba (2015); Duchi et al. (2011); Ward et al. (2020). (*ii*) Overall, except for the dense and randomly generated datasets on the linear eigenvalue problem, the proposed method achieves state-of-the-art performance compared to existing methods in both deterministic and stochastic settings. These results reinforce the conclusions presented in the main paper.

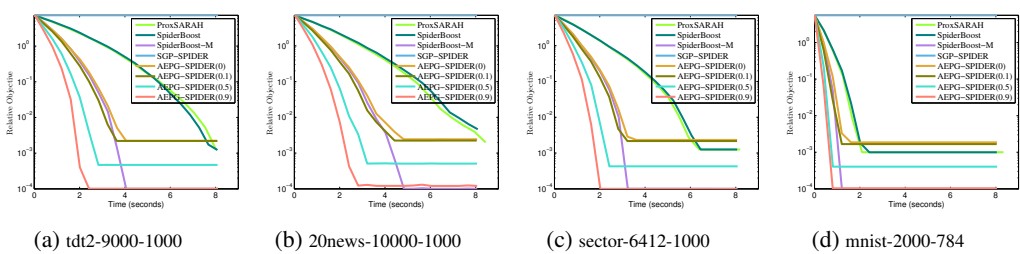

(a) tdt2-9000-1000      (b) 20news-10000-1000      (c) sector-6412-1000      (d) mnist-2000-784

Figure 3: The convergence curve for sparse phase retrieval with $\dot{\lambda} = 0.01$.

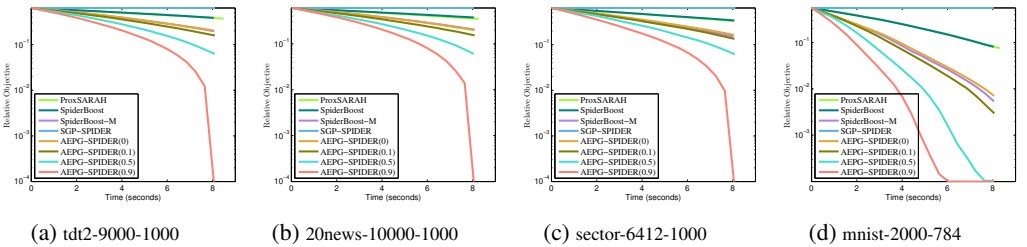

(a) tdt2-9000-1000      (b) 20news-10000-1000      (c) sector-6412-1000      (d) mnist-2000-784

Figure 4: The convergence curve for sparse phase retrieval with $\dot{\lambda} = 0.001$.

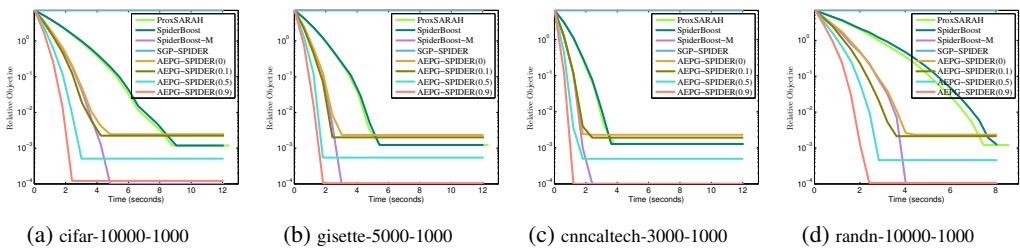

(a) cifar-10000-1000      (b) gisette-5000-1000      (c) cnncaltech-3000-1000      (d) randn-10000-1000

Figure 5: The convergence curve for sparse phase retrieval with $\dot{\lambda} = 0.01$.

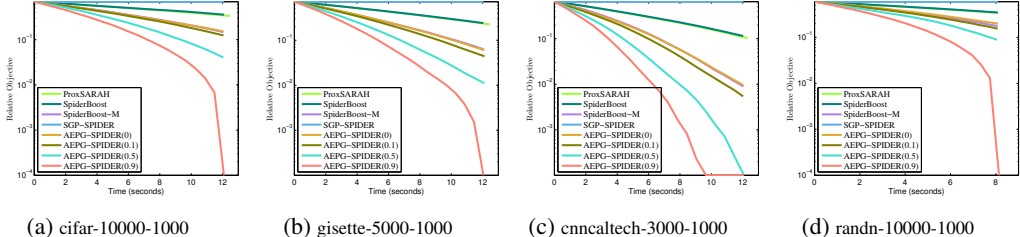

(a) cifar-10000-1000      (b) gisette-5000-1000      (c) cnncaltech-3000-1000      (d) randn-10000-1000

Figure 6: The convergence curve for sparse phase retrieval with $\dot{\lambda} = 0.001$.

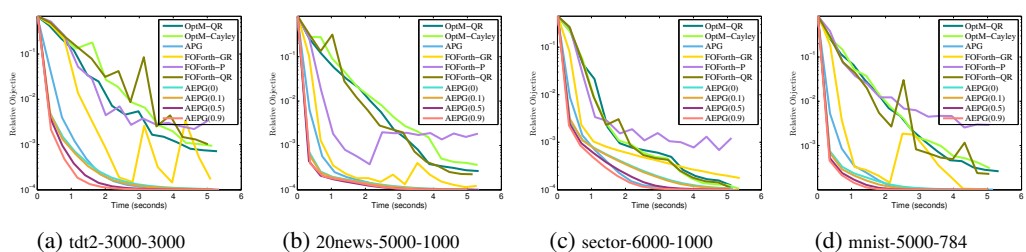

(a) tdt2-3000-3000     (b) 20news-5000-1000     (c) sector-6000-1000     (d) mnist-5000-784

Figure 7: The convergence curve for linear eigenvalue problems with $\dot{r} = 50$.

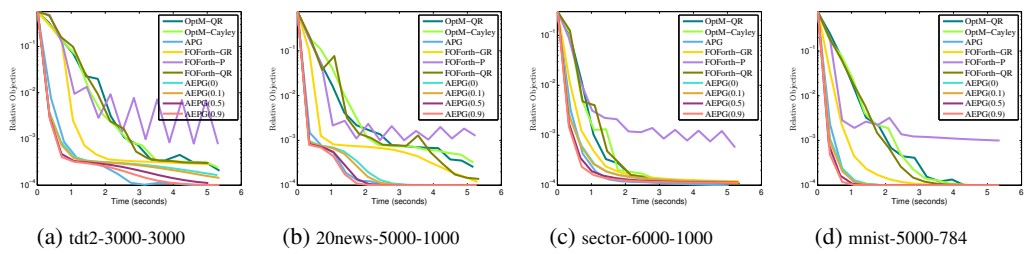

(a) tdt2-3000-3000     (b) 20news-5000-1000     (c) sector-6000-1000     (d) mnist-5000-784

Figure 8: The convergence curve for linear eigenvalue problems with $\dot{r} = 20$.

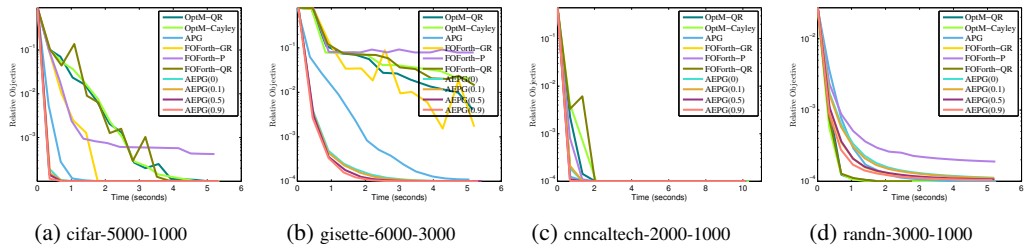

(a) cifar-5000-1000     (b) gisette-6000-3000     (c) cnncaltech-2000-1000     (d) randn-3000-1000

Figure 9: The convergence curve for linear eigenvalue problems with $\dot{r} = 20$.

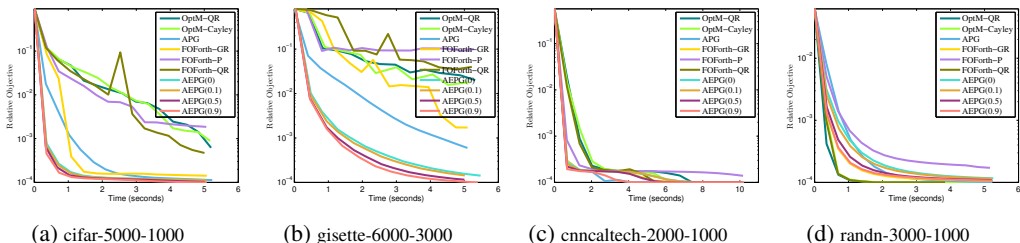

(a) cifar-5000-1000     (b) gisette-6000-3000     (c) cnncaltech-2000-1000     (d) randn-3000-1000

Figure 10: The convergence curve for linear eigenvalue problems with $\dot{r} = 50$.

