# OpenReview forum: "Adaptive Extrapolated Proximal Gradient Methods with Variance Reduction for Composite Nonconvex Finite-Sum Minimization"
_ICLR.cc/2026/Conference — Submitted to ICLR 2026_

### Official Review · Reviewer_HBTa · 2025-10-29

**Soundness:** 3
**Presentation:** 3
**Contribution:** 3
**Rating:** 2
**Confidence:** 3

**Summary:**

This paper develops AEPG-SPIDER and AEPG, which combine some well-established techniques, such as adaptive stepsizes, Nesterov extrapolation, and the SPIDER variance reducer, for composite nonconvex finite sum optimization. The authors provide theoretical analysis proving optimal iteration complexity and non-ergodic convergence rates under the KL condition.

**Strengths:**

+ The combination of adaptive stepsizes (based on iterate differences), Nesterov extrapolation, and variance reduction in a single framework for composite nonconvex problems is novel. The proposed update for the extrapolation parameter \sigma^t is creative and central to the analysis. Moving away from gradient norms to iterate differences for stepsize adaptation is a clever way to handle the nonsmooth term h(x).
+ Significant theoretical results in optimal iteration complexity and non-ergodic convergence rates under KL.
+ Clear presentation and comparison with existing work.

**Weaknesses:**

- Assumption 3.1 is a standard but potentially restrictive assumption. While commonly used in existing analysis, its necessity should be discussed since many real-world problems are unconstrained. Assumption 4.5/4/10 require the stepsize to be sufficiently large, which is non-standard and not sufficiently validated in numerical and practical experiments.
- The chosen problems (sparse phase retrieval, linear eigenvalue problems) are standard testbeds that have been used to validate proximal methods. They do not demonstrate the algorithm's performance on large-scale, modern machine learning problems (e.g., training regularized DNNs). The algorithm is a bit complicated requiring several hyperparameters (\alpha, beta, \theta, v0, and q, b). How to tune these parameters for optimal performance and the incurred computational burden shall be discussed.
- The requirement to compute a generalized proximal operator (Assumption 2.1) can be expensive for complex h(x).

**Questions:**

- To show the scalability and practility, can the algorithms be evaluated on a more contemporary DNNs (with billions/millions of parameters) learning problem?
- Why Adam or its modified/improved recent versions not included as baselines in the tests?

---

> ### Author Response · Authors · 2025-11-21
>
> Thank you for your efforts in evaluating our manuscript.
>
> ---
>
> **Q1.** Assumption 3.1 is a standard but potentially restrictive assumption. While commonly used in existing analysis, its necessity should be discussed since many real-world problems are unconstrained.
>
> **A1.** The bounded-domain (or bounded-diameter) assumption is **standard and mild**, and is widely adopted across multiple optimization settings—including convex composite minimization (Liu et al., 2022; Jaggi, 2013), non-convex composite optimization (Yun et al., 2021), fractional minimization (Yuan, 2025), and minimax problems (Xu et al., 2023).
>
> Moreover, such boundedness naturally arises in machine learning through **regularization**, a ubiquitous technique for preventing overfitting and improving generalization. Classical examples include $\ell_1$-regularized Lasso, sparse GMRF estimation, and $\ell_2$ weight decay. In deep learning, boundedness is also induced by **network pruning** (sparsity constraints) and **network quantization** (parameters constrained to discrete sets). These practices effectively enforce a bounded parameter domain, making Assumption 3.1 realistic and broadly applicable.
>
> ---
>
> **Q2.** Assumption 4.5/4/10 require the stepsize to be sufficiently large, which is non-standard and not sufficiently validated in numerical and practical experiments.
>
> **A2.**
>
> 1. Last-iterate convergence is **significantly stronger** than the usual ergodic complexity results for **nonconvex optimization**. It is standard in the literature that obtaining such guarantees requires **additional structural assumptions** on the problem or the algorithm.
>
> 2. If no sufficiently large $t_*$ satisfies Assumption 4.5/4.10, then $\min(v^t)$ must have already stabilized. Since $\min(v^t)$ is updated directly from the residuals, such stabilization indicates that the residuals have vanished and the algorithm has effectively converged.
>
> 2. Assumption 4.5/4.10 is a **local condition** designed to ensure that the convergence-rate analysis applies in the neighborhood of the limit point, i.e., for iterations $t \ge t^*$. This type of *limit-point-neighborhood assumption* is typical in convergence analysis: without restricting the analysis to such a region, the Lyapunov function may not exhibit any descent, and no meaningful **convergence rate** can be derived.
>
> ---
>
> **Q3.** The chosen problems (sparse phase retrieval, linear eigenvalue problems) are standard testbeds that have been used to validate proximal methods. They do not demonstrate the algorithm's performance on large-scale, modern machine learning problems (e.g., training regularized DNNs).
>
> **A3.** Our paper focuses primarily on establishing the **theoretical complexity guarantees** of the proposed Lipschitz-free proximal gradient methods. The advantages over AdaGrad-Norm, AGD, and ADA-SPIDER are clear: our methods **do not require tuning or knowing the Lipschitz constant**, even for **composite** optimization problems.
>
> While we evaluate our algorithms on standard proximal-method testbeds, we believe that with **appropriately chosen parameters**, the proposed method can also perform well on **large-scale modern machine learning tasks**, including regularized deep networks.
>
> ---
>
> **Q4.** The algorithm is a bit complicated requiring several hyperparameters (\alpha, beta, \theta, v0, and q, b). How to tune these parameters for optimal performance and the incurred computational burden shall be discussed.
>
> **Q4.** Thank you for raising this point. In practice, the hyperparameters of our method follow **simple and standard choices**, and the tuning burden is limited.
>
> 1. **VR parameters $(q, b)$.** For finite-sum problems, setting $q = b = \sqrt{N}$ achieves the optimal complexity and is widely adopted in SPIDER-type methods.
>
> 2. **Stepsize parameters ($\alpha$, $\beta$).** The parameter $\alpha > 0$ is typically chosen **very small** (e.g., $\alpha = 10^{-4}$), which effectively yields a coordinate-wise stepsize. The parameter $\beta$ is usually set to **0 or 1**, depending on whether coordinate-wise updates are desired.
>
> 3. **Extrapolation parameter $\theta$.** We choose $\theta$ **close to but strictly less than 1**, following standard practice in extrapolation methods.
>
> 4. **Initialization $v_0$.**  We initialize $v_0$ as a **small positive constant**, consistent with adaptive-stepsize methods such as AdaGrad-Norm.
>
> 5. **Concrete settings used in experiments.**
>
>    * **Coordinate-wise stepsizes:** $(\bar{v}, \alpha, \beta) = (0.001, 0.001, 1)$
>    * **Non-coordinate-wise stepsizes:** $(\bar{v}, \alpha, \beta) = (0.001, 0.001, 0)$
>
> Overall, these choices are straightforward and impose minimal computational burden, comparable to existing adaptive and VR-based methods.
>
> ---

---

> > ### Comment · Reviewer_HBTa · 2025-11-27
> >
> > Thanks for the clarification and efforts in addressing the issues. Since the proposed algorithm effectively combines several existing techniques to achieve improved iteration complexity and non-ergodic convergence rates. However, the analysis primarily relies on established tools, and the paper does not introduce novel advances to the optimization community. I will be keeping my score for now.

---

> ### Author Response · Authors · 2025-11-21
>
> **Q5.** The requirement to compute a generalized proximal operator (Assumption 2.1) can be expensive for complex $h(x)$.
>
> **A5.** Although computing a generalized proximal operator can be costly in general, it is efficient for many coordinate-wise separable regularizers, including $\ell_p$ norm with $p\in(0,1/2,2/3,1)$ (with or without bound constraints) and W-shaped penalties. These cases cover most sparsity-inducing functions used in practice. See L178-181.
>
> ---
>
>
> **Q6.** To show the scalability and practility, can the algorithms be evaluated on a more contemporary DNNs (with billions/millions of parameters) learning problem?
>
> **A6.** Thank you for the suggestion.
>
> 1. In the update manuscript, we will include detailed experiments on **regularized deep networks** to further demonstrate scalability.
>
> 2. We emphasize that this work is primarily **theoretical**, and its main contribution is providing the **first Lipschitz-free proximal gradient methods**.
>
> 3. The proposed algorithms are expected to perform well on large-scale modern ML tasks with appropriately chosen parameters.
>
> ---
>
> **Q7.** Why Adam or its modified/improved recent versions not included as baselines in the tests?
>
> **A7.** Adam and its variants are not included because they **cannot handle the nonsmooth composite structure** in our sparse phase-retrieval problem, i.e.,
>
> $$\min_x f(x) + \lambda \sum_{i=1}^n |\max(x_i,1)| \quad \text{s.t.} -1 \le x \le 1,$$
>
> which requires **proximal updates** to manage the nonsmooth regularizer and box constraints. Adam cannot incorporate such proximal steps, so it is not an appropriate baseline for this setting.
>
> ---

---

> ### Author Response · Authors · 2025-11-27
> **Clarifying the Novel Contributions of Our Method**
>
> While the proofs rely on standard linear algebra, the  **core algorithmic ideas and technical components** are new.
>
> Our main contributions are:
>
> 1. **Novel iterate-difference–based accumulator.**  We replace (sub)gradient-based accumulators with
> $$v^{t+1} = \sqrt{ v^0\odot v^0 + \sum_{i=0}^t || v^t \odot (x^{t+1}-x^t) ||_2^2 },$$
>  which enables **Lipschitz-free adaptivity**. This **self-normalized update rule** does not appear in any existing adaptive or proximal method.
>
> 2. **New recursive extrapolation rule.**
>   $$\sigma^t =\theta (1-\sigma^{t-1}) \min(v^t \div v^{t+1}),$$
>  coupling adaptivity and momentum in a **self-correcting** form not found in existing extrapolated or Nesterov-type schemes.
>
> 3. **A new nonlinear recursive inequality.**
>  $$Z_T \leq \dot{a} + \dot{b} \sqrt{\max_t^{T-1} Z_t },$$
>   which **removes the logarithmic factor in ADA-SPIDER** and cannot be derived from standard telescoping arguments.
>
> 4. **A new KL-compatible potential function**
> $$Z(x,x',\sigma,v) = F(x) - F(x^*) + \tfrac{1}{2}||x-x'||_{\sigma v + \sigma L}^2,$$
> tailored for **Lipschitz-free proximal gradient methods**, enabling non-ergodic guarantees.
>
> Together, these constitute the **first Lipschitz-free proximal-gradient framework** with improved complexity (up to a logarithmic factor) and non-ergodic convergence.
>
> These elements are not simple combinations of prior techniques but genuinely new algorithmic and analytical contributions.

---

### Official Review · Reviewer_Mmdw · 2025-10-31

**Soundness:** 3
**Presentation:** 3
**Contribution:** 2
**Rating:** 6
**Confidence:** 4

**Summary:**

This paper tackles constrained optimization problems of the form $\\min_x F(x):= f(x)+h(x)$, where $F$ is possibly nonconvex, $f$ is smooth and $h$ is a (nonconvex) regularizer with bounded domain. The authors propose a learning rate-free algorithm to tackle the problem in the deterministic setting, and incorporate a SPIDER type estimator for the finite sum setting. The convergence rate of the proposed methods to stationary points of $F$ in the nonconvex setting is shown and the (linear) convergence under the standard KL assumption is presented. Finally, the method is compared against state-of-the-art variance reduction methods in a number of relevant experiments.

**Strengths:**

1. The topic of the paper is interesting, since adaptive learning-rate free methods are of particular interest to the machine learning community.
2. The proposed method encompasses various techniques such as Nesterov momentum, AdaGrad type updates and variance reduction, potentially leading to faster algorithms.
3. The proposed method seems to outperform the rest of the methods in most experiments.

**Weaknesses:**

1. The quality of the presentation could improve substantially. There are a lot of inconsistencies in the notation and repetitions especially in the appendix. Please see the Questions section for some examples.
2. While many techniques are combined in the paper, the theoretical improvement over existing works in the unconstrained case is only in terms of eliminating a logarithmic term, as described in Remark 3.9.
3. The assumptions of the paper are not discussed enough and are not compared against the ones from recent papers. For example, recent works focus more on problems beyond traditional Lipschitz smoothness assumptions. More discussion is also required for Assumption 4.10 which seems tailored to the convergence analysis of the paper.
4. Although the theoretical guarantees are strong, the variance reduction techniques utilized in the paper are impractical for modern machine learning applications.

**Questions:**

**Major questions:**

1. In the experiments, how were the other methods tuned? Since the parameters of the proposed algorithms $(v, \alpha, \beta)$ differ a lot between the two experiments, the authors should report how they obtained these parameters and the difficulty of this procedure.

**Minor questions and typos:**
- Line 034: $f(\cdot)$ is used as a function and $h(x)$ which is the value of $h$ at $x$. It is better to use either $h$ or $h(\cdot)$ to denote a function. See also Line 095.
- Line 035: generalized proximal operator is stated but not defined yet. It is better to provide a link to its definition.
- Line 148: Vector $v$ in the generalized norm definition should have nonnegative elements so that the quantity is well-defined.
- Assumption 2.1: missing a verb, probably "computed".
- Line 183: "Given any solution $y^t$"?
- Line 829: Clearly, $g(0) \neq 0$ and in fact $g(0) = 1-1/p$. The result still holds since $g(0) \leq 0$ as $p \leq 1$.
- Line 834: Better to use $(x-y)f'(y)$ instead of the inner product notation.
- Line 879: "uses We have".
- Lemma A.8: the assumption should hold for all $t \in N$.
- Lemma A.10: For the r.h.s. of the inequality to be well-defined, the sequence should also be nonincreasing. This holds for the sequence where the Lemma is applied.
- Line 1075: $s^t$ is the sum of vector and scalar, a $\mathbf{1}$ is missing.
- $Z_t$ is defined in line 1159 and in line 1176 and in line 1216. This happens with many quantities in the paper.
- Line 1196: $a^t = y^t - \nabla f(y^t) / v^t$ or $a^t = y^t - g^t / v^t$?

---

> ### Author Response · Authors · 2025-11-21
>
> Thank you for the reviewer's careful reading and detailed suggestions. Below we provide point-by-point responses, following the reviewer’s wording for clarity.
>
> ---
>
> **Q1.** While many techniques are combined in the paper, the theoretical improvement over existing works in the unconstrained case is only in terms of eliminating a logarithmic term, as described in Remark 3.9.
>
> **A1.** Eliminating the logarithmic factor—achieving the same optimal rate as SPIDER—is only a **side contribution**. Our main contribution is the **first Lipschitz-free method for composite objectives**, which ADA-SPIDER cannot handle. This is enabled by our **residual-based stepsize update**, which relies on iterate differences rather than accumulated gradients—a fundamentally different adaptive mechanism (see “Existing Challenges (i)”, L105–125).
>
> ---
>
> **Q2.** The assumptions of the paper are not discussed enough and are not compared against the ones from recent papers. For example, recent works focus more on problems beyond traditional Lipschitz smoothness assumptions.
>
> **A2.**
>
> 1. The reviewer is referring to the $(L_1, L_2)$-smoothness framework used in works such as *“Why Gradient Clipping Accelerates Training: A Theoretical Justification for Adaptivity.”* That line of work relies on **both first- and second-order local smoothness parameters** to control the bias introduced by gradient clipping. In contrast, our analysis is developed under the **standard composite structure**, requiring only the usual smoothness assumption on the differentiable component (f).
>
> 2. Extending our method to weaker $(L_1,L_2)$ type generalized smoothness is indeed an interesting direction and is left for future work. We will cite and briefly discuss this related line of research in the revision.
>
> ---
>
> **Q3.** More discussion is also required for Assumption 4.10 which seems tailored to the convergence analysis of the paper.
>
> **A3.**
>
> 1. Last-iterate convergence is **significantly stronger** than the usual ergodic complexity results for **nonconvex optimization**. It is standard in the literature that obtaining such guarantees requires **additional structural assumptions** on the problem or the algorithm.
>
> 2. If no sufficiently large $t_*$ satisfies Assumption 4.10, then $\min(v^t)$ must have already stabilized. Since $\min(v^t)$ is updated directly from the residuals, such stabilization indicates that the residuals have vanished and the algorithm has effectively converged.
>
> 3. Assumption 4.10 is a **local condition** designed to ensure that the convergence-rate analysis applies in the neighborhood of the limit point, i.e., for iterations $t \ge t^*$. This type of *limit-point-neighborhood assumption* is typical in convergence analysis: without restricting the analysis to such a region, the Lyapunov function may not exhibit any descent, and no meaningful convergence rate can be derived.
>
> ---
>
> **Q4.** Although the theoretical guarantees are strong, the variance reduction techniques utilized in the paper are impractical for modern machine learning applications.
>
> **A4.** We agree that classical variance-reduction methods are not mini-batch algorithms and thus may be less convenient in some large-scale training pipelines. However, when **high-accuracy solutions** are required, variance reduction remains significantly more **sample-efficient** than standard stochastic optimization, achieving provably **better iteration complexity**. Our focus is precisely on this high-accuracy regime, where VR methods are widely used in optimization and remain highly relevant.
>
> ---
>
>
> **Q5.** Line 034: $f(\cdot)$ is used as a function and $h(x)$ which is the value of $h$ at $x$. It is better to use either $h$ or $h(\cdot)$ to denote a function. See also Line 095.
>
> **A5.** Thank you for pointing this out. We will use $h(\cdot)$ consistently to denote the function $h$.
>
> ---
>
> **Q6.** Line 035: generalized proximal operator is stated but not defined yet. It is better to provide a link to its definition.
>
>
> **A6.** We will add a pointer to the formal definition of the generalized proximal operator.
>
> ---
>
> **Q7.** Line 148: Vector $v$  in the generalized norm definition should have nonnegative elements so that the quantity is well-defined.
>
>
> **A7.** We will add the requirement $v \ge 0$ to ensure that the generalized norm is well-defined.
>
> ---
>
> **Q8.** Assumption 2.1: missing a verb, probably "computed".
>
> **A8.** Yes, it should be “exactly and efficiently computed”.
>
> ---
>
> **Q9.** Line 183: "Given any solution $y^t$"?
>
> **A9.** Yes, the statement is correct. $y^t \in \mathbb{R}^n$ can be any temporary solution.
>
> ---
>
> **Q10.** Line 829: Clearly, $g(0)\neq 0$ and in fact $g(0)=1-1/p$. The result still holds since $g(0)\leq0$ as $p\leq 1$.
>
>
> **A10.** Thank you for catching this typo. It should be $g(0)\leq 0$.
>
> ---

---

> ### Author Response · Authors · 2025-11-21
>
> **Q11.** Line 834: Better to use $(x-y)f'(y)$ instead of the inner product notation.
>
> **A11.** We will replace the inner-product notation with $(x-y)f'(y)$.
>
> ---
>
> **Q12.** Line 879: "uses We have".
>
> **A12.** It should be "step (2) uses".
>
> ---
>
> **Q13.** Lemma A.8: the assumption should hold for all $t\in N$
>
> **A13.** We will explicitly state the assumption as holding "for all integer $t \geq 0$".
>
> ---
>
> **Q14.** Lemma A.10: For the r.h.s. of the inequality to be well-defined, the sequence should also be nonincreasing. This holds for the sequence where the Lemma is applied.
>
> **A14.** Thank you for the observation. We will explicitly assume that the sequence $(S_t)_{t=0}^{\infty}$ is nonnegative and nonincreasing.
>
> ---
>
> **Q15.** Line 1075: $s^t$ is the sum of vector and scalar, a $\mathbf{1}$ is missing.
>
> **A15.** We will revise the expression to: $s^t = \alpha \|r^t\|^2 \mathbf{1} +\beta r^t \odot r^t$.
>
> ---
>
> **Q16.** $Z_t$ is defined in line 1159 and in line 1176 and in line 1216. This happens with many quantities in the paper.
>
> **A16.** Thank you for noting this redundancy. We will remove the repeated definitions of  $Z_t$.
>
> ---
>
>
> **Q17.** Line 1196: $a^t=y^t-\nabla f(y^t)/v^t$ or $a^t=y^t=g^t/v^t$?
>
> **A17.** Our definition is correct: $a^t=y^t-g^t/v^t$ ,as stated in Line 1173.
>
>
> ---

---

> > ### Comment · Reviewer_Mmdw · 2025-11-27
> >
> > Dear authors,
> >
> > Thank you for your reply. I will maintain my positive evaluation of the paper.

---

### Official Review · Reviewer_9GYz · 2025-11-01

**Soundness:** 2
**Presentation:** 1
**Contribution:** 2
**Rating:** 4
**Confidence:** 3

**Summary:**

This paper proposes an *Adaptive Extrapolated Proximal Gradient* method with variance reduction for solving composite nonconvex finite-sum optimization problems. The method integrates three mechanisms—adaptive stepsizes, Nesterov extrapolation, and the SPIDER estimator—into a unified proximal framework. The authors claim optimal iteration complexities \(O(N\epsilon^{-2})\) for AEPG and $O(N+\sqrt{N}\epsilon^{-2})$ for AEPG-SPIDER without requiring Lipschitz constants. Theoretical results under the Kurdyka–Łojasiewicz (KL) assumption provide non-ergodic convergence rates, and preliminary experiments on sparse phase retrieval and linear eigenvalue problems demonstrate faster empirical convergence.

Overall, while the paper contains several interesting elements, it is difficult to follow, motivations are not well articulated, and the technical novelty—especially in terms of complexity improvement—is limited compared to prior variance-reduced methods such as SPIDER and ADA-SPIDER.

**Strengths:**

The proposed framework unifies adaptive stepsizes, Nesterov extrapolation, and variance reduction. This shows an effort to generalize various optimization components into a single algorithmic structure.

**Weaknesses:**

-  The method combines several existing techniques (adaptive stepsize + extrapolation + variance reduction) but the motivation for this combination is vague. It is not clear what specific deficiency of prior algorithms is being addressed. The presentation is also dense and difficult to interpret, which hinders accessibility.
-    The claimed iteration complexities \(O(N\epsilon^{-2})\) and \(O(N+\sqrt{N}\epsilon^{-2})\)  are identical to existing optimal results achieved by prior methods such as SPIDER [Fang et al., 2018] and ADA-SPIDER [Kavis et al., 2022b]. Therefore, the contribution is incremental, and the extrapolation and adaptivity do not yield theoretical acceleration.

**Questions:**

Could the authors clarify the *main motivation* for combining adaptivity, extrapolation, and SPIDER? What specific limitation of ADA-SPIDER or ProxSARAH does AEPG-SPIDER overcome?

---

> ### Author Response · Authors · 2025-11-21
>
> We sincerely appreciate the reviewer’s thoughtful comments and suggestions.
>
> ---
>
> **Q1.** The method combines several existing techniques (adaptive stepsize + extrapolation + variance reduction) but the motivation for this combination is vague. It is not clear what specific deficiency of prior algorithms is being addressed.
>
> **A1.** Our motivation is precisely driven by the concrete limitations of existing methods, as summarized in Table 1:
>
> 1. **Non-adaptive PG/VR methods** (APG, SVRG-APG, ProxSVRG, SPIDER, SpiderBoost, ProxSARAH) **all require a correctly tuned Lipschitz constant**. If the stepsize is even mildly misspecified, these methods provide **no convergence guarantees**. Only **three** adaptive methods exist to date—AdaGrad-Norm, AGD, and ADA-SPIDER.
>
> 2. **All three adaptive methods fail on composite objectives**. To address this gap, our method introduces a **residual-based update rule**, using iterate differences rather than accumulated gradients to adjust the stepsize—fundamentally different from prior adaptive approaches. See "Existing Challenges(i)" in L105-125 in the manuscript.
>
> 3. Among the three adaptive algorithms, **only ADA-SPIDER handles finite-sum problems**, yet it still suffers from:
>    (i) lack of Nesterov-type extrapolation,
>    (ii) no coordinate-wise updates (unlike Adam-style methods),
>    (iii) **suboptimal iteration complexity** (up to a log factor), and
>    (iv) **no last-iterate convergence guarantees**.
>
> Our method is designed to simultaneously resolve these issues.
>
> ---
>
> **Q2.** The claimed iteration complexities (O(N\epsilon^{-2})) and (O(N+\sqrt{N}\epsilon^{-2})) are identical to existing optimal results achieved by prior methods such as SPIDER [Fang et al., 2018] and ADA-SPIDER [Kavis et al., 2022b]. Therefore, the contribution is incremental, and the extrapolation and adaptivity do not yield theoretical acceleration.
>
> **A2.** While the iteration complexities match the optimal rates, our method applies in a strictly broader regime:
>
>
> 1. **SPIDER** is not Lipschitz-free, cannot handle **composite objectives**, does not support **extrapolation**, does not allow **coordinate-wise updates**, and has **no last-iterate guarantees**.
>
> 2. **ADA-SPIDER** resolves only the Lipschitz-free issue; all other limitations remain.
>
> 3. Our method addresses **all** the shortcomings of **SPIDER** while retaining the optimal complexities.
>
>
> ---

---

### Meta-Review · Area_Chair_xoU9 · 2025-12-16

**Summary:**

The reviewers concur that the paper considers the important and challenging problem of unifying variance-reduction techniques with adaptive step sizes and composite problems. Nevertheless, several weaknesses should be addressed by the authors before being accepted to ICLR, including i) the quality of the presentation and proofs should be improved; ii) the paper introduces additional assumptions that are not discussed enough and not compared to the previous methods; iii) the adaptive nature is not convincing since the method still depends on $\alpha,$ $\beta,$ and $\theta$; the dependence on them is hidden under the Big-O notation and the choice of them is not clear from the theorems; and iv) the role of Nesterov momentum is also not obvious since in the end the authors do not get Nesterov's or other acceleration.

**Reviewer Concerns:**

The authors respond to Reviewer 9GYz and explain their contribution. This response partially addresses the raised weaknesses; however, the presentation weakness is not addressed in the comment or in the PDF. The weakness that SPIDER [Fang et al., 2018] has already obtained the optimal complexity $O(N+\sqrt{N}\epsilon^{-2})$ is still valid. Moreover, the weakness identified by Reviewer HBTa regarding the assumptions should be addressed in a more formal manner.

**Reviewer Scores:**

I am not sure that Reviewers 9GYz, Mmdw, and HBTa would change the scores.

---

### Decision · Program_Chairs · 2026-01-26

Reject